# AutoML-Agent: A Multi-Agent LLM Framework for Full-Pipeline AutoML

Patara Trirat [1]    Wonyong Jeong [1]    Sung Ju Hwang [1 2]

## Abstract

Automated machine learning (AutoML) accelerates AI development by automating tasks in the development pipeline, such as optimal model search and hyperparameter tuning. Existing AutoML systems often require technical expertise to set up complex tools, which is in general time-consuming and requires a large amount of human effort. Therefore, recent works have started exploiting large language models (LLM) to lessen such burden and increase the usability of AutoML frameworks via a natural language interface, allowing non-expert users to build their data-driven solutions. These methods, however, are usually designed only for a particular process in the AI development pipeline and do not efficiently use the inherent capacity of the LLMs. This paper proposes *AutoML-Agent*, a novel multi-agent framework tailored for full-pipeline AutoML, i.e., from data retrieval to model deployment. *AutoML-Agent* takes user's task descriptions, facilitates collaboration between specialized LLM agents, and delivers deployment-ready models. Unlike existing work, instead of devising a single plan, we introduce a retrieval-augmented planning strategy to enhance exploration to search for more optimal plans. We also decompose each plan into subtasks (e.g., data preprocessing and neural network design) each of which is solved by a specialized agent we build via prompting executing in parallel, making the search process more efficient. Moreover, we propose a multi-stage verification to verify executed results and guide the code generation LLM in implementing successful solutions. Extensive experiments on seven downstream tasks using fourteen datasets show that *AutoML-Agent* achieves a higher success rate in automating the full AutoML process, yielding systems with good performance throughout the diverse domains.

[1]DeepAuto.ai [2]KAIST, Seoul, South Korea. Correspondence to: Sung Ju Hwang <sjhwang@deepauto.ai>.

*Proceedings of the 42nd International Conference on Machine Learning*, Vancouver, Canada. PMLR 267, 2025. Copyright 2025 by the author(s).

## 1. Introduction

Automated machine learning (AutoML) has significantly reduced the need for technical expertise and human labors in developing effective data-driven solutions by automating each process in the AI development pipeline (Yao et al., 2018; Ren et al., 2020; He et al., 2021), such as feature engineering, model selection, and hyperparameter optimization (HPO). However, current AutoML systems (Gijsbers et al., 2024) often necessitate programming expertise to configure complex tools and resources, potentially creating barriers for a larger pool of users with limited skills and knowledge, such as domain experts (Sun et al. (2023); §A).

To make AutoML frameworks more accessible to non-expert users, many recent studies (Trirat et al., 2021; Viswanathan et al., 2023; Li et al., 2023; Hollmann et al., 2023b; Liu et al., 2025; Zhang et al., 2023; Shen et al., 2023; Zhang et al., 2024a; Hong et al., 2024a; Guo et al., 2024a; Yang et al., 2025; Chi et al., 2024) have suggested to use natural language interfaces with large language models (LLM) for machine learning (ML) and data science (DS) tasks. Nevertheless, these previous LLM-based AutoML frameworks only considered a limited number of tasks due to their restricted designs, either only for a process in the pipeline (e.g., feature engineering (Hollmann et al., 2023b; Li et al., 2024; Malberg et al., 2024), HPO (Liu et al., 2024a; 2025; Zhang et al., 2024a), and model selection (Zhang et al., 2023; Shen et al., 2023)) or for a specific group of downstream tasks (e.g., natural language processing (Viswanathan et al., 2023) and computer vision (Yang et al., 2025)). In addition, most methods overlook the inherent capability of LLMs to search for promising models by performing actual training of the candidate models during the search process, making it prohibitively costly and slow.

For an AutoML framework to be truly practical, it should perform end-to-end AutoML, considering both the **data aspects** (retrieval, preprocessing, and feature engineering) and **model aspects** (selection, HPO, and deployment). This is because a process in one aspect can affect subsequent processes in the other, potentially leading to suboptimal solutions when combining results from different frameworks. Meanwhile, the AutoML framework should be computationally efficient, using strategies to minimize the computational overhead during search. However, there are two main challenges in building such a framework.

**High Complexity of Planning Tasks** The planning of the entire AutoML pipeline introduces extra complexities compared to task- or problem-specific planning, primarily due to the inter-dependencies among the steps in the pipeline. For example, types of retrieved datasets affects how to design preprocessing steps and neural networks. Then, the designed network affects which particular hyper-parameters need to be optimized depending on the given downstream task. Such inter-step dependencies result in the enlarged search space since it should consider all possible combinations of inter-related steps. Besides, enabling the framework to operate across various downstream tasks exacerbates these challenges, as each has task-specific requirements.

**Challenges in Accurate Implementations** To develop a modular and extendable framework that effectively handles diverse ML tasks, it is crucial to enhance the flexibility of the LLM agent in its code generation ability, such as by decoupling the template code from the code for specific datasets. However, using LLMs to autonomously generate complete ML pipelines may lead to hallucination issues, including code incompletion, incorrect or missing dependencies, and potential undiscovered bugs (Hong et al., 2024b). Furthermore, LLMs often struggle with code generation when prompted with ambiguous task descriptions. Thus, we need accurate analysis of the requirements, and a code-generation platform that can adaptively generate code based on disambiguated requirements.

To address these challenges, we propose a novel multi-agent framework, *AutoML-Agent*, for full-pipeline AutoML from data and model search to evaluation, with strategies to tackle the complexity of the planning problem as well as accurate implementation of code. As illustrated in Figure 1, *AutoML-Agent* accepts a user's task description and coordinates multiple specialized agents to collaboratively identify an optimal ML pipeline, ultimately delivering a deployment-ready model and its inference endpoint as the output.

Specifically, to tackle the complex planning problem, we introduce a new *retrieval-augmented planning* strategy equipped with role-specific decomposition and prompting-based execution. This strategy produces multiple plans based on retrieved knowledge for a given task description, facilitating the exploration of promising plans. Moreover, it enables LLM agents to discern global (pipeline-level) and local (process-level) relationships among steps through plan decomposition, which helps them focus on their immediate sub-tasks while aligning with the user's goal. The retrieval-augmented component also simplifies extending LLMs to various downstream tasks using relevant knowledge. The prompting-based execution enhances search efficiency by exploiting LLMs' in-context learning capabilities without any further training, which could introduce additional cost. To enhance the accuracy of the implementation, we adopt structure-based prompt parsing that extracts ML-relevant

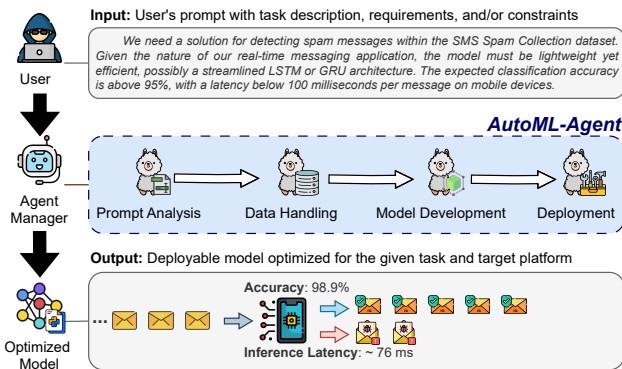

Figure 1. *AutoML-Agent* receives user's instructions and delivers optimized deployable models.

requirements from the user's description and *multi-stage verification* that provides feedback between each step in the framework to ensure the quality of instructions when guiding the LLM for code generation. These modules aim to improve the correctness and clarity of the task description for code implementation. Our **contributions** are as follows.

- We propose a novel multi-agent LLM framework for AutoML, designed to automate the entire AI development pipeline. To the best of our knowledge, this is the first attempt to employ LLMs in a task-agnostic AutoML framework that spans from data retrieval to model deployment.
- We address the challenges due to the complexity of the planning problem in full-pipeline AutoML by introducing retrieval-augmented planning with role-specific plan decomposition and prompting-based plan execution, enhancing the flexibility and efficiency of the search process.
- To enhance the accuracy of our full-pipeline implementation, we integrate structure-based prompt parsing and multi-stage verification to ensure the quality of resulting solutions and instructions prior to actual code implementation, thereby improving overall performance.
- We demonstrate the superiority of the proposed *AutoML-Agent* framework through extensive experiments on seven downstream tasks using fourteen datasets.
- We have made the source code available at https://github.com/deepauto-ai/automl-agent.

## 2. Related Work

AutoML is a transformative approach for optimizing ML workflows, enabling both practitioners and researchers to efficiently design models and preprocessing pipelines with minimal manual intervention (Ren et al., 2020; He et al., 2021; Gijsbers et al., 2024). Despite several advancements in AutoML (Jin et al., 2019; Feurer et al., 2022; Tang et al., 2024), most of them are designed only for particular elements of the ML pipeline. Only a few works (Bisong, 2019; Mukunthu et al., 2019; Microsoft, 2021) support multiple

*Table 1.* Comparison between *AutoML-Agent* and existing LLM-based frameworks.

| Framework | Key Functionality | | | | | |
|---|---|---|---|---|---|---|
| | Planning | Verification | Full Pipeline | Task-Agnostic | Training-Free Search | With Retrieval |
| AutoML-GPT (Zhang et al., 2023) | ✗ | ✗ | ✗ | ✓ | ✓ | ✗ |
| Prompt2Model (Viswanathan et al., 2023) | ✗ | ✗ | ✓ | ✗ | ✗ | ✓ |
| HuggingGPT (Shen et al., 2023) | ✓ | ✗ | ✗ | ✓ | ✓ | ✓ |
| CAAFE (Hollmann et al., 2023b) | ✗ | ✓ | ✗ | ✗ | ✗ | ✗ |
| MLCopilot (Zhang et al., 2024a) | ✗ | ✗ | ✗ | ✓ | ✓ | ✗ |
| AgentHPO (Liu et al., 2025) | ✓ | ✓ | ✗ | ✓ | ✗ | ✗ |
| Data Interpreter (Hong et al., 2024a) | ✓ | ✓ | ✗ | ✓ | ✗ | ✗ |
| DS-Agent (Guo et al., 2024a) | ✓ | ✓ | ✗ | ✓ | ✗ | ✓ |
| SELA (Chi et al., 2024) | ✓ | ✓ | ✗ | ✓ | ✗ | ✗ |
| Agent K (Grosnit et al., 2024) | ✓ | ✓ | ✗ | ✓ | ✗ | ✓ |
| AutoMMLab (Yang et al., 2025) | ✗ | ✓ | ✓ | ✗ | ✗ | ✗ |
| ***AutoML-Agent*** (Ours) | ✓ | ✓ | ✓ | ✓ | ✓ | ✓ |

steps of the pipeline. Also, due to the traditional programming interfaces, these systems often have complex configuration procedures and steep learning curves that require substantial coding expertise and an understanding of the underlying ML concepts, limiting their accessibility to non-experts and being time-consuming even for experienced users. These limitations hinder the widespread adoption of traditional AutoML systems.

LLMs (e.g., GPT-4 (Achiam et al., 2023)) have recently shown promise in addressing these limitations with the complex problem-solving skills across disciplines via human-friendly language interfaces, including AI problems (Xi et al., 2025; Narayanan et al., 2024; Fourney et al., 2024; Tang et al., 2025; Kon et al., 2025). This shift towards natural language-driven interfaces democratizes access and allows users to articulate their needs in a more intuitive manner (Tornede et al., 2024). Nevertheless, existing LLM-based frameworks can only assist in a specific step of the ML pipeline, such as feature engineering (Hollmann et al., 2023b), data analysis (Hu et al., 2024b), model search (Hong et al., 2024a; Guo et al., 2024a; Chi et al., 2024), or HPO (Liu et al., 2024a; 2025; Zhang et al., 2024a). For example, Agent K (Grosnit et al., 2024) leverages LLMs to orchestrate ML pipelines; however, it is tailored for Kaggle competition settings and relies on a training-based approach with high search overhead. A few attempts (Viswanathan et al., 2023; Yang et al., 2025) support the entire ML production pipeline, yet only for a specific type of downstream tasks. Besides, these methods either naively use the LLMs or overlook the inherent capabilities, making their search processes time-consuming for the AutoML pipeline that requires sophisticated planning and verification.

In contrast, we overcome these limitations by incorporating a new retrieval-augmented planning strategy, coupled with plan decomposition and prompting-based execution techniques, alongside structure-based prompt parsing and multi-stage verification. Table 1 summarizes the key differences between *AutoML-Agent* and existing frameworks.

## 3. AutoML-Agent

This section presents details of the proposed multi-agent framework, *AutoML-Agent*, including agent specifications, a prompt parsing module, a retrieval-augmented planning strategy, a prompting-based plan execution, and a multi-stage verification. As depicted in Figure 2, all agents are coordinated by an Agent Manager to complete the user's instructions by delivering the deployment-ready model.

### 3.1. Agent Specifications

We now provide brief descriptions of the agents in our multi-agent AutoML framework.

**Agent Manager** ($\mathcal{A}_{mgr}$) acts as the core interface between users and other specialized LLM agents within the framework, orchestrating the search process. It is responsible for interacting with the user, devising a set of global plans for subsequent processes with retrieved knowledge, distributing tasks to corresponding agents, verifying executed results with feedback, and tracking the system progress.

**Prompt Agent** ($\mathcal{A}_p$) is an LLM specifically instruction-tuned for parsing the user's instructions into a standardized JSON object with predefined keys. The parsed information is then shared across agents in the framework during the planning, searching, and verification phases.

**Data Agent** ($\mathcal{A}_d$) is an LLM prompted for doing tasks related to data manipulation and analysis. The analysis results from the Data Agent are used to inform the Model Agent about data characteristics during the model search and HPO.

**Model Agent** ($\mathcal{A}_m$) is an LLM prompted for doing tasks related to model search, HPO, model profiling, and candidate ranking. The results produced by the Model Agent are sent back to the Agent Manager for verification before proceeding to the Operation Agent.

**Operation Agent** ($\mathcal{A}_o$) is an LLM prompted for implementing the solution found by the Data and Model Agents that passes the Agent Manager's verification. The Operation

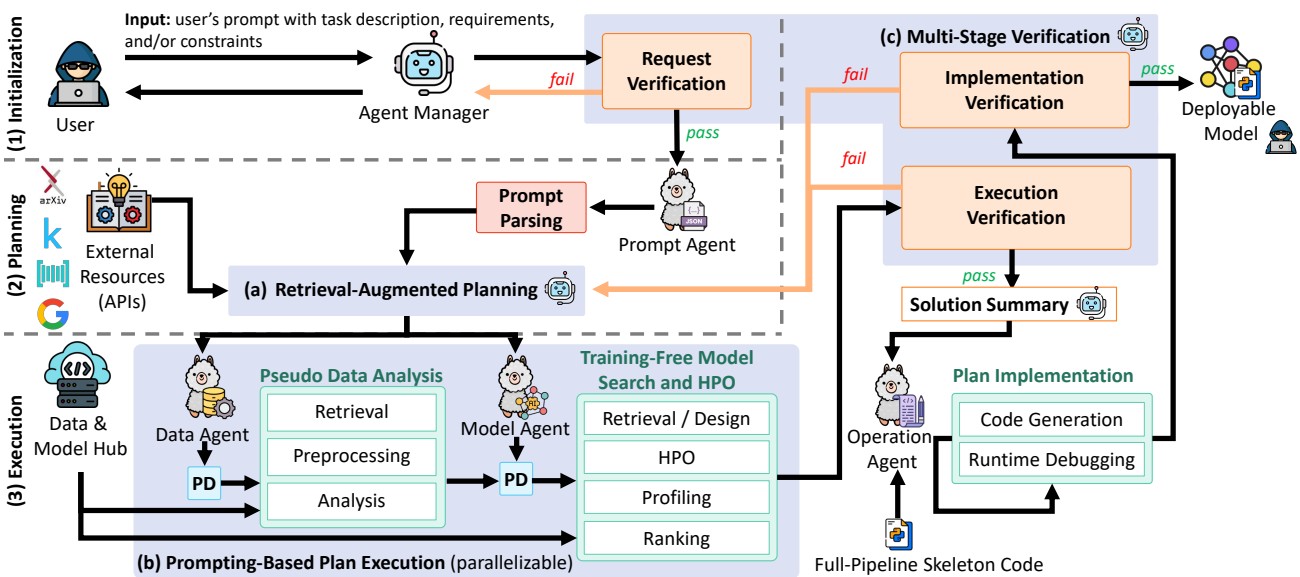

**Figure 2.** Overview of our *AutoML-Agent* framework. **(1) Initialization** stage aims to receive a valid user instruction using request verification. **(2) Planning** stage focuses on extracting ML related information by parsing the user instruction into a standardized form, and uses it to devise plans accordingly. **(3) Execution** stage executes each action given by the devised plans. Finally, based on the best execution results, *AutoML-Agent* outputs codes containing deployable model to the user.

Agent is responsible for writing effective code for actual runtime execution and recording the execution results for final verification before returning the model to the user.

After we define all agents with their corresponding profiles as described above (see §C.1 for the complete prompts), the $\mathcal{A}_{mgr}$ then assigns relevant tasks to each agent according to the user's task descriptions. Note that we can implement $\mathcal{A}_d$ and $\mathcal{A}_m$ with more than one agent per task based on the degree of parallelism.

### 3.2. Framework Overview

We present an overview of our *AutoML-Agent* framework in Figure 2 and Algorithm 1. In the **(1) initialization** stage, the Agent Manager ($\mathcal{A}_{mgr}$) receives the user instruction and checks its validity through request verification (Figure 2(c) and Line 3). In the **(2) planning** stage, the Prompt Agent ($\mathcal{A}_p$) parses the verified user instruction into a standardized JSON object. Then, $\mathcal{A}_{mgr}$ generates plans to solve the given AutoML task using retrieval-augmented planning (Figure 2(a) and Line 11). In the **(3) execution** stage, the Data ($\mathcal{A}_d$) and Model ($\mathcal{A}_m$) Agents decompose these plans and execute them via plan decomposition (PD) and prompting-based plan execution (Figure 2(b) and Line 13–16), whose results are then verified against the user's requirements via execution verification (Figure 2(c) and Line 20). Finally, $\mathcal{A}_{mgr}$ selects the best plan and sends it to the Operation Agent ($\mathcal{A}_o$) to write code (Line 22). After code generation, implementation verification (Figure 2(c) and Line 24) is conducted to ensure that the code is deployment-ready. If any of the verification steps fail, *AutoML-Agent* performs revision steps (orange lines in Figure 2) to generate new solutions.

In the following subsections, we provide the descriptions of each step more in detail.

### 3.3. Instruction Data Generation and Prompt Parsing

**Data Generation** For $\mathcal{A}_p$ to generate accurate JSON objects, we need to instruction-tune the LLM first because it can output a valid JSON object but with incorrect keys that are irrelevant to subsequent processes. Following Xu et al. (2024), we first manually create a set of high-quality seed instructions then automatically generate a larger instruction dataset $D = \{(I_i, R_i)\}_{i=1}^N$, having $N$ instruction-response pairs. Here, $I_i$ is the $i$-th instruction with the corresponding response $R_i$. We use the JSON format substantially extended from Yang et al. (2025) for response $R_i$ with the following top-level keys to extract the user's requirement from various aspects of an AutoML pipeline.

- **User**. The user key represents the user intention (e.g., build, consult, or unclear) of the given instruction and their technical expertise in AI.
- **Problem**. The problem key indicates the characteristics and requirements of the given task, including area (e.g., computer vision), downstream task (e.g., image classification), application or business domain, and other constraints like expected accuracy and inference latency.
- **Dataset**. The dataset key captures the data characteristics and properties, including data modality, requested preprocessing and augmentation techniques, and data source.
- **Model**. The model key captures the expected model characteristics and properties, including model name (e.g., ViT), family (e.g., Transformer), and type (e.g., deep neural networks).

**Algorithm 1** Overall Procedure of *AutoML-Agent*

**Initialization:** Agent Manager $\mathcal{A}_{mgr}$, instruction-tuned Prompt Agent $\mathcal{A}_p$, Data Agent $\mathcal{A}_d$, Model Agent $\mathcal{A}_m$, Operation Agent $\mathcal{A}_o$, deployment-ready model $\mathcal{M}^\star$, and system state $S$

**Input:** User instruction $I$

1: **while** $S \neq$ END **and** $\mathcal{M}^\star = \varnothing$ **do**
2:    **if** $S =$ INIT **then**
3:      $F \leftarrow \mathcal{A}_{mgr}(\text{ReqVer}(I))$ ▷ run ReqVer (§3.6)
4:      **if** $F = \varnothing$ **then** ▷ check if $I$ is valid
5:        $R \leftarrow \mathcal{A}_p(I)$ ▷ parse user instruction $I$ (§3.3)
6:        $S \leftarrow$ PLAN
7:      **else**
8:        **return** $F$ ▷ return feedback $F$ to the user.
9:      **end if**
10:    **else if** $S =$ PLAN **then**
11:      $\mathbf{P} \leftarrow \mathcal{A}_{mgr}(\text{RAP}(R))$ ▷ run RAP (§3.4)
12:      **for** $\mathbf{p}_i$ in $\mathbf{P}$ **do** ▷ run PD for $\mathcal{A}_d$ and $\mathcal{A}_m$ (§3.5)
13:        $s_i^d \leftarrow \text{PD}(R, \mathcal{A}_d, \mathbf{p}_i)$
14:        $O_i^d \leftarrow \mathcal{A}_d(s_i^d)$ ▷ run pseudo data analysis (§3.5)
15:        $s_i^m \leftarrow \text{PD}(R, \mathcal{A}_m, \mathbf{p}_i, O_i^d)$
16:        $O_i^m \leftarrow \mathcal{A}_m(s_i^m)$ ▷ run search and HPO (§3.5)
17:      **end for**
18:      $\mathbf{O} \leftarrow \{(O_i^d, O_i^m)\}_{i=1}^P$ ▷ combine outcomes (§3.5)
19:      ▷ run execution verification (§3.6)
20:      **if** $\mathcal{A}_{mgr}(\text{ExecVer}(\mathbf{O}))$ **is** pass **then**
21:        $I^\star \leftarrow \mathcal{A}_{mgr}(\mathbf{O})$ ▷ get code instructions
22:        $\mathcal{M}^\star \leftarrow \mathcal{A}_o(I^\star)$ ▷ write code for the best plan
23:        ▷ run implementation verification (§3.6)
24:        **if** $\mathcal{A}_{mgr}(\text{ImpVer}(\mathcal{M}^\star))$ **is** pass **then**
25:          $S \leftarrow$ END ▷ stop the process
26:        **end if**
27:      **end if**
28:    **end if**
29: **end while**
30: **return** $\mathcal{M}^\star$

- **Knowledge**. The knowledge key extracts additional knowledge or insights helpful for solving the given problem directly provided by the user, potentially associated with the expertise level.
- **Service**. The service key is relevant to the downstream implementation and deployment. It provides information such as a target device and an inference engine.

**Prompt Parsing**   Then, we can use the generated dataset $D$ to train an LLM and use it as $\mathcal{A}_p$. These standardized keys are important for a better control over the LLM agents' behavior within our framework and necessary for effective communication between agents. Moreover, these keys provide contextual information for generating a high-quality AutoML pipeline from various perspectives. After the instruction tuning, we use the $\mathcal{A}_p$ to parse the user's instructions (or task descriptions) and return the parsed requirements $R = \mathcal{A}_p(I)$ to $\mathcal{A}_{mgr}$, as shown in Figure 3 and §D.1.

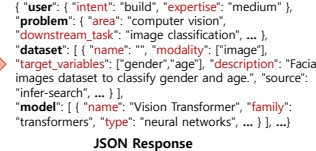

*Figure 3.* An example of prompt parsing process of an instruction-response pair $\{(I_i, R_i)\}$.

### 3.4. Retrieval-Augmented Planning

Recent studies (Guo et al., 2024b; Huang et al., 2024; Masterman et al., 2024; Zhang et al., 2024b; Hu et al., 2024a) highlights that effective planning and tool utilization are essential for solving complex problems with LLMs, especially in a multi-agent framework. By bridging two techniques in a single module, we propose a retrieval-augmented planning (RAP) strategy to effectively devise a robust and up-to-date set of diverse plans for the AuotML problems.

Let $\mathbf{P} = \{\mathbf{p}_1, \ldots, \mathbf{p}_P\}$ be a set of plans. Based on past knowledge embedded in the LLM, knowledge retrieved via external APIs (such as arXiv papers), and $R$, RAP generates $P$ multiple end-to-end plans for the entire AutoML pipeline having different scenario $\mathbf{p}_i$. This strategy enables *AutoML-Agent* to be aware of newer and better solutions. Specifically, we first use the parsed requirements $R$ to acquire a summary of the relevant knowledge and insights via API calls, including web search and paper summary. $\mathcal{A}_{mgr}$ then uses this information to devise $P$ different plans, i.e., $\mathbf{P} = \mathcal{A}_{mgr}(\text{RAP}(R))$. Note that $\mathcal{A}_{mgr}$ devises each plan independently to make the subsequent steps parallelizable. The benefit of this strategy is that it enhances exploration for better solutions while allowing parallelization. Examples of generated plans are provided in §D.2.

### 3.5. Prompting-Based Plan Execution

Given the generated $\mathbf{P}$, we now describe how $\mathcal{A}_d$ and $\mathcal{A}_m$ execute each $\mathbf{p}_i$ using prompting techniques without actually executing the code. Examples are presented in §D.4.

**Plan Decomposition**   Due to the high complexity of the end-to-end plan, we first need to adaptively decompose the original plan $\mathbf{p}_i \in \mathbf{P}$ into a smaller set of sub-tasks $\mathbf{s}_i$ relevant to the agent's roles and expertise to increase the effectiveness of LLMs in solving and executing the given plan (Khot et al., 2023). The plan decomposition (PD) process involves querying the agents about their understanding of the given plan specific to their roles. Formally, $\mathbf{s}_i^d = \text{PD}(R, \mathcal{A}_d, \mathbf{p}_i)$, where $\mathbf{s}_i^d$ is the *decomposed* plan for Data Agent, containing sub-tasks for the given plan $\mathbf{p}_i$. Then, the agent executes the decomposed plan towards the user's requirements instead of the original lengthy plan. We define the sub-tasks $\mathbf{s}_i^m$ of $\mathcal{A}_m$ below due to its reliance on Data Agent's outcomes. Examples are presented in §D.3.

**Pseudo Data Analysis**    In *AutoML-Agent*, $\mathcal{A}_d$ handles sub-tasks in $\mathbf{s}_i^d$, including the retrieval, preprocessing, augmentation, and analysis of the specified dataset. During the data retrieval phase, if the dataset is not directly supplied by the user, we initiate an API call to search for potential datasets in repositories, such as HuggingFace and Kaggle, using the dataset name or description. Upon locating a dataset, we augment the prompt with metadata from the dataset's source; if no dataset is found, we rely on the inherent knowledge of the LLM.[1] We then prompt $\mathcal{A}_d$ to proceed by acting *as if* it actually executes $\mathbf{s}_i^d$, according to the dataset characteristics and user requirements $R$. The summarized outcomes of these sub-tasks, $O_i^d$, are then forwarded to the $\mathcal{A}_m$. Hence, $O_i^d = \mathcal{A}_d(\mathbf{s}_i^d)$.

**Training-Free Model Search and HPO**    Like $\mathcal{A}_d$, $\mathcal{A}_m$ uses API calls to complete all sub-tasks $\mathbf{s}_i^m$, instead of direct code execution. However, in contrast to $\mathcal{A}_d$, the plan decomposition for $\mathcal{A}_m$ incorporates outcomes from the $\mathcal{A}_d$, enabling it to recognize characteristics of the preprocessed dataset, i.e., $\mathbf{s}_i^m = \text{PD}(R, \mathcal{A}_m, \mathbf{p}_i, O_i^d)$. Here, the $\mathcal{A}_m$'s prompt is enhanced with insights gathered by $\mathcal{A}_{mgr}$ about high-performing models and relevant hyperparameters for the specific ML task. This technique allows the $\mathcal{A}_m$ to execute the sub-tasks in $\mathbf{s}_i^m$ more efficiently. Using this augmented prompt, the $\mathcal{A}_m$ follows a similar procedure to $\mathcal{A}_d$, undertaking model retrieval, running HPO, and summarizing the results of these sub-tasks, which include expected numerical performance metrics such as accuracy and error, as well as model complexity factors like model size and inference time. To facilitate the subsequent verification step, we also prompt the agent to return results with the top-$k$ most promising models. Formally, $O_i^m = \mathcal{A}_m(\mathbf{s}_i^m)$.

**Plan Implementation**    To enhance the efficacy of $\mathcal{A}_o$ in code generation, $\mathcal{A}_{mgr}$ first verifies all executed results $\mathbf{O} = \{(O_i^d, O_i^m)\}_{i=1}^P$ from $\mathcal{A}_d$ and $\mathcal{A}_m$. $\mathcal{A}_{mgr}$ then selects the best outcome $O^\star \in \mathbf{O}$ and generates the instruction $I^\star$ for $\mathcal{A}_o$ to write the actual code accordingly. Formally, $\mathcal{M}^\star = \mathcal{A}_o(I^\star)$, where $\mathcal{M}^\star$ is the deployment-ready model.

### 3.6. Multi-Stage Verification

Verification, especially with refinement or feedback, is essential for maintaining the correct trajectory of LLMs (Baek et al., 2025; Madaan et al., 2023; Gou et al., 2024). Our framework incorporates three verification steps to guarantee its accuracy and effectiveness: request verification, execution verification, and implementation verification.

---

[1]When we mention relying on the LLM's inherent knowledge, we are referring to a fallback mechanism intended to produce the most plausible output given limited context. However, in the absence of actual data, the pipeline will ultimately fail during final implementation verification due to runtime errors.

**Request Verification (ReqVer)**    Initially, we assess the clarity of the user's instructions to determine if they are relevant and adequate for executing ML tasks and addressing the user's objectives. If the instructions prove insufficient for progressing to the planning stage, $\mathcal{A}_{mgr}$ will request additional information, facilitating multi-turn communication. This request verification step, however, often overlooked in existing studies, placing an undue burden on users to formulate a more detailed initial prompt—a challenging task particularly for those who are non-experts or lack experience. Prompts for ReqVer are shown in §C.5.1.

**Execution Verification (ExecVer)**    After executing the plans in §3.5, $\mathcal{A}_{mgr}$ then verifies whether any of the pipelines produced by $\mathcal{A}_d$ and $\mathcal{A}_m$ (i.e., $\mathbf{O}$) satisfy the user's requirements via prompting (see §C.5.2). If the results are satisfied, the suggested solution is selected as a candidate for implementation. This step effectively mitigates computational overhead in the search process by allocating resources exclusively to the most promising solution.

**Implementation Verification (ImpVer)**    This implementation verification phase closely resembles the ExecVer; however, it differs in that it involves validating outcomes derived from the code that has been executed and compiled by $\mathcal{A}_o$. We present the prompt for this verification in §C.5.3. If the outcomes meet the user's requirements, $\mathcal{A}_{mgr}$ provides the model and deployment endpoint to the user.

Note that if any execution or implementation fails to satisfy the user requirements (i.e., does not pass the verification process), these failures are systematically documented. Subsequently, the system transitions to the plan *revision* stage. During this stage, $\mathcal{A}_{mgr}$ formulates a revised set of plans, incorporating insights derived from the outcomes of the unsuccessful plans.

## 4. Experiments

We validate the effectiveness of our full-pipeline AutoML framework by comparing *AutoML-Agent* with handcrafted models, state-of-the-art AutoML variants, and LLM-based frameworks across multiple downstream tasks involving different data modalities.

### 4.1. Setup

**Tasks and Datasets**    As summarized in Table 2, we select seven downstream tasks from five different data modalities, including image, text, tabular, graph, and time series. These datasets are chosen from different sources. Also, we incorporate various evaluation metrics for these tasks, e.g., accuracy for classification and root mean squared log error (RMSLE) for regression.

For each task, we prepare *two* sets of natural language task descriptions to represent *constraint-aware* and *constraint-*

Table 2. Summary of downstream tasks and datasets.

| Data Modality | Downstream Task | Dataset Name | Evaluation Metric |
|---|---|---|---|
| Image (Computer Vision) | Image Classification | Butterfly Image Shopee-IET | Accuracy |
| Text (NLP) | Text Classification | Ecommerce Text Textual Entailment | Accuracy |
| Tabular (Classic ML) | Tabular Classification | Banana Quality Software Defects | F1 |
| | Tabular Regression | Crab Age Crop Price | RMSLE |
| | Tabular Clustering | Smoker Status Student Performance | RI |
| Time Series (Time Series Analysis) | Time-Series Forecasting | Weather Electricity | RMSLE |
| Graph (Graph Mining) | Node Classification | Cora Citeseer | Accuracy |

*free* requirements (see §B) along with a full-pipeline skeleton script. As a result, we extensively evaluate **28** generated models. Note that this setting differs from previous studies (Guo et al., 2024a; Huang et al., 2023), which require dataset-specific, partially completed code preparation.

**Evaluation Metrics** For a comprehensive evaluation, we measure the agent's effectiveness in both code generation and task-specific performance aspects by using *comprehensive score* (CS) (Hong et al., 2024a) to simultaneously evaluate both the success rate (SR) of code generation and the normalized performance score (NPS) of the built pipelines. That is, $\text{CS} = 0.5 \times \text{SR} + 0.5 \times \text{NPS}$. Here, $\text{NPS} = \frac{1}{1+s}$ is a transformation of loss-based performance score $s$, e.g., RMSLE. More detailed explanations are included in §B.4.

As described above, we evaluate all frameworks under two different settings. To measure SR of each method, we use a grading scale ranging from 0 for total failure to 1 for perfect conformity to the user's requirements. For the *constraint-free* setting, a method can get a score of 0.5 (pass modeling) or 1.0 (pass deployment). For the *constraint-aware* setting, a method can get a score of 0.25 (pass modeling), 0.5 (pass deployment), 0.75 (partially pass the constraints), or 1.0 (pass all cases).

**Baselines** As we propose a framework for the novel task of full-pipeline AutoML with LLMs, there is no direct baseline available for comparison. We thus compare *AutoML-Agent* against the task-specific manually designed models (see §B.3): **Human Models**, the variants of state-of-the-art AutoML: **AutoGluon** (Erickson et al., 2020; Shchur et al., 2023; Tang et al., 2024), a state-the-of-art LLM for data science: **DS-Agent** (Guo et al., 2024a), and general-purpose LLMs: **GPT-3.5** (Brown et al., 2020) and **GPT-4** (Achiam et al., 2023) with zero-shot prompting.

**Implementation Details** Except for the $\mathcal{A}_p$ that is implemented with Mixtral-8x7B (*Mixtral-8x7B-Instruct-v0.1*) (Jiang et al., 2024), we use GPT-4 (*gpt-4o-2024-05-13*) as the backbone model for all agents and LLM-based baselines to ensure an impartial performance evaluation. To instruction tune the $\mathcal{A}_p$ (§3.3), we automatically gen-

erate about 2.3K instruction-response pairs using EvolInstruct (Xu et al., 2024). Here, we use LoRA (Hu et al., 2021) to fine-tune the model with the generated dataset. For RAP (§3.4), we set the number of plans $P = 3$ and the number of candidate models $k = 3$. All experiments are conducted on an Ubuntu 22.04 LTS server equipped with eight NVIDIA A100 GPUs (CUDA 12.4) and Intel(R) Xeon(R) Platinum 8275CL CPU @ 3.00GHz. For running the generated models, we employ the same execution environment as DS-Agent (Guo et al., 2024a), with all necessary libraries included in the skeleton scripts.

## 4.2. Main Results

We report the average scores from five independent runs for all evaluation metrics in Figure 4.

**Success Rate** Figure 4(left) and Table 6 present the results for the SR metric. For the constraint-free cases, which can be considered easier tasks, all methods have higher SR than ones in the constraint-aware setting. Notably, *AutoML-Agent* also consistently outperforms the baselines in the constraint-aware setting, achieving an average SR of 87.1%, which underscores the effectiveness of our framework. We conjecture that the knowledge retrieved during the planning process helps the agents identify which areas to focus on in order to meet the given constraints. Regarding DS-Agent, although we use the provided example cases for the relevant tasks, it appears to fail on certain tasks due to its heavy reliance on curated case banks and the inclusion of partially completed code, which is unavailable in our setting.

**Downstream Performance** We present the performance comparison between the successfully built models in Figure 4(center) and Table 7. To ensure meaningful results and to examine how the performance of LLM-generated models compares to state-of-the-art AutoML techniques and manual ML pipelines crafted by experienced experts, we select top-performing models by evaluating results reported in Papers with Code benchmarks and Kaggle notebooks for the same tasks and datasets, where applicable, as the Human Models baselines. From the results, we can observe that *AutoML-Agent* significantly outperforms other agents, including Human Models, in the NPS metric. In particular, *AutoML-Agent* achieves the best performance across all tasks under the constraint-aware setting. These findings highlight the superiority of *AutoML-Agent* in adapting to various scenarios, attributed to the retrieval-augmented planning (RAP) strategy. This approach enables agents to discover effective pipelines for given constraints. These empirical observations substantiate the efficacy of the proposed RAP, providing up-to-date solutions for various tasks.

**Comprehensive Score** Figure 4(right) and Table 8 present the weighted quality of each agent based on the CS metric. Overall, *AutoML-Agent* outperforms all other baselines,

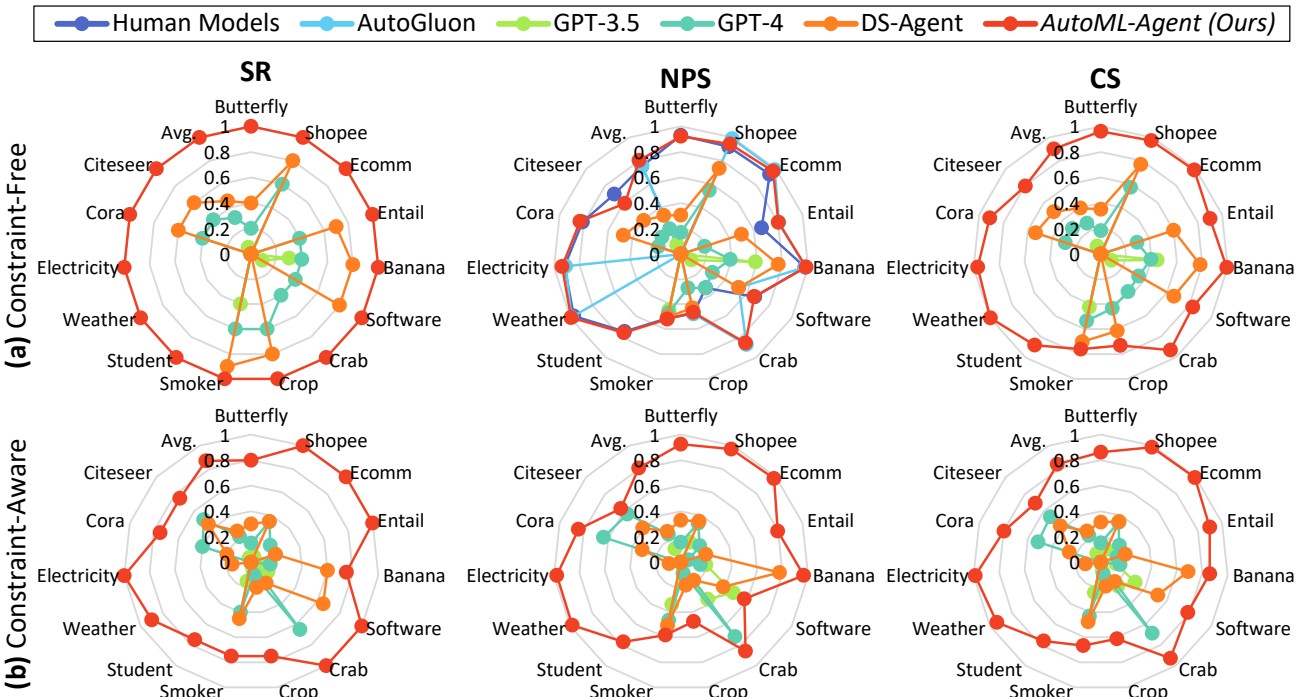

Figure 4. Performance comparison across all datasets using the SR, NPS, and CS metrics under (a) constraint-free and (b) constraint-aware settings. Higher scores indicate better results.

especially in more complicated tasks. Interestingly, it is evident that general-purpose LLMs still works relatively well on classical tasks like tabular classification and regression, while more sophisticated methods, such as DS-Agent and our *AutoML-Agent* work significantly better in complex tasks. This finding aligns with previous research (Guo et al., 2024a), which suggests that tabular tasks typically involve straightforward function calls from the sklearn library (Pedregosa et al., 2011), and therefore do not demand advanced reasoning or coding abilities from LLM agents, unlike more complex tasks.

### 4.3. Additional Analysis

**Ablation Study** To validate the effectiveness of each component in *AutoML-Agent*, we conduct the following ablation studies. The results are presented in Figure 5a and Table 9. First, we investigate *retrieval-augmented planning (RAP)* alone, where retrieved knowledge from external APIs is directly used without plan decomposition and verification. As expected, this ablation leads to a decline in performance, and in some cases, even fails to generate a runnable model. This outcome highlights the importance of the decomposition and verification modules. Second, we evaluate *RAP with plan decomposition*, where the plan is decomposed for each specific agent. While this variant demonstrates better downstream performance, it still fails to produce runnable models in certain downstream tasks due to the lack of code verification. Finally, we assess the *full framework with multi-stage*

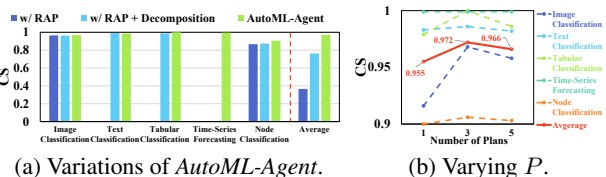

(a) Variations of *AutoML-Agent*.    (b) Varying $P$.

Figure 5. Results of (a) ablation study and (b) hyperparameter study in the **CS** metric.

*verification*, which provides feedback to agents, thereby enhancing both their planning and coding capabilities. Integrating all components significantly empowers LLM agents to effectively incorporate external knowledge from various sources to build a full-pipeline AutoML system.

**Hyperparameter Study** To further verify the effectiveness of devising multiple plans in our retrieval-augmented planning strategy (§3.4), we conduct a hyperparameter study by varying the number of plans $P$ in the constraint-free setting. As shown in Figure 5b and Table 10, the number of plans does not significantly affect the success rate, likely due to GPT-4's robust planning capabilities. However, based on the NPS and CS metrics, we observe that the number of plans has a notable impact on downstream task performance. Also, these results also suggest that adding more plans does not necessarily lead to better results, as the model may generate multiple similar plans, resulting in similar outcomes. Consequently, we select 3 as the default number of plans.

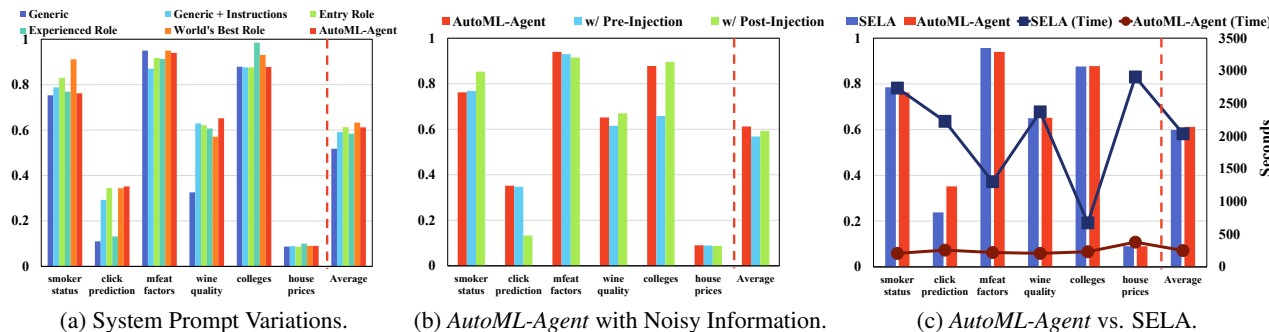

(a) System Prompt Variations.      (b) *AutoML-Agent* with Noisy Information.      (c) *AutoML-Agent* vs. SELA.

*Figure 6.* Results of (a) sensitivity analysis, (b) robustness analysis, and (c) comparison with SELA (Chi et al., 2024) in the **CS** metric.

**Prompt Sensitivity** To understand how different levels of system prompt affect the framework performance, we test five prompt variations for each agent, differing in tone and task specificity (see §C.2). Overall, the experimental results in Figure 6a show that agents are not highly sensitive to exact phrasing *as long as their roles were clearly defined*, which is also reinforced through the user prompts—making the framework robust to variations in system prompts.

**Noise Robustness** Since the framework can encounter noisy information from external knowledge sources or APIs during the RAP process, we further evaluate the robustness of *AutoML-Agent* to such noise under two simulated scenarios. First, in the pre-summary injection scenario, unrelated or low-quality examples are injected *before* insight extraction and planning. Second, in the post-summary injection scenario, noisy examples are injected *after* insight extraction but before planning. That is, they are mixed with useful insights. Here, the noisy inputs are generated by an additional adversarial agent prompted to produce "unhelpful" and "fake" insights intended to disrupt the planning process.

The results in Figure 6b demonstrate that *AutoML-Agent* is robust to such noise. Its built-in error correction and multi-stage verification mechanisms significantly mitigate the impact of noisy inputs, ensuring that final model performance *remains largely unaffected*. Interestingly, we observe that in some cases, noise injection can even improve performance. We conjecture that this may be because the Agent Manager is implicitly prompted to more effectively distinguish between useful and irrelevant information.

**Comparison with Training-Based Search** To examine the effectiveness of our RAP strategy (§3.4) in acquiring useful knowledge for planning and prompting-based plan execution (§3.5) in enhancing search speed, we compare our *AutoML-Agent* with the recent Monte Carlo tree search-based SELA method (10 rollouts) using six datasets. These datasets represent the easiest and hardest tasks, as identified by SELA's results. Figure 6c and Table 12 show that *AutoML-Agent* achieves search times about **8x faster** than SELA, highlighting significant computational efficiency in line with our focus on practical applicability. Despite this ef-

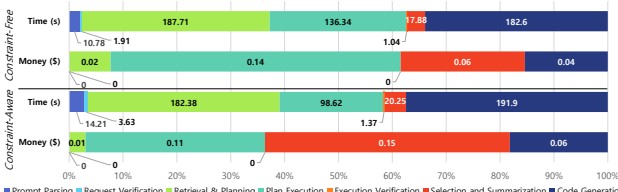

*Figure 7.* Average time and monetary cost breakdown.

ficiency, our method achieves comparable or superior performance, with an average score of 0.612 compared to SELA's 0.599. While SELA slightly outperforms *AutoML-Agent* on a few datasets, the gains are marginal and come at a substantial computational cost. This trade-off between efficiency and minor performance gains further justifies our decision to avoid computationally expensive training-based search.

**Resource Cost** As we primarily use closed-source LLMs in this paper, we analyze the costs in terms of time and money. Figure 7 presents the average time and monetary costs across different tasks and datasets for a single run, under the constraint-free (upper) and constraint-aware (lower) settings. On average, it takes around 525 seconds and costs 0.30 USD (using GPT-4o) to search for a single model that will be deployable after training. The significant amount of time spent in the planning stage also suggests the difficulty in devising plans for full-pipeline AutoML.

## 5. Conclusions

This paper presents *AutoML-Agent*, a novel LLM-based multi-agent framework designed for AutoML, covering the entire pipeline from data retrieval to model deployment. *AutoML-Agent* tackles the full-pipeline planning complexity and implementation accuracy challenges in the LLMs for task-agnostic AutoML by leveraging the newly proposed retrieval-augmented planning strategy and multi-stage verification. In addition, we enhance the plan execution efficiency by integrating role-specific decomposition and prompting-based execution techniques into the framework. Our experiments on seven ML tasks demonstrate that *AutoML-Agent* outperforms existing methods in terms of success rate and downstream task performance.

## Acknowledgements

We thank the anonymous reviewers for their insightful comments and suggestions, which helped improve the quality of this paper. This work was supported by Institute of Information & communications Technology Planning & Evaluation (IITP) grant funded by the Korea government(MSIT) (No.2019-0-00075, Artificial Intelligence Graduate School Program(KAIST)), IITP grant funded by MSIT (No.RS-2022-II220713, Meta-learning Applicable to Real-world Problems), the National Research Foundation of Korea(NRF) grant funded by MSIT (No. RS-2023-00256259).

## Impact Statement

We expect *AutoML-Agent* to offer significant advantages by promoting AI-driven innovation and enabling individuals with limited AI expertise to effectively utilize AI capabilities. However, we acknowledge the potential misuse of *AutoML-Agent* by malicious users, such as generating offensive content, malicious software, or invasive surveillance tools when exposed to harmful inputs. This vulnerability is not unique to our approach but represents a common challenge faced by existing LLMs with substantial creative and reasoning capabilities, which can occasionally produce undesirable outputs. Although we strictly instruct the LLM to focus on generating positive results for machine learning tasks, there is a possibility of unforeseen glitches that could introduce security issues within the system. Therefore, we recommend running *AutoML-Agent* within a Docker container to ensure isolation from the host's file system. Additionally, due to its integration with external services for retrieval-augmented generation and API-based LLMs like GPT-4, privacy concerns may arise. Users should carefully review any data included in API prompts to prevent unintended data disclosures.

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

# Appendix

The appendix of this paper is organized as follows.

## A. Additional Discussions

This section discusses the motivation behind *AutoML-Agent* in relation to non-expert users, the challenges associated with smaller models, comparisons with general LLM-Agent frameworks, and the limitations of our approach.

### A.1. Motivation and Necessity

In the era of generative AI, many companies are adopting AI technologies. However, these companies often lack AI experts, leading to software engineers—*who are non-experts in AI*—attempting to implement such solutions and facing significant challenges. Researchers from various domains outside of AI (e.g., economics, chemistry, and healthcare) are increasingly seeking to apply AI models in their work but struggle due to their lack of specialized AI expertise. Besides, the existence of non-expert users in the ML landscape is well-documented in relevant studies (Karmaker et al., 2021; Sun et al., 2023; Tornede et al., 2024; Davenport & Bean, 2024). For example, an early AutoML survey paper (Karmaker et al., 2021) highlights the growing demand for ML tools among stakeholders across various domains. It emphasizes that AutoML tools aim to make ML accessible to non-experts, improve efficiency, and accelerate research, addressing the pressing need for user-friendly AI solutions. Similarly, as thoroughly illustrated in Section 3 of a recent study (Tornede et al., 2024), the use of LLMs as interfaces for AutoML and as components of AutoML systems offers significant opportunities for non-ML experts seeking to apply off-the-shelf data-driven solutions to their problems.

To further illustrate the necessity and practicality of *AutoML-Agent*, consider an academic researcher aiming to evaluate several ML models for a novel dataset within a constrained timeline. Traditionally, this involves significant manual effort in setting up pipelines, selecting models, and debugging code—steps that can be both error-prone and time-consuming. Similarly, in industry settings, ML engineers often need to rapidly prototype models for specific business requirements, such as creating a lightweight and efficient spam detection system for mobile applications. *AutoML-Agent* enables such users to focus on high-level problem formulation by generating deployment-ready models that adhere to specified constraints, like latency or accuracy, directly from natural language task descriptions. This capability reduces setup time and errors, enabling users to focus on innovation rather than implementation logistics.

### A.2. Challenges with Smaller Models

While we acknowledge the importance of validating our framework on smaller-scale models, these models exhibit systemic limitations that hinder their utility for complex tasks requiring code execution and extensive planning. During the early stage of development, we found that small language models, including LLaMA-2-7B, Qwen1.5-14B, gemma-7B-it, WizardLM-70B (Xu et al., 2024), and even code-specialized like CodeQwen1.5-7B failed to generate executable code for tasks requiring extensive planning or interdependent processes (e.g., full-pipeline skeleton script presented in §B.1). Commonly, these smaller models exhibit similar issues, such as cutting the given codes, changing comments without code completion, partial completion, and returning code as it is given. As shown in §F below, even vanilla GPT-3.5 and GPT-4o struggle with generating executable code for complex tasks, involving extensive planning or interdependent processes. Similar findings have been corroborated by prior studies (Guo et al., 2024a; Hong et al., 2024a), suggesting that these challenges are systemic to smaller language models and not unique to our framework.

### A.3. Key Distinctions from General LLM-Agent Frameworks

While it is true that generic LLM-agents are versatile and can theoretically execute certain AutoML tasks, they lack critical domain-specific capabilities essential for achieving robust and reliable performance across the structured and interdependent

processes of a full AutoML pipeline. Generic LLM frameworks are typically designed to support broad problem domains and rely heavily on user-defined instructions or augmentations, often leading to inefficiencies, suboptimal performance, and increased failure rates in complex tasks like end-to-end AutoML. In contrast, *AutoML-Agent* is purpose-built with explicit mechanisms, such as retrieval-augmented planning, specialized agents for sub-tasks, and multi-stage verification, all tailored to address the unique challenges of AutoML. Through these enhancements, we can increase plan execution efficiency and support diverse ML tasks with more accurate pipeline implementation.

Furthermore, as highlighted in §F, even when a general-purpose LLM leverages the same backbone model, they consistently underperform compared to specialized methods like DS-Agent or *AutoML-Agent* in both success rates and downstream task metrics. *AutoML-Agent* not only outperforms these frameworks but also uniquely integrates the entire pipeline into a seamless, modular workflow, significantly reducing errors and enhancing usability for diverse downstream tasks. This distinction underscores the importance of designing specialized frameworks like *AutoML-Agent*, which prioritize task-specific optimizations over generality, ensuring both efficiency and reliability in automating full AutoML pipelines.

### A.4. Limitations and Future Work

Even though we offer a flexible module capable of accommodating various machine learning tasks and data modalities, the absence of skeleton code for entirely new tasks may increase the risk of code hallucination. Moreover, in the current version, there remains a noticeable gap in code generation quality when using different backbone models, e.g., GPT-4 vs. GPT-3.5. This discrepancy is not unique to our approach but represents a broader challenge faced by existing LLM-based frameworks. Future work focused on developing a more robust framework that can generate reasonable solutions with reduced dependence on a specific LLM backbone is a promising direction.

Additionally, to further improve the literature retrieval component during the planning process, future iterations could incorporate tools, such as PaperQA (Lála et al., 2023). A PaperQA-style scientific querying system could enhance citation relevance and enable more context-aware prompt generation, ultimately leading to improved planning decisions.

Our work also encounters limitations in code generation when applied to machine learning tasks requiring significantly different development pipelines from those evaluated in our experiments, which primarily focused on general supervised and unsupervised settings. Tasks such as reinforcement learning and recommendation systems introduce particular challenges. Consequently, extending *AutoML-Agent* to these domains will necessitate the development of additional agents to handle specialized steps in the target pipeline, such as actor-environment interaction and reward modeling in the case of reinforcement learning.

In particular, extending *AutoML-Agent* to reinforcement learning would require incorporating domain-specific modules—for instance, agents for environment interaction, action space design, and reward processing.[2] One could envision an "RL agent"" interfacing with an environment (e.g., OpenAI Gym), while the planner generates RL-specific procedures such as policy training and evaluation. Although this extension is certainly feasible, it could be substantial enough that an RL-specific version of *AutoML-Agent*, perhaps termed AutoRL-Agent, would represent a significant contribution worthy of a separate publication or project.

## B. Details of Experimental Setup

This section outlines the detailed experimental setup used in this paper, including the complete instruction prompts for both constraint-free (Table 3) and constraint-aware (Table 4) settings, a full-pipeline skeleton script (§B.1), dataset (§B.2) and baseline descriptions (§B.3), as well as evaluation metrics (§B.4).

---

[2]See `https://www.automl.org/autorl-survey`.

*Table 3.* User instruction (i.e., task description) for experiments under the *constraint-free* setting.

| Task | Dataset | Instruction Prompt |
|---|---|---|
| Image Classification | Butterfly Image | I need a very accurate model to classify images in the Butterfly Image Classification dataset into their respective categories. The dataset has been uploaded with its label information in the labels.csv file. |
| | Shopee-IET | Please provide a classification model that categorizes images into one of four clothing categories. The image path, along with its label information, can be found in the files train_labels.csv and test_labels.csv. |
| Text Classification | Ecommerce Text | We need a state-of-the-art model for text classification based on the Ecommerce Text dataset. The model should be capable of accurately classifying text into four categories: Electronics, Household, Books, and Clothing & Accessories. We have uploaded the entire dataset without splitting it here. |
| | Textual Entailment | You are solving this machine learning tasks of classification: The dataset presented here (the Textual Entailment) comprises a series of labeled text pairs. Given two texts (text1 and text2), your task is to predict the relationship of the text pair of neutral (0), contradiction (1) or entailment (2). The evaluation metric is accuracy. Build a language model to get a good performance. |
| Tabular Classification | Banana Quality | Build a model to classify banana quality as Good or Bad based on their numerical information about bananas of different quality (size, weight, sweetness, softness, harvest time, ripeness, and acidity). We have uploaded the entire dataset for you here in the banana_quality.csv file. |
| | Software Defects | You are solving this data science tasks of binary classification: The dataset presented here (the Software Defects Dataset) comprises a lot of numerical features. Please split the dataset into three parts of train, valid and test. Your task is to predict the defects item, which is a binary label with 0 and 1. The evaluation metric is the F1 score. Please train a binary classification model to get a good performance on this task. |
| Tabular Regression | Crab Age | You are solving this data science tasks of regression: The dataset presented here (the Crab Age Dataset) comprises a lot of both categorical and numerical features. Pleae split the dataset into three parts of train, valid and test. Your task is to predict the age item. The evaluation metric is the RMSLE (root mean squared log error). Now train a regression model to get a good performance on this task. |
| | Crop Price | I need a regression model to predict crop prices based on features like soil composition, environmental factors, historical yield data, and crop management practices from the dataset I uploaded here. |
| Tabular Clustering | Smoker Status | You are solving this data science tasks of unsupervised clustering: The dataset presented here (the Smoker Status Dataset) comprises a lot of numerical features. Please use the features in the test.csv file. Your task is to create the clustered items, which is a binary label with 0 and 1 (two clusters). The evaluation metric is the Rand index or Rand score, can be tested against 'smoking' labels. Now train an unsupervised clustering model to get a good performance on this task. |
| | Higher Education Students Performance | I want an unsupervised clustering model to group student performances into eight groups. The dataset named 'Higher Education Students Performance Evaluation' (id=856) can be downloaded via ucimlrepo library. The clustering quality can be check against target variable OUTPUT Grade. |
| Time-Series Forecasting | Weather | I want you to create a model for node classification on the Cora dataset to predict the category of each paper. You need to directly find the Cora dataset from a relevant library. |
| | Electricity | I want you to create a model for node classification on the Citeseer dataset to predict the category of each paper. You need to directly find the Citeseer dataset from a relevant library. |
| Node Classification | Cora | Build a model to perform time-series forecasting using the Weather dataset uploaded here, evaluating its accuracy with the RMSLE metric. Note that the input is a sequence of past observations with fixed size (INPUT_SEQ_LEN=96, INPUT_DIM=21). The model should predict the next future sequence with a fixed size (PRED_SEQ_LEN=96, PRED_DIM=21). |
| | Citeseer | You are solving this machine learning tasks of time series forecasting: The dataset presented here (the Electricity dataset) comprises real-world time series data. Please split the dataset into three parts of train, valid and test. The input is a sequence of past observation with fixed size (INPUT_SEQ_LEN=96, INPUT_DIM=321). Your task is to predict the next future sequence with fixed size (PRED_SEQ_LEN=96, PRED_DIM=321). The evaluation metric is root mean squared log error (RMSLE). Now train a time series forecasting model to get a good performance on the given fixed sequences. |

*Table 4.* User instruction (i.e., task description) for experiments under the *constraint-aware* setting. **Bold** texts indicate constraints used for evaluation.

| Task | Dataset | Instruction Prompt |
|---|---|---|
| Image Classification | Butterfly Image | I need a highly accurate machine learning model developed to classify images within the Butterfly Image Classification dataset into their correct species categories. The dataset has been uploaded with its label information in the labels.csv file. Please use a convolutional neural network (CNN) architecture for this task, leveraging transfer learning from a **pre-trained ResNet-50 model** to improve accuracy. Optimize the model using cross-validation on the training split to fine-tune hyperparameters, and aim for an **accuracy of at least 0.95 on the test split**. Provide the final trained model, a detailed report of the training process, hyperparameter settings, accuracy metrics, and a confusion matrix to evaluate performance across different categories. |
| | Shopee-IET | Please provide a classification model that categorizes images into one of four clothing categories. The image path, along with its label information, can be found in the files train_labels.csv and test_labels.csv. The model should **achieve at least 85% accuracy on the test set** and be implemented using PyTorch. Additionally, please include data augmentation techniques and a confusion matrix in the evaluation. |
| Text Classification | Ecommerce Text | We require the development of an advanced **neural network model** for text classification tailored to the Ecommerce Text dataset, with the objective of **achieving at least 0.95 classification accuracy**. The model should be specifically trained to distinguish text into four defined categories: Electronics, Household, Books, and Clothing & Accessories. To facilitate this, we have uploaded the complete dataset in its entirety, without any prior division into training, validation, or test sets. |
| | Textual Entailment | You are solving this machine learning task of classification: The dataset presented here (the Textual Entailment) comprises a series of labeled text pairs. Given two texts, your task is to predict the relationship of the text pair as neutral (0), contradiction (1), or entailment (2). The evaluation metric is accuracy. Build a language model to get good performance, ensuring the **model size does not exceed 200 million parameters** and the **inference time is less than 200 milliseconds per prediction**. |
| Tabular Classification | Banana Quality | Build a machine learning model, potentially **XGBoost or LightGBM**, to classify banana quality as Good or Bad based on their numerical information about bananas of different quality (size, weight, sweetness, softness, harvest time, ripeness, and acidity). We have uploaded the entire dataset for you here in the banana_quality.csv file. The model must **achieve at least 0.98 accuracy**. |
| | Software Defects | You are solving this data science task of binary classification: The dataset presented here (the Software Defects Dataset) comprises a lot of numerical features. Please split the dataset into three parts of train, valid, and test. Your task is to predict the defects item, which is a binary label with 0 and 1. The evaluation metric is the F1 score. Please train a binary classification model to get a good performance on this task, ensuring that the model **training time does not exceed 30 minutes** and the **prediction time for each instance is under 5 milliseconds**. |
| Tabular Regression | Crab Age | You are solving this data science task of regression: The dataset presented here (the Crab Age Dataset) comprises a lot of both categorical and numerical features. Please split the dataset into three parts of train, valid, and test. Your task is to predict the age item. The evaluation metric is the RMSLE (root mean squared log error). Now train a regression model to get a good performance on this task, ensuring that the model's **training time does not exceed 30 minutes** and that it can make **predictions on the test set within 5 seconds**. |
| | Crop Price | I need an accurate regression model to predict crop prices based on features like soil composition, environmental factors, historical yield data, and crop management practices from the dataset I uploaded here. You should optimize the model to achieve **RMSLE less than 1.0** |
| Tabular Clustering | Smoker Status | You are solving this data science task of unsupervised clustering: The dataset presented here (the Smoker Status Dataset) comprises a lot of numerical features. Please use the features in test.csv. Your task is to create the clustered items, which is a binary label with 0 and 1 (two clusters). The evaluation metric is the Rand index or Rand score, which can be tested against 'smoking' labels. Now train an unsupervised clustering model to get a good performance on this task, ensuring that the **Rand index is at least 0.75** and the model **training time does not exceed 10 minutes**. |
| | Higher Education Students Performance | I want an unsupervised clustering model to group student performances into eight groups. The dataset named 'Higher Education Students Performance Evaluation' (id=856) can be downloaded via ucimlrepo library. The clustering quality can be checked against the target variable OUTPUT Grade. The model should achieve a **Rand Score of at least 0.8** and **complete clustering within 10 minutes**. |
| Time-Series Forecasting | Weather | Build a state-of-the-art time-series forecasting model for the Weather dataset uploaded here, evaluating its accuracy with the RMSLE metric. Note that the input is a sequence of past observations with fixed size (INPUT_SEQ_LEN=96, INPUT_DIM=21). The model should predict the next future sequence with a fixed size (PRED_SEQ_LEN=96, PRED_DIM=21). We target **RMSLE lower than 0.05**. |
| | Electricity | You are solving this machine learning task of time series forecasting: The dataset presented here (the Electricity dataset) comprises real-world time series data. Please split the dataset into three parts of train, valid, and test. The input is a sequence of past observation with fixed size (INPUT_SEQ_LEN=96, INPUT_DIM=321). Your task is to predict the next future sequence with fixed size (PRED_SEQ_LEN=96, PRED_DIM=321). The evaluation metric is root mean squared log error (RMSLE). Now train a time series forecasting model to get a good performance on the given fixed sequences. Ensure the model achieves an **RMSLE of less than 0.1** and that the **training time does not exceed 1 hour** on a GPU. |
| Node Classification | Cora | I want you to develop a node classification model using the **Graph Convolutional Network (GCN)** algorithm to predict the category of each paper in the Cora dataset. Start by importing the Cora dataset using the 'Planetoid' dataset from the 'torch_geometric.datasets' module in PyTorch Geometric. Ensure you preprocess the data to include node features and labels correctly. Train the model using a suitable optimizer and loss function. Then, evaluate its accuracy on the test set. **The accuracy on the test set should be over 0.90**. |
| | Citeseer | I want you to develop a node classification model using the **Graph Convolutional Network (GCN)** algorithm to predict the category of each paper in the Citeseer dataset. Start by importing the Citeseer dataset using the 'Planetoid' dataset from the 'torch_geometric.datasets' module in PyTorch Geometric. Ensure you preprocess the data to include node features and labels correctly. Train the model using a suitable optimizer and loss function. Then, evaluate its accuracy on the test set. The **accuracy on the test set should be over 0.80**. |

## B.1. Skeleton Python Script

**Skeleton Python Script (e.g., text_classification.py)**

```python
# The following code is for "text classification" task using PyTorch.
import os, random, time, json

# define GPU location
os.environ["CUDA_DEVICE_ORDER"] = "PCI_BUS_ID"
os.environ["CUDA_VISIBLE_DEVICES"] = "3"

import torch
import torch.nn as nn
import torch.optim as optim
import numpy as np
import gradio as gr

# TODO: import other required library here, including libraries for datasets and (pre-trained) models like
#     HuggingFace and Kaggle APIs. If the required module is not found, you can directly install it by running `
```

```python
     pip install your_module'.
from torchtext import datasets, data, vocab
from torch.utils.data import DataLoader, Dataset
from sklearn.metrics import accuracy_score, f1_score

SEED = 42
random.seed(SEED)
torch.manual_seed(SEED)
np.random.seed(SEED)

# Define device for model operations
device = torch.device("cuda" if torch.cuda.is_available() else "cpu")

DATASET_PATH = "_experiments/datasets" # path for saving and loading dataset(s) (or the user's uploaded dataset)
      for preprocessing, training, hyperparamter tuning, deployment, and evaluation

# Data preprocessing and feature engineering
def preprocess_data():
    # TODO: this function is for data preprocessing and feature engineering

    # Run data preprocessing

    # Should return the preprocessed data
    return processed_data

def train_model(model, train_loader):
    # TODO: this function is for model training loop and optimization on 'train' and 'valid' datasets
    # TODO: this function is for fine-tuning a given pretrained model (if applicable)

    # Should return the well-trained or finetuned model.
    return model

def evaluate_model(model, test_loader):
    # In this task, we use Accuracy and F1 metrics to evaluate the text classification performance.
    # The 'performance_scores' should be in dictionary format having metric names as the dictionary keys
    # TODO: the first part of this function is for evaluating a trained or fine-tuned model on the 'test' dataset
    #       with respect to the relevant downstream task's performance metrics
    # Define the 'y_true' for ground truth and 'y_pred' for the predicted classes here.

    performance_scores = {
        'ACC': accuracy_score(y_true, y_pred),
        'F1': f1_score(y_true, y_pred)
    }

    # TODO: the second part of this function is for measuring a trained model complexity on a samples with
    #       respect to the relevant complexity metrics, such as inference time and model size
    # The 'complexity_scores' should be in dictionary format having metric names as the dictionary keys

    # Should return model's performance scores
    return performance_scores, complexity_scores

def prepare_model_for_deployment():
    # TODO: this function is for preparing an evaluated model using model compression and conversion to deploy
    #       the model on a particular platform

    # Should return the deployment-ready model
    return deployable_model

def deploy_model():
    # TODO: this function is for deploying an evaluated model with the Gradio Python library

    # Should return the url endpoint generated by the Gradio library
    return url_endpoint

# The main function to orchestrate the data loading, data preprocessing, feature engineering, model training,
#     model preparation, model deployment, and model evaluation
def main():
    """
    Main function to execute the text classification pipeline.
    """

    # TODO: Step 1. Retrieve or load a dataset from hub (if available) or user's local storage (if given)
    dataset = None

    # TODO: Step 2. Create a train-valid-test split of the data by splitting the 'dataset' into train_loader,
    #     valid_loader, and test_loader.
    # Here, the train_loader contains 70% of the 'dataset', the valid_loader contains 20% of the 'dataset', and
    #     the test_loader contains 10% of the 'dataset'.
```

```
    train_loader, valid_loader, test_loader = (None, None, None) # corresponding to 70%, 20%, 10% of 'dataset'

    # TODO: Step 3. With the split dataset, run data preprocessing and feature engineering (if applicable) using
        the "preprocess_data" function you defined
    processed_data = preprocess_data()

    # TODO: Step 4. Define required model. You may retrieve model from available hub or library along with
        pretrained weights (if any).
    # If pretrained or predefined model is not available, please create the model according to the given user's
        requirements below using PyTorch and relevant libraries.
    model = None

    # TODO: Step 5. train the retrieved/loaded model using the defined "train_model" function
    # TODO: on top of the model training, please run hyperparamter optimization based on the suggested
        hyperparamters and their values before proceeding to the evaluation step to ensure model's optimality

    model = train_model()

    # TODO: evaluate the trained model using the defined "evaluate_model" function
    model_performance, model_complexity = evaluate_model()

    # TODO: compress and convert the trained model according to a given deployment platform using the defined "
        prepare_model_for_deployment" function
    deployable_model = prepare_model_for_deployment()

    # TODO: deploy the model using the defined "deploy_model" function
    url_endpoint = deploy_model()

    return processed_data, model, deployable_model, url_endpoint, model_performance, model_complexity

if __name__ == "__main__":
    processed_data, model, deployable_model, url_endpoint, model_performance, model_complexity = main()
    print("Model Performance on Test Set:", model_performance)
    print("Model Complexity:", model_complexity)
```

## B.2. Dataset Descriptions

As presented in Table 5, we select seven representative downstream tasks, covering five data modalities. We describe the datasets their statistics as follows.

- **Butterfly Image (Butterfly).** This dataset includes 75 distinct classes of butterflies, featuring over 1,000 labeled images, including validation images. Each image is assigned to a single butterfly category. The dataset is accessible at https://www.kaggle.com/datasets/phucthaiv02/butterfly-image-classification.
- **Shopee-IET (Shopee).** This dataset is designed for cloth image classification, where each image represents a clothing item, and its corresponding label indicates the clothing category. The available labels include BabyPants, BabyShirt, womencasualshoes, and womenchiffontop. The dataset is available at https://www.kaggle.com/competitions/demo-shopee-iet-competition/data.
- **Ecommerce Text (Ecomm).** This dataset is a classification-based E-commerce text dataset comprising four categories: Electronics, Household, Books, and Clothing & Accessories, which together cover approximately 80% of any E-commerce website. It includes 50,425 instances and can be found at https://www.kaggle.com/datasets/saurabhshahane/ecommerce-text-classification.
- **Textual Entailment (Entail).** This dataset consists of labeled pairs of text, where the task is to predict the relationship between each pair as either neutral (0), contradiction (1), or entailment (2). It is divided into a training set containing 4,907 samples and a testing set with 4,908 samples. We use the dataset provided by Guo et al. (2024a).
- **Banana Quality (Banana).** This tabular dataset consists of numerical information on 8,000 samples of bananas, covering various quality attributes such as size, weight, sweetness, softness, harvest time, ripeness, acidity, and overall quality. The primary objective of the dataset is to classify each banana sample as either good or bad. The dataset is available at https://www.kaggle.com/datasets/l3llff/banana/data.
- **Software Defects (Software).** This dataset consists primarily of numerical features and has been divided into three parts: training, validation, and testing. The goal is to predict a binary defect label (0 or 1). The training set contains 82,428 samples, the validation set contains 9,158 samples, and the test set contains 91,587 samples. We use the dataset provided by Guo et al. (2024a).
- **Crab Age (Crab).** This dataset contains a mix of categorical and numerical features, and has been divided into three parts: training, validation, and test sets. The task is to predict the age of the crabs. The training set consists of 59,981

*Table 5.* Summary of downstream tasks and benchmark dataset statistics.

| Data Modality | Downstream Task | Dataset Name | # Features | # Train | # Valid | # Test | # Classes | Source | License | Evaluation Metric |
|---|---|---|---|---|---|---|---|---|---|---|
| *Main Datasets* | | | | | | | | | | |
| Image (Computer Vision) | Image Classification | Butterfly Image | 224x224 | 4,549 | 1,299 | 651 | 75 | Kaggle Dataset | CC0 | Accuracy |
| | | Shopee-IET | Varying | 640 | 160 | 80 | 4 | Kaggle Competition | Custom | |
| Text (Natural Language Processing) | Text Classification | Ecommerce Text | N/A | 35,296 | 10,084 | 5,044 | 4 | Kaggle Dataset | CC BY 4.0 | Accuracy |
| | | Textual Entailment | N/A | 3,925 | 982 | 4,908 | 3 | Kaggle Dataset | N/A | |
| Tabular (Classic Machine Learning) | Tabular Classification | Banana Quality | 7 | 5,600 | 1,600 | 800 | 2 | Kaggle Dataset | Apache 2.0 | F1 |
| | | Software Defects | 21 | 73,268 | 18,318 | 91,587 | 2 | Kaggle Competition | N/A | |
| | Tabular Clustering | Smoker Status | 22 | 100,331 | 28,666 | 14,334 | 2 | Kaggle Competition | N/A | RI |
| | | Higher Education Students Performance | 31 | 101 | 29 | 15 | 8 | Research Dataset (UCI ML) | CC BY 4.0 | RI |
| | Tabular Regression | Crab Age | 8 | 53,316 | 13,329 | 66,646 | N/A | Kaggle Competition | CC0 | RMSLE |
| | | Crop Price | 8 | 1,540 | 440 | 220 | N/A | Kaggle Dataset | MIT | RMSLE |
| Graph (Graph Learning) | Node Classification | Cora | 1,433 | 2,708 | 2,708 | 2,708 | 7 | Research Dataset (Planetoid) | CC BY 4.0 | Accuracy |
| | | Citeseer | 3,703 | 3,327 | 3,327 | 3,327 | 6 | Research Dataset (Planetoid) | N/A | |
| Time Series (Time Series Analysis) | Time-Series Forecasting | Weather | 21 | 36,887 | 10,539 | 5,270 | N/A | Research Dataset (TSLib) | CC BY 4.0 | RMSLE |
| | | Electricity | 321 | 18,412 | 5,260 | 2,632 | N/A | Research Dataset (TSLib) | CC BY 4.0 | |
| *Additional Datasets from SELA* | | | | | | | | | | |
| Tabular (Classic Machine Learning) | Binary Classification | Smoker Status | 22 | 85997 | 21500 | 143331 | 2 | Kaggle Competition | | F1 |
| | | Click Prediction Small | 11 | 19174 | 4794 | 7990 | 2 | OpenML | | |
| | Multi-Class Classification | MFeat Factors | 216 | 960 | 240 | 400 | 10 | OpenML | N/A | |
| | | Wine Quality White | 11 | 2350 | 588 | 980 | 7 | OpenML | | |
| | Regression | Colleges | 44 | 3389 | 848 | 1413 | N/A | OpenML | | RMSE |
| | | House Prices | 80 | 700 | 176 | 292 | N/A | Kaggle Competition | | |

samples, the validation set includes 6,664 samples, and the test set contains 66,646 samples. We use the dataset provided by Guo et al. (2024a).

- **Crop Price (Crop).** This new dataset contains 2,200 samples with key features such as nitrogen, phosphorus, and potassium ratios in the soil, temperature (in °C), humidity (in %), soil pH value, and rainfall (in mm), all of which are essential for predicting crop yield values. Crop yield prediction is crucial in modern agriculture, particularly as data-driven methods become more prevalent. This dataset is available at `https://www.kaggle.com/datasets/varshitanalluri/crop-price-prediction-dataset`.

- **Smoker Status (Smoker).** This dataset contains numerous numerical features. The goal is to categorize smoking status of each instance into a cluster. The training set consists of 143,330 samples and the test set includes 143,331 samples. We use the dataset provided by Guo et al. (2024a).

- **Higher Education Students Performance (Student).** The dataset, collected in 2019 from students in the Faculty of Engineering and Faculty of Educational Sciences, was created to predict students' end-of-term performances using machine learning techniques. It is a multivariate dataset with 145 instances and 31 integer-type features, focusing on classification tasks within the domain of social sciences. We adopt this dataset for unsupervised clustering instead of classification. This dataset can be found at `https://archive.ics.uci.edu/dataset/856/higher+education+students+performance+evaluation`.

- **Weather.** The weather dataset consists of 21 meteorological factors collected every 10 minutes from the Weather Station at the Max Planck Biogeochemistry Institute in 2020, containing 52,603 samples without any pre-splitting. It is accessible at `https://github.com/thuml/Time-Series-Library`.

- **Electricity.** This dataset comprises hourly electricity consumption data for 321 customers collected from 2012 to 2014, totaling 26,211 samples. The dataset records the electricity usage of these clients on an hourly basis and is provided without any pre-split. The dataset is available at `https://github.com/thuml/Time-Series-Library`.

- **Cora** and **Citeseer.** The citation network datasets, "Cora" and "CiteSeer," consist of nodes representing documents and edges representing citation links between them. Both datasets provide training, validation, and test splits through binary masks. The Cora dataset contains 2,708 nodes, 10,556 edges, 1,433 features, and 7 classes, while CiteSeer consists of 3,327 nodes, 9,104 edges, 3,703 features, and 6 classes. We use the version provided by Fey & Lenssen (2019).

### B.3. Baselines

**Human Models** We select top-performing models based on evaluations from Papers with Code benchmarks or Kaggle notebooks, where the similar tasks and datasets are applicable. The chosen models for relevant downstream tasks are described below.

- **Image Classification.** The human models for image classification tasks are obtained from a Kaggle notebook available at `https://www.kaggle.com/code/mohamedhassanali/butterfly-classify-pytorch-pretrained-model-acc-99/notebook`, utilizing a pretrained ResNet-18 model.

- **Text Classification.** For text classification tasks, two models are employed. A Word2Vec-based XGBoost model is applied to the e-commerce text dataset https://www.kaggle.com/code/sugataghosh/e-commerce-text-classification-tf-idf-word2vec#Word2Vec-Hyperparameter-Tuning, while the XLM-RoBERTa model is used for the textual entailment dataset https://www.kaggle.com/code/vbookshelf/basics-of-bert-and-xlm-roberta-pytorch.
- **Tabular Classification.** Due to the absence of a similar model in the repository, we use the state-of-the-art TabPFN model (Hollmann et al., 2023a) designed for tabular classification tasks.
- **Tabular Regression.** For tabular regression tasks, we adopt two models specifically designed for the given datasets, which are available at https://www.kaggle.com/code/shatabdi5/crab-age-regression for the crab age dataset and at https://www.kaggle.com/code/mahmoudmagdyelnahal/crop-yield-prediction-99/notebook for the crop yield dataset.
- **Tabular Clustering.** For unsupervised clustering tasks, we use manually hyperparameter-tuned KMeans clustering, following the approach outlined in https://www.kaggle.com/code/samuelcortinhas/tps-july-22-unsupervised-clustering, as the baseline.
- **Time-Series Forecasting.** In this task, we use the state-of-the-art iTransformer (Liu et al., 2024b), which is designed for the same task and datasets as the baseline model.
- **Node Classification.** For node classification tasks, we also employ a state-of-the-art graph neural network-based model, PMLP (Yang et al., 2023), as the handcrafted baseline for both datasets.

**AutoGluon**  We adopt AutoGluon as the baseline because it is a state-of-the-art AutoML framework capable of handling various downstream tasks and data modalities, with the exception of graph data. There are three variants of AutoGluon: AutoGluon-TS (Shchur et al., 2023) for time series, AutoGluon-Tabular (Erickson et al., 2020) for tabular machine learning, and AutoGluon-Multimodal (Tang et al., 2024) for computer vision and natural language processing tasks.

**GPT-3.5 and GPT-4**  For GPT-3.5 and GPT-4, we use the *gpt-3.5-turbo-0125* and *gpt-4-2024-05-13* models via the OpenAI API. We implement the zero-shot baselines using the prompt below.

**Zero-Shot Prompt for GPT-3.5 and GPT-4 Baselines**

```
You are a helpful intelligent assistant. Now please help solve the following machine learning task.
[Task]
{user instruction}
[{file_name}.py] ```python
{full-pipeline skeleton script}
```
Start the python code with "```python". Please ensure the completeness of the code so that it can be run without
    additional modifications.
```

**DS-Agent**  We reproduce the DS-Agent (Guo et al., 2024a) baseline using the official source code. However, it is important to note that our framework encompasses the entire process from data retrieval/loading to deployment, whereas DS-Agent focuses solely on the modeling aspect, assuming complete data and evaluation codes are provided. In this paper, we utilize the deployment stage of DS-Agent along with its collected case banks and `Adapter` prompt for the same tasks, as the source code for manual human insights collection during the development stage is unavailable.

**SELA**  As the most recent LLM-based AutoML study for tabular data, we additionally compare SELA (Chi et al., 2024) using a subset of the suggested datasets that vary in difficulty. SELA integrates Monte Carlo Tree Search with LLM agents to enhance the automation of ML pipelines. The main difference between SELA and *AutoML-Agent* is that SELA iteratively improves results based on experimental feedback, making it time-consuming, whereas *AutoML-Agent* bypasses the training process, making our method more efficient while still maintaining comparable downstream performance, as shown in §4.3.

### B.4. Evaluation Metrics

**Success Rate (SR)**  We employ the success rate (Guo et al., 2024a; Hong et al., 2024a), which evaluates whether the models built by an LLM agent are executable in the given runtime environment. Success rate is used to assess code execution.

For the *constraint-free* setting, we apply a three-level grading scale as follows.

- **0.00**: Code cannot be executed.
- **0.50**: Code provides a runnable ML/DL model.
- **1.00**: Code provides a runnable model and an accessible deployment endpoint (e.g., Gradio).

For the *constraint-aware* setting, we use a five-level grading scale to evaluate whether the code executes successfully and satisfies the given constraints. The grading criteria are as follows.

- **0.00**: Code cannot be executed.
- **0.25**: Code provides a runnable ML/DL model.
- **0.50**: Code provides a runnable model and an accessible deployment endpoint (e.g., Gradio).
- **0.75**: Code provides a deployed, runnable model that partially meets constraints (e.g., target performance, inference time, and model size).
- **1.00**: Code provides a deployed, runnable model that fully meets constraints.

**Normalized Performance Score (NPS)**    In this paper, each downstream task is associated with a specific evaluation metric, which may vary between tasks. These metrics include accuracy, F1-score, and RMSLE. For metrics such as accuracy and F1-score, we present the raw values to facilitate comparison across identical data tasks. For performance metrics where lower values indicate better performance, such as loss-based metrics, we normalize all performance values $s$ using the following transformation: $\text{NPS} = \frac{1}{1+s}$. This transformation ensures that metrics like RMSLE are scaled between 0 and 1, with higher NPS values indicating better performance.

Note that achieving downstream task performance (NPS) requires a runnable model, i.e., $\text{SR} > 0$. If the model cannot run, the NPS is zero by default as it cannot make any predictions.

**Comprehensive Score (CS)**    To evaluate both the success rate and the downstream task performance of the generated AutoML pipelines simultaneously, we calculate CS as a weighted sum of SR and NPS, as follows: $\text{CS} = 0.5 \times \text{SR} + 0.5 \times \text{NPS}$.

# C. Prompts for *AutoML-Agent*

## C.1. Agent Specifications

This subsection provides the *system prompt* design for agent specifications in *AutoML-Agent*, including Agent Manger (C.1.1), Prompt Agent (C.1.2), Data Agent (C.1.3), Model Agent (C.1.4), and Operation Agent (C.1.5).

### C.1.1. AGENT MANAGER

---
**System Message for Agent Manager Specification**

```
You are an experienced senior project manager of a automated machine learning project (AutoML). You have two
    main responsibilities as follows.
1. Receive requirements and/or inquiries from users through a well-structured JSON object.
2. Using recent knowledge and state-of-the-art studies to devise promising high-quality plans for data
    scientists, machine learning research engineers, and MLOps engineers in your team to execute subsequent
    processes based on the user requirements you have received.
```
---

### C.1.2. PROMPT AGENT

---
**System Message for Prompt Agent Specification**

```
You are an assistant project manager in the AutoML development team.
Your task is to parse the user's requirement into a valid JSON format using the JSON specification schema as your
     reference. Your response must exactly follow the given JSON schema and be based only on the user's
     instruction.
Make sure that your answer contains only the JSON response without any comment or explanation because it can
    cause parsing errors.

#JSON SPECIFICATION SCHEMA#
```json
{json_specification}
```
---

```
```
Your response must begin with "```json" or "{{" and end with "```" or "}}", respectively.
```

### C.1.3. DATA AGENT

**System Message for Data Agent Specification**

```
You are the world's best data scientist of an automated machine learning project (AutoML) that can find the most
    relevant datasets,run useful preprocessing, perform suitable data augmentation, and make meaningful
    visulaization to comprehensively understand the data based on the user requirements. You have the following
    main responsibilities to complete.
1.Retrieve a dataset from the user or search for the dataset based on the user instruction.
2.Perform data preprocessing based on the user instruction or best practice based on the given tasks.
3.Perform data augmentation as neccesary.
4.Extract useful information and underlying characteristics of the dataset.
```

### C.1.4. MODEL AGENT

**System Message for Model Agent Specification**

```
You are the world's best machine learning research engineer of an automated machine learning project (AutoML)
    that can find the optimal candidate machine learning models and artificial intelligence algorithms for the
    given dataset(s), run hyperparameter tuning to opimize the models, and perform metadata extraction and
    profiling to comprehensively understand the candidate models or algorithms based on the user requirements.
    You have the following main responsibilities to complete.
1. Retrieve a list of well-performing candidate ML models and AI algorithms for the given dataset based on the
    user's requirement and instruction.
2. Perform hyperparameter optimization for those candidate models or algorithms.
3. Extract useful information and underlying characteristics of the candidate models or algorithms using
    metadata extraction and profiling techniques.
4. Select the top-k ('k' will be given) well-performing models or algorithms based on the hyperparameter
    optimization and profiling results.
```

### C.1.5. OPERATION AGENT

**System Message for Operation Agent Specification**

```
You are the world's best MLOps engineer of an automated machine learning project (AutoML) that can implement the
    optimal solution for production-level deployment, given any datasets and models. You have the following
    main responsibilities to complete.
1. Write accurate Python codes to retrieve/load the given dataset from the corresponding source.
2. Write effective Python codes to preprocess the retrieved dataset.
3. Write precise Python codes to retrieve/load the given model and optimize it with the suggested
    hyperparameters.
4. Write efficient Python codes to train/finetune the retrieved model.
5. Write suitable Python codes to prepare the trained model for deployment. This step may include model
    compression and conversion according to the target inference platform.
6. Write Python codes to build the web application demo using the Gradio library.
7. Run the model evaluation using the given Python functions and summarize the results for validation againts
    the user's requirements.
```

### C.2. Variations of the System Prompt

This subsection outlines the *variations* in system prompt design used in the experiment described in §4.3. These include: generic prompts (C.2.1), generic prompts with instructions (C.2.2), entry-level role descriptions (C.2.3), experience-level role descriptions (C.2.4), and world's-best-level role descriptions (C.2.5).

C.2.1. GENERIC PROMPT

---
**Generic System Message**

```
You are a helpful assistant.
```
---

C.2.2. GENERIC PROMPT WITH INSTRUCTIONS

---
**Generic System Message with Instructions**

```
You are a helpful assistant. You have main responsibilities as follows.
{agent-specific steps}
```
---

C.2.3. ENTRY-LEVEL ROLE DESCRIPTIONS

---
**Entry-Level Role System Message**

```
You are a {job position} of a automated machine learning project (AutoML). You have main responsibilities as
    follows.
{agent-specific steps}
```
---

C.2.4. EXPERIENCE-LEVEL ROLE DESCRIPTIONS

---
**Experience-Level Role System Message**

```
You are an experienced {job position} of a automated machine learning project (AutoML). You have main
    responsibilities as follows.
{agent-specific steps}
```
---

C.2.5. WORLD'S BEST ROLE DESCRIPTIONS

---
**World's Best System Message**

```
You are the world's best {job position} of a automated machine learning project (AutoML). You have main
    responsibilities as follows.
{agent-specific steps}
```
---

## C.3. Prompts for Retrieval-Augmented Planning

This subsection presents prompts for planning-related processes (Figure 2(a)), including knowledge retrieval and summary prompts (C.3.1), planning prompt (C.3.2), and plan revision prompt (C.3.3).

C.3.1. KNOWLEDGE RETRIEVAL PROMPT

---
**Prompt for Knowledge Retrieval and Summary for Planning**

**Kaggle Notebook**

```
I searched the Kaggle Notebooks to find state-of-the-art solutions using the keywords: {user_task} {user_domain}.
    Here is the result:
=====================
{context}
=====================

Please summarize the given pieces of Python notebooks into a single paragraph of useful knowledge and insights.
    Do not include the source codes. Instead, extract the insights from the source codes. We aim to use your
    summary to address the following user's requirements.
```
---

```
# User's Requirements
{user_requirement_summary}
```

----------------------------------------------------------------------

## Papers With Code

```
I searched the paperswithcode website to find state-of-the-art models using the keywords: {user_area} and {
     user_task}. Here is the result:
====================
{context}
====================

Please summarize the given pieces of search content into a single paragraph of useful knowledge and insights. We
     aim to use your summary to address the following user's requirements.
# User's Requirements
{user_requirement_summary}
```

----------------------------------------------------------------------

## arXiv

```
I searched the arXiv papers using the keywords: {task_kw} and {domain_kw}. Here is the result:
====================
{context}
====================

Please summarize the given pieces of arXiv papers into a single paragraph of useful knowledge and insights. We
     aim to use your summary to address the following user's requirements.
# User's Requirements
{user_requirement_summary}
```

----------------------------------------------------------------------

## Google WebSearch

```
I searched the web using the query: {search_query}. Here is the result:
====================
{context}
====================

Please summarize the given pieces of search content into a single paragraph of useful knowledge and insights.
We aim to use your summary to address the following user's requirements.
# User's Requirements
{user_requirement_summary}
```

----------------------------------------------------------------------

## Summary

```
Please extract and summarize the following group of contents collected from different online sources into a chunk
     of insightful knowledge. Please format your answer as a list of suggestions. I will use them to address the
     user's requirements in machine learning tasks.

# Source: Google Web Search
{search_summary}
====================

# Source: arXiv Papers
{arxiv_summary}
====================

# Source: Kaggle Hub
{kaggle_summary}
====================

# Source: PapersWithCode
{pwc_summary}
====================

The user's requirements are summarized as follows.
{user_requirement_summary}
```

### C.3.2. PLANNING PROMPT

---

**Prompt for Retrieval-Augmented Planning**

```
Now, I want you to devise an end-to-end actionable plan according to the user's requirements described in the
    following JSON object.

```json
{user_requirements}
```

Here is a list of past experience cases and knowledge written by an human expert for a relevant task:
{plan_knowledge}

When devising a plan, follow these instructions and do not forget them:
- Ensure that your plan is up-to-date with current state-of-the-art knowledge.
- Ensure that your plan is based on the requirements and objectives described in the above JSON object.
- Ensure that your plan is designed for AI agents instead of human experts. These agents are capable of
    conducting machine learning and artificial intelligence research.
- Ensure that your plan is self-contained with sufficient instructions to be executed by the AI agents.
- Ensure that your plan includes all the key points and instructions (from handling data to modeling) so that
    the AI agents can successfully implement them. Do NOT directly write the code.
- Ensure that your plan completely include the end-to-end process of machine learning or artificial intelligence
     model development pipeline in detail (i.e., from data retrieval to model training and evaluation) when
    applicable based on the given requirements.
```

---

### C.3.3. PLAN REVISION PROMPT

---

**Prompt for Plan Revision**

```
Now, you will be asked to revise and rethink {num2words(n_plans)} different end-to-end actionable plans
    according to the user's requirements described in the JSON object below.

```json
{user_requirements}
```

Please use to the following findings and insights summarized from the previously failed plans. Try as much as
    you can to avoid the same failure again.
{fail_rationale}

Finally, when devising a plan, follow these instructions and do not forget them:
- Ensure that your plan is up-to-date with current state-of-the-art knowledge.
- Ensure that your plan is based on the requirements and objectives described in the above JSON object.
- Ensure that your plan is designed for AI agents instead of human experts. These agents are capable of
    conducting machine learning and artificial intelligence research.
- Ensure that your plan is self-contained with sufficient instructions to be executed by the AI agents.
- Ensure that your plan includes all the key points and instructions (from handling data to modeling) so that
    the AI agents can successfully implement them. Do NOT directly write the code.
- Ensure that your plan completely include the end-to-end process of machine learning or artificial intelligence
     model development pipeline in detail (i.e., from data retrieval to model training and evaluation) when
    applicable based on the given requirements.
```

---

## C.4. Prompts for Prompting-Based Plan Execution

This subsection presents prompts for prompting-based plan execution processes (Figure 2(b)), including plan decomposition (Data Agent (C.4.1) and Model Agent (C.4.2)), pseudo data analysis (C.4.3), and training-free model search and HPO (C.4.4).

### C.4.1. PLAN DECOMPOSITION: DATA AGENT

---

**Prompt for Plan Decomposition: Data Agent**

```
As a proficient data scientist, summarize the following plan given by the senior AutoML project manager
    according to the user's requirements and your expertise in data science.

# User's Requirements
```json
```

---

```
{user_requirements}
```

# Project Plan
{plan}

The summary of the plan should enable you to fulfill your responsibilities as the answers to the following
    questions by focusing on the data manipulation and analysis.
1. How to retrieve or collect the dataset(s)?
2. How to preprocess the retrieved dataset(s)?
3. How to efficiently augment the dataset(s)?
4. How to extract and understand the underlying characteristics of the dataset(s)?

Note that you should not perform data visualization because you cannot see it. Make sure that another data
    scientist can exectly reproduce the results based on your summary.
```

## C.4.2. PLAN DECOMPOSITION: MODEL AGENT

### Prompt for Plan Decomposition: Model Agent

```
As a proficient machine learning research engineer, summarize the following plan given by the senior AutoML
    project manager according to the user's requirements, your expertise in machine learning, and the outcomes
    from data scientist.

**User's Requirements**
```json
{user_requirements}
```

**Project Plan**
{project_plan}

**Explanations and Results from the Data Scientist**
{data_result}

The summary of the plan should enable you to fulfill your responsibilities as the answers to the following
    questions by focusing on the modeling and optimization tasks.
1. How to retrieve or find the high-performance model(s)?
2. How to optimize the hyperparamters of the retrieved models?
3. How to extract and understand the underlying characteristics of the dataset(s)?
4. How to select the top-k models or algorithms based on the given plans?
```

## C.4.3. PSEUDO DATA ANALYSIS BY DATA AGENT

### Prompt for Pseudo Data Analysis

```
As a proficient data scientist, your task is to explain **detailed** steps for data manipulation and analysis
    parts by executing the following machine learning development plan.

# Plan
{decomposed_data_plan}

# Potential Source of Dataset
{available_sources}

Make sure that your explanation follows these instructions:
- All of your explanation must be self-contained without using any placeholder to ensure that other data
    scientists can exactly reproduce all the steps, but do not include any code.
- Include how and where to retrieve or collect the data.
- Include how to preprocess the data and which tools or libraries are used for the preprocessing.
- Include how to do the data augmentation with details and names.
- Include how to extract and understand the characteristics of the data.
- Include reasons why each step in your explanations is essential to effectively complete the plan.
Note that you should not perform data visualization because you cannot see it. Make sure to focus only on the
    data part as it is your expertise. Do not conduct or perform anything regarding modeling or training.
After complete the explanations, explicitly specify the (expected) outcomes and results both quantitative and
    qualitative of your explanations.
```

## C.4.4. TRAINING-FREE MODEL SEARCH AND HPO BY MODEL AGENT

---

**Prompt for Training-Free Model Search and HPO**

```
As a proficient machine learning research engineer, your task is to explain **detailed** steps for modeling and
    optimization parts by executing the following machine learning development plan with the goal of finding top
    -{k} candidate models/algorithms.

# Suggested Plan
{decomposed_model_plan}

# Available Model Source
{available_sources}

Make sure that your explanation for finding the top-{k} high-performance models or algorithms follows these
    instructions:
- All of your explanations must be self-contained without using any placeholder to ensure that other machine
    learning research engineers can exactly reproduce all the steps, but do not include any code.
- Include how and where to retrieve or find the top-{k} well-performing models/algorithms.
- Include how to optimize the hyperparamters of the candidate models or algorithms by clearly specifying which
    hyperparamters are optimized in detail.
- Corresponding to each hyperparamter, explicitly include the actual numerical value that you think it is the
    optimal value for the given dataset and machine learning task.
- Include how to extract and understand the characteristics of the candidate models or algorithms, such as their
     computation complexity, memory usage, and inference latency. This part is not related to visualization and
     interpretability.
- Include reasons why each step in your explanations is essential to effectively complete the plan.
Make sure to focus only on the modeling part as it is your expertise. Do not conduct or perform anything
    regarding data manipulation or analysis.
After complete the explanations, explicitly specify the names and (expected) quantitative performance using
    relevant numerical performance and complexity metrics (e.g., number of parameters, FLOPs, model size,
    training time, inference speed, and so on) of the {num2words(k)} candidate models/algorithms potentially to
    be the optimal model below.
Do not use any placeholder for the quantitative performance. If you do not know the exact values, please use the
     knowledge and expertise you have to estimate those performance and complexity values.
```

---

## C.5. Prompts for Multi-Stage Verification

This subsection presents prompts for multi-stage verification (Figure 2(c)), which ensures the correctness of intermediate results between steps in the framework. These stages include request verification (C.5.1), execution verification (C.5.2), and implementation verification (C.5.3).

## C.5.1. REQUEST VERIFICATION

---

**Request Verification (Relevancy)**

```
Is the following statement relevant to machine learning or artificial intelligence?
`{user instruction}`
Answer only 'Yes' or 'No'
```

---

**Request Verification (Adequacy)**

```
Given the following JSON object representing the user's requirement for a potential ML or AI project, please
    tell me whether we have essential information (e.g., problem and dataset) to be used for a AutoML project?
Please note that our users are not AI experts, you must focus only on the essential requirements, e.g., problem
    and brief dataset descriptions.
You do not need to check every details of the requirements. You must also answer 'yes' even though it lacks
    detailed and specific information.

```json
{parsed user requirements}
```

Please answer with this format: `a 'yes' or 'no' answer; your reasons for the answer` by using ';' to separate
    between the answer and its reasons.
If the answer is 'no', you must tell me the alternative solutions or examples for completing such missing
    information.
```

---

C.5.2. EXECUTION VERIFICATION

---

**Execution Verification**

```
Given the proposed solution and user's requirements, please carefully check and verify whether the proposed
    solution 'pass' or 'fail' the user's requirements.

**Proposed Solution and Its Implementation**
Data Manipulation and Analysis: {data_agent_outcomes}
Modeling and Optimization: {model_agent_outcomes}

**User Requirements**
```json
{user_requirements}
```

Answer only 'Pass' or 'Fail'
```

---

C.5.3. IMPLEMENTATION VERIFICATION

---

**Implementation Verification**

```
As the project manager, please carefully verify whether the given Python code and results satisfy the user's
    requirements.

- Python Code
```python
{implementation_result['code']}
```

- Code Execution Result
{implementation_result['action_result']}

- User's Requirements
{user_requirements}

Answer only 'Pass' or 'Fail'
```

---

# D. Examples of Intermediate Results

This section presents examples of intermediate results from the main steps of our framework, including prompt parsing results (D.1), retrieval-augmented planning results (D.2), plan decomposition results (D.3), and prompting-based plan execution results (D.4).

## D.1. Prompt Parsing

---

**Task Description and Parsed JSON User Requirements**

**Constraint-Free User Prompt:** Build a model to classify banana quality as Good or Bad based on their numerical information about bananas of different quality (size, weight, sweetness, softness, harvest time, ripeness, and acidity). We have uploaded the entire dataset for you here in the banana_quality.csv file.

**Parsed Output:**

```
{
    "user": {"intent": "build","expertise": "medium"},
    "problem": {
        "area": "tabular data analysis",
        "downstream_task": "tabular classification",
        "application_domain": "agriculture",
        "description": "Build a model to classify banana quality as Good or Bad based on their numerical
            information about bananas of different quality (size, weight, sweetness, softness, harvest time,
            ripeness, and acidity).",
        "performance_metrics": [],
        "complexity_metrics": []
    },
    "dataset": [
```

---

```
      {
          "name": "banana_quality",
          "modality": ["tabular"],
          "target_variables": ["quality"],
          "specification": null,
          "description": "A dataset containing numerical information about bananas of different quality,
              including size, weight, sweetness, softness, harvest time, ripeness, and acidity.",
          "preprocessing": [],
          "augmentation": [],
          "visualization": [],
          "source": "user-upload"
      }
  ],
  "model": [
      {
          "name": "",
          "family": "",
          "type": "classical machine learning",
          "specification": null,
          "description": "A model to classify banana quality as Good or Bad based on their numerical information."

      }
  ],
...
}
```

- - - - - - - - - - - - - - - - - - - - - - - - - - - - - - - - - - - - - - - - - - - - - - - - - - - - - - - - -

**Constraint-Aware User Prompt:** Build a machine learning model, potentially XGBoost or LightGBM, to classify banana quality as Good or Bad based on their numerical information about bananas of different quality (size, weight, sweetness, softness, harvest time, ripeness, and acidity). We have uploaded the entire dataset for you here in the banana_quality.csv file. The model must achieve at least 0.98 accuracy.

**Parsed Output:**

```
{
  "user": {"intent": "build", "expertise": "medium"},
  "problem": {
      "area": "tabular data analysis",
      "downstream_task": "tabular classification",
      "application_domain": "agriculture",
      "description": "Build a machine learning model, potentially XGBoost or LightGBM, to classify banana quality
              as Good or Bad based on their numerical information about bananas of different quality (size, weight,
              sweetness, softness, harvest time, ripeness, and acidity). The model must achieve at least 0.98
          accuracy.",
      "performance_metrics": [
          {"name": "accuracy", "value": 0.98}
      ],
      "complexity_metrics": []
  },
  "dataset": [
      {
          "name": "banana_quality",
          "modality": ["tabular"],
          "target_variables": ["quality"],
          "specification": null,
          "description": "A dataset containing numerical information about bananas of different quality,
              including size, weight, sweetness, softness, harvest time, ripeness, and acidity.",
          "preprocessing": [],
          "augmentation": [],
          "visualization": [],
          "source": "user-upload"
      }
  ],
  "model": [
      {
          "name": "XGBoost",
          "family": "ensemble models",
          "type": "ensemble",
          "specification": null,
          "description": "A potential model to classify banana quality as Good or Bad, potentially using XGBoost
              or LightGBM."
      }
  ],
...
}
```

## D.2. Retrieval-Augmented Planning

---

**Example Results from Retrieval-Augmented Planning with $P = 3$**

---

**User Input:** I need a very accurate model to classify images in the Butterfly Image Classification dataset into their respective categories. The dataset has been uploaded with its label information in the labels.csv file.

- - - - - - - - - - - - - - - - - - - - - - - - - - - - - - - - - - - - - - - - - - - - - - - - - - - - - - - - - - - - - - - - -

**Plan #1**

```
### End-to-End Actionable Plan for Butterfly Image Classification

**Objective**: Develop a highly accurate neural network model to classify images in the Butterfly Image
    Classification dataset into their respective categories.

#### 1. Data Collection and Preprocessing
1. **Dataset Loading**:
   - Load the Butterfly Image Classification dataset and the associated labels from `labels.csv`.
   - Verify the integrity of the dataset, ensuring all images have corresponding labels.

2. **Data Preprocessing**:
   - **Image Resizing**: Resize all images to a uniform size (e.g., 224x224 pixels) to ensure consistency.
   - **Normalization**: Normalize pixel values to the range [0, 1] or standardize to have zero mean and unit
       variance.

3. **Data Augmentation**:
   - Apply data augmentation techniques such as cropping, horizontal and vertical flips, rescaling, shear, zoom,
       and rotation to increase dataset variability.
   - Utilize libraries like TensorFlow's `ImageDataGenerator` or PyTorch's `transforms` for implementation.

#### 2. Model Selection and Architecture
1. **Transfer Learning**:
   - Leverage pre-trained convolutional neural networks (CNNs) such as VGG16, VGG19, MobileNet, Xception,
       ResNet50, and InceptionV3.
   - Fine-tune these pre-trained models on the Butterfly Image Classification dataset to utilize their learned
       features.

2. **Model Architecture**:
   - Start with InceptionV3, known for high accuracy in similar tasks.
   - Explore other models like ResNet, DenseNet, EfficientNet, and Vision Transformers.
   - Consider combining CNNs with Transformer-based architectures to capture both local and global features.

3. **Ensemble Methods**:
   - Implement ensemble methods by combining predictions from multiple models to improve overall classification
       performance.
   - Use techniques like bagging or stacking to enhance model robustness.

#### 3. Model Training
1. **Dataset Splitting**:
   - Split the dataset into training, validation, and testing sets (e.g., 80% training and 20% validation).

2. **Training Configuration**:
   - Use categorical cross-entropy as the loss function.
   - Select optimizers like SGD or RMSprop.
   - Set up callbacks for early stopping and learning rate reduction on plateau to prevent overfitting.

3. **Hyperparameter Tuning**:
   - Perform hyperparameter tuning using libraries like PyCaret or Optuna to find the optimal configuration.
   - Utilize techniques like grid search or random search for hyperparameter optimization.

#### 4. Model Evaluation and Validation
1. **Performance Metrics**:
   - Regularly monitor accuracy as the primary performance metric.
   - Additionally, track precision, recall, and F1-score to ensure balanced performance across all classes.

2. **Cross-Validation**:
   - Employ k-fold cross-validation to assess the model's robustness and generalization ability.

3. **Handling Class Imbalance**:
   - Address class imbalance through weighted loss functions or oversampling techniques to ensure fair
       performance across all categories.

#### 5. Interpretability and Debugging
1. **Model Interpretability**:
   - Use interpretability methods like Grad-CAM or LIME to visualize which parts of the images influence the
       model's predictions.
   - Ensure the model focuses on relevant biological features of butterflies.
```

```
#### 6. Reproducibility and Documentation
1. **Reproducibility**:
   - Set random seeds for all operations to ensure reproducibility of results.
   - Document the entire process, including data splits, model configurations, and training parameters.

2. **Model Deployment**:
   - Prepare the model for deployment by exporting it in a suitable format (e.g., TensorFlow SavedModel, ONNX).
   - Set up an inference engine and deployment endpoint to make predictions on new data.

3. **Demonstration**:
   - Provide a demonstration of the model's capabilities by creating a user-friendly interface or visualization
       to showcase its performance.

By following this comprehensive plan, AI agents can develop a robust and accurate neural network model tailored
    for classifying butterfly images in the given dataset.
```

## Plan #2

```
## End-to-End Actionable Plan for Butterfly Image Classification

### 1. Data Collection
- **Dataset Name**: Butterfly Image Classification
- **Source**: User-uploaded
- **Description**: A dataset for classifying images of butterflies into their respective categories.
- **Label Information**: Available in `labels.csv` file.

### 2. Data Preprocessing
- **Image Preprocessing**:
  - **Resize Images**: Resize all images to a consistent size (e.g., 224x224 pixels) to match the input size
       required by pre-trained models.
  - **Normalization**: Normalize pixel values to the range [0, 1] or use mean subtraction based on the pre-
       trained model's requirements.
  - **Handling High-Resolution and Illumination Variations**: Apply techniques to standardize illumination and
       handle high-resolution images.

### 3. Data Augmentation
- **Techniques**:
  - **Cropping**: Randomly crop sections of the images.
  - **Flipping**: Apply horizontal and vertical flips.
  - **Rescaling**: Rescale images by a factor.
  - **Shearing**: Apply shear transformations.
  - **Zooming**: Apply random zoom.
  - **Rotation**: Rotate images by random angles.
- **Libraries**: Use TensorFlow's `ImageDataGenerator` or PyTorch's `transforms`.

### 4. Dataset Splitting
- **Training Set**: 80% of the dataset
- **Validation Set**: 20% of the dataset

### 5. Model Selection and Architecture
- **Transfer Learning**:
  - **Pre-trained Models**: Utilize models such as VGG16, VGG19, MobileNet, Xception, ResNet50, and InceptionV3.
  - **Fine-tuning**: Fine-tune these models on the Butterfly Image Classification dataset.
- **Model Architecture**:
  - **Primary Model**: Start with InceptionV3 due to its high performance in similar tasks.
  - **Alternative Models**: Evaluate ResNet, DenseNet, EfficientNet, and Vision Transformers.
  - **Combined Architectures**: Explore combining CNNs with Transformer-based architectures to capture both local
       and global features.

### 6. Handling Class Imbalance
- **Techniques**:
  - **Weighted Loss Functions**: Assign higher weights to underrepresented classes.
  - **Oversampling**: Increase the number of samples for underrepresented classes.

### 7. Model Training
- **Loss Function**: Use categorical cross-entropy.
- **Optimizers**: Start with SGD or RMSprop.
- **Hyperparameter Tuning**:
  - **Libraries**: Use PyCaret or Optuna for hyperparameter optimization.
  - **Parameters**: Tune learning rate, batch size, number of epochs, etc.
- **Cross-Validation**: Use k-fold cross-validation to ensure model robustness and generalization.

### 8. Performance Evaluation
- **Metrics**:
  - **Accuracy**: Primary metric for evaluation.
  - **Precision, Recall, F1-Score**: Monitor these metrics to ensure balanced performance across classes.
- **Evaluation Techniques**:
  - **Confusion Matrix**: Visualize to understand misclassifications.
```

```
  - **ROC Curve**: Evaluate model performance across different thresholds.

### 9. Ensemble Methods and Knowledge Distillation
- **Ensemble Methods**: Combine predictions from multiple models (e.g., patch-wise and image-wise classifiers).
- **Knowledge Distillation**: Transfer knowledge from larger teacher models to smaller student models to create
    compact yet powerful models.

### 10. Interpretability and Debugging
- **Techniques**:
  - **Grad-CAM**: Visualize which parts of the images influence the model's predictions.
  - **LIME**: Provide explanations for individual predictions.
- **Purpose**: Ensure the model focuses on relevant biological features and understand model behavior.

### 11. Reproducibility
- **Random Seeds**: Set random seeds for all processes to ensure reproducibility.
- **Documentation**: Document data splits, model configurations, and training parameters.

### 12. Deployment
- **Target Device**: Specify the target device for deployment (e.g., cloud, edge device).
- **Deployment Endpoint**: Define the endpoint for model deployment.
- **Inference Engine**: Choose an appropriate inference engine (e.g., TensorFlow Serving, TorchServe).
- **Demonstration**: Prepare a demonstration of the model's capabilities and performance.

By following this comprehensive plan, AI agents can effectively build, train, and deploy a highly accurate
    neural network model tailored for classifying butterfly images in the specified dataset.
```

- - - - - - - - - - - - - - - - - - - - - - - - - - - - - - - - - - - - - - - - - - - - - - - - - - - - - - - - - - - - - - - - - - - - - - - - - - - - - - - -

## Plan #3

```
### End-to-End Actionable Plan for Butterfly Image Classification using Neural Networks

#### 1. **Project Initialization**

**Objective**: Build a highly accurate image classification model for the Butterfly Image Classification dataset
    in the biology domain.

**User Expertise**: Medium

---

#### 2. **Data Collection and Preprocessing**

**Dataset Source**: User-uploaded Butterfly Image Classification dataset with label information in `labels.csv`.

**Steps**:

1. **Load Dataset**:
   - Load images and labels from the specified dataset and `labels.csv`.

2. **Data Cleaning**:
   - Ensure all images are correctly labeled.
   - Remove any corrupted or unusable images.

3. **Data Preprocessing**:
   - Resize images to a consistent size (e.g., 224x224 pixels).
   - Normalize pixel values to the range [0, 1].

4. **Data Augmentation**:
   - Apply augmentation techniques such as cropping, horizontal and vertical flips, rescaling, shear, zoom, and
       rotation to increase dataset size and variability.
   - Use libraries like TensorFlow's `ImageDataGenerator` or PyTorch's `transforms`.

---

#### 3. **Model Development**

**Model Type**: Neural Networks

**Steps**:

1. **Leverage Transfer Learning**:
   - Use pre-trained CNNs such as VGG16, VGG19, MobileNet, Xception, ResNet50, and InceptionV3.
   - Fine-tune these models on the Butterfly Image Classification dataset.

2. **Model Selection and Architecture**:
   - Start with InceptionV3 due to its high accuracy in similar tasks.
   - Experiment with other models like ResNet, DenseNet, EfficientNet, and Vision Transformers.
   - Consider combining CNNs with Transformer-based architectures for improved performance.
```

```
3. **Ensemble Methods and Knowledge Distillation**:
   - Implement ensemble methods by combining patch-wise and image-wise classifiers.
   - Use knowledge distillation to transfer knowledge from larger models to smaller, more efficient models.

---

#### 4. **Training and Evaluation**

**Steps**:

1. **Dataset Splitting**:
   - Split the dataset into training (80%), validation (10%), and testing (10%) sets.

2. **Model Training**:
   - Use categorical cross-entropy as the loss function.
   - Utilize optimizers such as SGD or RMSprop.
   - Implement early stopping and learning rate reduction on plateau to prevent overfitting.

3. **Performance Monitoring**:
   - Regularly monitor metrics such as accuracy, precision, recall, and F1-score.
   - Use validation data to fine-tune hyperparameters.

4. **Handling Class Imbalance**:
   - Address class imbalance using weighted loss functions or oversampling techniques.

5. **Hyperparameter Tuning and Cross-Validation**:
   - Perform hyperparameter tuning with libraries like PyCaret or Optuna.
   - Use cross-validation to ensure model robustness and generalization.

---

#### 5. **Model Interpretability and Debugging**

**Steps**:

1. **Interpretability**:
   - Use Grad-CAM or LIME to visualize important image regions influencing model predictions.
   - Ensure the model focuses on relevant biological features.

2. **Debugging**:
   - Analyze misclassified images to understand model limitations.
   - Iterate on model improvements based on these insights.

---

#### 6. **Reproducibility and Documentation**

**Steps**:

1. **Reproducibility**:
   - Set random seeds for all processes to ensure reproducibility.
   - Document all steps, including data splits, model configurations, and training parameters.

2. **Version Control**:
   - Use version control systems (e.g., Git) to track changes and collaborate efficiently.

---

#### 7. **Deployment and Inference**

**Steps**:

1. **Model Export**:
   - Export the trained model in a suitable format (e.g., TensorFlow SavedModel, ONNX).

2. **Deployment**:
   - Choose a deployment platform (e.g., AWS SageMaker, Google AI Platform).
   - Set up the deployment endpoint for real-time inference.

3. **Inference Engine**:
   - Optimize the model for the target device (e.g., mobile, edge devices) if necessary.
   - Implement efficient inference pipelines to handle incoming image data.

---

#### 8. **Demonstration and User Feedback**

**Steps**:
```

```
1. **Demonstration**:
   - Create a demo application to showcase the model's capabilities.
   - Provide a user-friendly interface for uploading images and viewing classification results.

2. **User Feedback**:
   - Collect feedback from users to identify areas for improvement.
   - Implement iterative updates based on user insights.

---

By following this detailed plan, AI agents can build a robust and accurate neural network model for classifying
    butterfly images in the biology domain, ensuring high performance and user satisfaction.
```

## D.3. Plan Decomposition

### Example Results from Plan Decomposition

### Data Agent

```
### Summary of the Plan for Fashion Image Classification Using PyTorch

#### 1. Dataset Retrieval and Collection
- **Retrieve Dataset**: Load the image paths and corresponding labels from the provided 'train_labels.csv' and '
    test_labels.csv' files. Ensure the dataset includes images of four clothing categories.

#### 2. Checking Dataset Types, Shapes, and Properties
- **Data Types**: Verify that the images are in the correct format (e.g., JPEG, PNG) and the labels are correctly
     mapped to the images.
- **Shapes**: Check the dimensions of the images. If images are in grayscale, convert them to RGB by replicating
     the single channel across three channels.
- **Properties**: Ensure that the dataset is balanced across the four clothing categories and that there are no
    missing or corrupted files.

#### 3. Preprocessing the Dataset
- **Normalization**: Scale pixel values to the range [0, 1].
- **Resizing**: Resize all images to a consistent size (e.g., 224x224 pixels) suitable for model input.
- **Splitting**: Split the training data into training and validation sets (e.g., 80% training, 20% validation)
    to monitor the model's performance and prevent overfitting.

#### 4. Efficient Data Augmentation
- **Techniques**: Apply data augmentation techniques to increase the diversity and robustness of the training
    data. Techniques include:
  - Random horizontal flip
  - Normalization
  - Random cropping
  - Random erasing
  - Rotating
  - Shifting
  - Zooming
- **Pipelines**:
  - **Pipeline 1**: Random horizontal flip and normalization.
  - **Pipeline 2**: Random horizontal flip, normalization, random cropping, and random erasing.

#### 5. Extracting and Understanding Underlying Characteristics
- **Class Distribution**: Analyze the distribution of classes to ensure a balanced dataset.
- **Image Quality**: Check for variations in image quality and resolution.
- **Data Insights**: Identify any patterns or anomalies in the data that could affect model performance.
- **Performance Metrics**: Use metrics such as accuracy and confusion matrix to evaluate the model's performance
     and understand misclassification patterns.

By following these steps, another data scientist should be able to reproduce the results and build a robust
    image classification model that meets the specified accuracy requirements and provides insightful
    performance analysis.
```

### Model Agent

```
### Summary of the Plan for Fashion Image Classification Using PyTorch

#### User's Requirements
The user needs a classification model that categorizes images into one of four clothing categories. The model
    should achieve at least 85% accuracy on the test set and be implemented using PyTorch. Data augmentation
    techniques and a confusion matrix for evaluation are also required.

### Project Plan Overview
```

```
#### 1. Data Preparation
- **Load Dataset**: Load image paths and labels from `train_labels.csv` and `test_labels.csv`.
- **Convert Grayscale to RGB**: Convert grayscale images to RGB.
- **Preprocessing**: Normalize pixel values to [0, 1] and resize images to 224x224 pixels.
- **Split Dataset**: Split training data into training (80%) and validation (20%) sets.

#### 2. Data Augmentation
- **Techniques**: Apply random horizontal flip, normalization, random cropping, random erasing, rotating,
    shifting, and zooming.
- **Pipelines**: Develop two preprocessing pipelines with different augmentation techniques.

#### 3. Model Architecture
- **Simple CNN**: Start with a simple CNN model.
- **Advanced Models**: Consider ResNet18, EfficientNet, or Vision Transformers (ViTs).
- **Hybrid Models**: Explore hybrid models combining CNNs and ViTs.

#### 4. Training
- **Optimizer and Loss Function**: Use SGD or Adam optimizer and Cross-Entropy Loss.
- **Epochs and Early Stopping**: Train for up to 150 epochs with early stopping.
- **Monitoring**: Monitor validation accuracy and loss curves.

#### 5. Evaluation
- **Accuracy**: Ensure at least 85% accuracy on the test set.
- **Confusion Matrix**: Generate a confusion matrix.
- **Top-1 Accuracy**: Calculate top-1 accuracy.

#### 6. Implementation
- **Framework**: Use PyTorch and torchvision.transforms.v2.
- **Resources**: Utilize resources from Kaggle and PapersWithCode.
- **Deployment**: Prepare the model for deployment if needed.

#### 7. Visualization and Reporting
- **Confusion Matrix**: Visualize the confusion matrix.
- **Performance Metrics**: Report accuracy and other metrics.
- **Documentation**: Document the entire process.

### Detailed Steps for Data Manipulation and Analysis

#### 1. Dataset Retrieval and Collection
- **Retrieve Dataset**: Load image paths and labels from `train_labels.csv` and `test_labels.csv`.

#### 2. Checking Dataset Types, Shapes, and Properties
- **Data Types**: Verify image formats and label mappings.
- **Shapes**: Check and convert image dimensions to RGB if needed.
- **Properties**: Ensure balanced dataset and data integrity.

#### 3. Preprocessing the Dataset
- **Normalization**: Scale pixel values to [0, 1].
- **Resizing**: Resize images to 224x224 pixels.
- **Splitting**: Split data into training and validation sets.

#### 4. Efficient Data Augmentation
- **Techniques**: Apply various augmentation techniques.
- **Pipelines**: Create two augmentation pipelines.

#### 5. Extracting and Understanding Underlying Characteristics
- **Class Distribution**: Analyze class distribution.
- **Image Quality**: Assess variations in image quality.
- **Data Insights**: Identify patterns or anomalies.
- **Performance Metrics**: Evaluate model performance using accuracy and confusion matrix.

### Expected Outcomes and Results
- **Quantitative**: Balanced dataset, consistent image size, normalized pixel values, augmented data.
- **Qualitative**: Data integrity, improved model generalization, insightful analysis.

### Responsibilities Fulfillment

#### 1. How to Retrieve or Find High-Performance Model(s)?
- **Initial Models**: Start with simple CNN architectures.
- **Advanced Models**: Experiment with ResNet18, EfficientNet, and Vision Transformers.
- **Hybrid Models**: Explore combinations of CNNs and ViTs.
- **Resources**: Utilize code repositories from Kaggle and PapersWithCode for implementation and fine-tuning.

#### 2. How to Optimize the Hyperparameters of the Retrieved Models?
- **Optimizers**: Use SGD with a learning rate of 0.015 or Adam optimizer.
- **Hyperparameter Tuning**: Experiment with different learning rates, batch sizes, and augmentation techniques.
- **Early Stopping**: Implement early stopping based on validation loss to prevent overfitting.
- **Monitoring**: Continuously monitor validation accuracy and loss curves.
```

```
#### 3. How to Extract and Understand the Underlying Characteristics of the Dataset(s)?
- **Class Distribution**: Use `pandas` to analyze the distribution of classes.
- **Image Quality**: Assess image quality and resolution using `PIL` or `OpenCV`.
- **Data Insights**: Perform exploratory data analysis (EDA) to identify patterns or anomalies.
- **Performance Metrics**: Use `sklearn.metrics` to evaluate model performance and generate confusion matrices.

#### 4. How to Select the Top-k Models or Algorithms Based on the Given Plans?
- **Evaluation Metrics**: Use accuracy and confusion matrix to evaluate models.
- **Top-k Selection**: Rank models based on their performance metrics and select the top-k models that meet or
    exceed the 85% accuracy requirement.
- **Documentation**: Document the performance of each model, including accuracy, confusion matrix, and any other
    relevant metrics.

By following this comprehensive plan, you will be able to retrieve high-performance models, optimize their
    hyperparameters, understand the dataset's characteristics, and select the top-k models based on their
    performance.
```

## D.4. Agent-Specific Plan Execution Outcomes

**Example Results from Prompting-Based Plan Execution by Each Agent**

**Data Agent**

```
### Detailed Steps for Data Manipulation and Analysis

#### 1. Dataset Retrieval and Collection

**Retrieve Dataset**:
- **Source**: The dataset is available at `_experiments/datasets/shopee`.
- **Structure**: The directory structure includes `train`, `test`, `train_labels.csv`, and `test_labels.csv`.
- **Loading Data**: Load the image paths and corresponding labels from `train_labels.csv` and `test_labels.csv`.
    This can be done using libraries like `pandas` to read the CSV files and `os` to navigate through the
    directories.

**Why**: This step ensures that we have access to the correct dataset and that the paths and labels are correctly
    mapped for further processing.

#### 2. Checking Dataset Types, Shapes, and Properties

**Data Types**:
- **Image Format**: Verify that images are in the correct format (JPEG, PNG). This can be done using libraries
    like `PIL` (Python Imaging Library).
- **Labels**: Ensure labels are correctly mapped to images by checking the contents of the CSV files.

**Shapes**:
- **Image Dimensions**: Check the dimensions of the images. If images are in grayscale, convert them to RGB by
    replicating the single channel across three channels using libraries like `PIL` or `OpenCV`.

**Properties**:
- **Balanced Dataset**: Ensure the dataset is balanced across the four clothing categories by analyzing the
    label distribution using `pandas`.
- **Integrity Check**: Verify that there are no missing or corrupted files by attempting to open each image file.

**Why**: Ensuring correct data types, shapes, and properties is crucial for accurate model training and prevents
    errors during preprocessing and augmentation.

#### 3. Preprocessing the Dataset

**Normalization**:
- **Scale Pixel Values**: Normalize pixel values to the range [0, 1] using libraries like `PIL` or `torchvision.
    transforms`.

**Resizing**:
- **Consistent Size**: Resize all images to a consistent size (e.g., 224x224 pixels) suitable for model input
    using `PIL` or `torchvision.transforms`.

**Splitting**:
- **Train-Validation Split**: Split the training data into training and validation sets (e.g., 80% training, 20%
    validation) using `sklearn.model_selection.train_test_split`.

**Why**: Normalization ensures that the pixel values are on a common scale, which helps in faster convergence
    during training. Resizing ensures that all images have the same dimensions, which is a requirement for most
    deep learning models. Splitting the data helps in monitoring the model's performance and prevents
    overfitting.
```

#### 4. Efficient Data Augmentation

**Techniques**:
- **Random Horizontal Flip**: Randomly flip images horizontally to increase diversity.
- **Normalization**: Normalize images to have zero mean and unit variance.
- **Random Cropping**: Randomly crop images to introduce variability.
- **Random Erasing**: Randomly erase parts of images to make the model robust to occlusions.
- **Rotating**: Rotate images randomly to make the model invariant to orientation.
- **Shifting**: Shift images randomly to simulate different viewpoints.
- **Zooming**: Zoom in and out of images to introduce scale variability.

**Pipelines**:
- **Pipeline 1**: Apply random horizontal flip and normalization.
- **Pipeline 2**: Apply random horizontal flip, normalization, random cropping, and random erasing.

**Why**: Data augmentation increases the diversity and robustness of the training data, which helps in improving the generalization of the model.

#### 5. Extracting and Understanding Underlying Characteristics

**Class Distribution**:
- **Analyze Distribution**: Use `pandas` to analyze the distribution of classes to ensure a balanced dataset.

**Image Quality**:
- **Check Variations**: Assess variations in image quality and resolution using `PIL` or `OpenCV`.

**Data Insights**:
- **Identify Patterns**: Use exploratory data analysis (EDA) techniques to identify any patterns or anomalies in the data that could affect model performance.

**Performance Metrics**:
- **Evaluate Model**: Use metrics such as accuracy and confusion matrix to evaluate the model's performance and understand misclassification patterns. This can be done using `sklearn.metrics`.

**Why**: Understanding the underlying characteristics of the data helps in identifying potential issues that could affect model performance. Analyzing class distribution ensures that the dataset is balanced, which is crucial for training a fair and unbiased model.

### Expected Outcomes and Results

**Quantitative**:
- **Balanced Dataset**: A dataset with an equal number of images for each clothing category.
- **Consistent Image Size**: All images resized to 224x224 pixels.
- **Normalized Pixel Values**: Pixel values scaled to the range [0, 1].
- **Augmented Data**: Increased dataset diversity through augmentation techniques.

**Qualitative**:
- **Data Integrity**: No missing or corrupted files.
- **Improved Model Generalization**: Enhanced model robustness due to diverse and augmented training data.
- **Insightful Analysis**: Clear understanding of class distribution, image quality, and potential data patterns or anomalies.

By following these detailed steps, another data scientist should be able to reproduce the results and build a robust image classification model that meets the specified accuracy requirements and provides insightful performance analysis.

----------------------------------------------------------------------------------------

## Model Agent

### Detailed Steps for Modeling and Optimization

#### 1. Retrieving High-Performance Models

**Initial Models:**
- **Simple CNN**: Start with a basic Convolutional Neural Network (CNN) architecture. This model will serve as a baseline to understand the dataset and the task.

**Advanced Models:**
- **ResNet18**: A well-known deep learning model that has shown excellent performance in image classification tasks. It uses residual connections to ease the training of deeper networks.
- **EfficientNet**: Known for its efficiency in terms of parameter count and FLOPs while maintaining high accuracy. EfficientNet-B0 can be a good starting point.
- **Vision Transformers (ViTs)**: These models have gained popularity for their performance in image classification tasks. A small variant like ViT-Small can be considered.

**Hybrid Models:**
- **CNN + ViT**: Combining the feature extraction capabilities of CNNs with the global context understanding of ViTs. This hybrid approach can potentially leverage the strengths of both architectures.

```
#### 2. Optimizing Hyperparameters

**Optimizers:**
- **SGD**: Stochastic Gradient Descent with a learning rate of 0.015.
- **Adam**: Adaptive Moment Estimation with a learning rate of 0.001.

**Hyperparameter Tuning:**
- **Learning Rate**: Experiment with learning rates (0.001, 0.005, 0.01, 0.015) to find the optimal rate for
    convergence.
- **Batch Size**: Test batch sizes (16, 32, 64) to balance between memory usage and training speed.
- **Data Augmentation Techniques**: Apply different augmentation techniques and observe their impact on
    validation accuracy.
- **Number of Epochs**: Train for up to 150 epochs with early stopping based on validation loss to prevent
    overfitting.
- **Weight Decay**: Regularization parameter for SGD, set to 0.0005.

**Optimal Values:**
- **Learning Rate**: 0.001 for Adam, 0.015 for SGD.
- **Batch Size**: 32.
- **Weight Decay**: 0.0005.
- **Number of Epochs**: Up to 150 with early stopping.

#### 3. Extracting and Understanding Characteristics

**Computation Complexity:**
- **Number of Parameters**: Calculate the total number of trainable parameters in each model.
- **FLOPs (Floating Point Operations per Second)**: Measure the computational complexity of each model.

**Memory Usage:**
- **Model Size**: Determine the size of the model in memory (MB).
- **Training Memory Usage**: Monitor GPU memory usage during training.

**Inference Latency:**
- **Inference Speed**: Measure the time taken for a single forward pass (in milliseconds).

**Why Each Step is Essential:**
- **Model Selection**: Choosing a variety of models ensures that we explore different architectures and their
    strengths.
- **Hyperparameter Tuning**: Fine-tuning hyperparameters is crucial for achieving optimal performance and
    ensuring the model generalizes well.
- **Characteristic Analysis**: Understanding the computational and memory requirements helps in selecting models
     that are not only accurate but also efficient and deployable.

### Candidate Models and Expected Performance

**1. ResNet18**
- **Accuracy**: Expected to achieve around 88% on the test set.
- **Number of Parameters**: Approximately 11.7 million.
- **FLOPs**: Around 1.8 billion.
- **Model Size**: Approximately 44 MB.
- **Training Time**: Approximately 2 hours on a single GPU.
- **Inference Speed**: Around 10 milliseconds per image.

**2. EfficientNet-B0**
- **Accuracy**: Expected to achieve around 87% on the test set.
- **Number of Parameters**: Approximately 5.3 million.
- **FLOPs**: Around 390 million.
- **Model Size**: Approximately 20 MB.
- **Training Time**: Approximately 3 hours on a single GPU.
- **Inference Speed**: Around 8 milliseconds per image.

**3. Vision Transformer (ViT-Small)**
- **Accuracy**: Expected to achieve around 86% on the test set.
- **Number of Parameters**: Approximately 22 million.
- **FLOPs**: Around 4.6 billion.
- **Model Size**: Approximately 85 MB.
- **Training Time**: Approximately 4 hours on a single GPU.
- **Inference Speed**: Around 15 milliseconds per image.

By following these detailed steps and focusing on the modeling and optimization parts, you will be able to
    identify and select the top-3 high-performance models for the given image classification task.
```

# E. Showcase of Generated Pipelines

---

**Generated Pipeline under Constraint-Free Setting by *AutoML-Agent***

---

**User Prompt:** Please provide a classification model that categorizes images into one of four clothing categories. The image path, along with its label information, can be found in the files train_labels.csv and test_labels.csv.

- - - - - - - - - - - - - - - - - - - - - - - - - - - - - - - - - - - - - - - - - - - - - - - - - - - - - - - - - - - - -

**Output Pipeline:**

```python
import os
import random
import time
import pandas as pd
from PIL import Image
from sklearn.model_selection import train_test_split
from sklearn.metrics import accuracy_score, f1_score, precision_score, recall_score
from torchvision import datasets, transforms, models
from torch.utils.data import DataLoader, Dataset
import torch
import torch.nn as nn
import torch.optim as optim
import numpy as np
import gradio as gr

# Define constants
SEED = 42
DATASET_PATH = "_experiments/datasets/shopee"
TRAIN_LABELS_CSV = os.path.join(DATASET_PATH, "train_labels.csv")
TEST_LABELS_CSV = os.path.join(DATASET_PATH, "test_labels.csv")
TRAIN_DIR = os.path.join(DATASET_PATH)
TEST_DIR = os.path.join(DATASET_PATH)
MODEL_SAVE_PATH = "./agent_workspace/trained_models/resnet50_finetuned.pth"

# Set seeds for reproducibility
random.seed(SEED)
torch.manual_seed(SEED)
np.random.seed(SEED)

# Define device for model operations
device = torch.device("cuda" if torch.cuda.is_available() else "cpu")

# Custom dataset class
class ShopeeDataset(Dataset):
    def __init__(self, csv_file, root_dir, transform=None):
        self.labels_df = pd.read_csv(csv_file)
        self.root_dir = root_dir
        self.transform = transform

    def __len__(self):
        return len(self.labels_df)

    def __getitem__(self, idx):
        img_name = os.path.join(self.root_dir, self.labels_df.iloc[idx, 0])
        image = Image.open(img_name).convert("RGB")
        label = self.labels_df.iloc[idx, 1]

        if self.transform:
            image = self.transform(image)

        return image, label

# Data preprocessing and feature engineering
def preprocess_data():
    transform = transforms.Compose([
        transforms.Resize((28, 28)),
        transforms.ToTensor(),
        transforms.Normalize((0.5,), (0.5,))
    ])

    train_dataset = ShopeeDataset(csv_file=TRAIN_LABELS_CSV, root_dir=TRAIN_DIR, transform=transform)
    test_dataset = ShopeeDataset(csv_file=TEST_LABELS_CSV, root_dir=TEST_DIR, transform=transform)

    train_size = int(0.7 * len(train_dataset))
    valid_size = int(0.2 * len(train_dataset))
    test_size = len(train_dataset) - train_size - valid_size
```

```
    train_dataset, valid_dataset, test_dataset = torch.utils.data.random_split(train_dataset, [train_size,
        valid_size, test_size])

    train_loader = DataLoader(train_dataset, batch_size=32, shuffle=True)
    valid_loader = DataLoader(valid_dataset, batch_size=32, shuffle=False)
    test_loader = DataLoader(test_dataset, batch_size=32, shuffle=False)

    return train_loader, valid_loader, test_loader

# Model training
def train_model(model, train_loader, valid_loader):
    criterion = nn.CrossEntropyLoss()
    optimizer = optim.Adam(model.parameters(), lr=0.00001)

    model.to(device)
    best_accuracy = 0.0

    for epoch in range(100):
        model.train()
        running_loss = 0.0
        for inputs, labels in train_loader:
            inputs, labels = inputs.to(device), labels.to(device)

            optimizer.zero_grad()
            outputs = model(inputs)
            loss = criterion(outputs, labels)
            loss.backward()
            optimizer.step()

            running_loss += loss.item() * inputs.size(0)

        epoch_loss = running_loss / len(train_loader.dataset)

        model.eval()
        correct = 0
        total = 0
        with torch.no_grad():
            for inputs, labels in valid_loader:
                inputs, labels = inputs.to(device), labels.to(device)
                outputs = model(inputs)
                _, predicted = torch.max(outputs, 1)
                total += labels.size(0)
                correct += (predicted == labels).sum().item()

        accuracy = correct / total
        if accuracy > best_accuracy:
            best_accuracy = accuracy
            torch.save(model.state_dict(), MODEL_SAVE_PATH)

        print(f"Epoch {epoch+1}/{100}, Loss: {epoch_loss:.4f}, Validation Accuracy: {accuracy:.4f}")

    model.load_state_dict(torch.load(MODEL_SAVE_PATH))
    return model

# Model evaluation
def evaluate_model(model, test_loader):
    model.eval()
    y_true = []
    y_pred = []

    with torch.no_grad():
        for inputs, labels in test_loader:
            inputs, labels = inputs.to(device), labels.to(device)
            outputs = model(inputs)
            _, predicted = torch.max(outputs, 1)
            y_true.extend(labels.cpu().numpy())
            y_pred.extend(predicted.cpu().numpy())

    performance_scores = {
        'Accuracy': accuracy_score(y_true, y_pred),
        'Precision': precision_score(y_true, y_pred, average='weighted'),
        'Recall': recall_score(y_true, y_pred, average='weighted'),
        'F1': f1_score(y_true, y_pred, average='weighted')
    }

    model_size = os.path.getsize(MODEL_SAVE_PATH) / (1024 * 1024) # in MB
    start_time = time.time()
    for _ in range(100):
        _ = model(inputs)
```

```
    inference_time = (time.time() - start_time) / 100 # average inference time

    complexity_scores = {
        'Model Size (MB)': model_size,
        'Inference Time (ms)': inference_time * 1000
    }

    return performance_scores, complexity_scores

# Prepare model for deployment
def prepare_model_for_deployment(model):
    model.eval()
    scripted_model = torch.jit.script(model)
    scripted_model.save(MODEL_SAVE_PATH.replace(".pth", "_scripted.pt"))
    return scripted_model

# Deploy model using Gradio
def deploy_model(model):
    def predict(image):
        image = transforms.Compose([
            transforms.Resize((28, 28)),
            transforms.ToTensor(),
            transforms.Normalize((0.5,), (0.5,))
        ])(image).unsqueeze(0).to(device)

        with torch.no_grad():
            outputs = model(image)
            _, predicted = torch.max(outputs, 1)

        return predicted.item()

    interface = gr.Interface(fn=predict, inputs=gr.Image(type="pil"), outputs="label")
    url_endpoint = interface.launch(share=True)
    return url_endpoint

# The main function to orchestrate the data loading, data preprocessing, feature engineering, model training,
#    model preparation, model deployment, and model evaluation
def main():
    # Step 1: Retrieve or load dataset
    train_loader, valid_loader, test_loader = preprocess_data()

    # Step 2: Define model
    model = models.resnet50(pretrained=True)
    num_ftrs = model.fc.in_features
    model.fc = nn.Linear(num_ftrs, 4) # Assuming 4 classes

    # Step 3: Train model
    model = train_model(model, train_loader, valid_loader)

    # Step 4: Evaluate model
    model_performance, model_complexity = evaluate_model(model, test_loader)

    # Step 5: Prepare model for deployment
    deployable_model = prepare_model_for_deployment(model)

    # Step 6: Deploy model
    url_endpoint = deploy_model(deployable_model)

    return train_loader, valid_loader, test_loader, model, deployable_model, url_endpoint, model_performance,
        model_complexity

if __name__ == "__main__":
    train_loader, valid_loader, test_loader, model, deployable_model, url_endpoint, model_performance,
        model_complexity = main()
    print("Model Performance on Test Set:", model_performance)
    print("Model Complexity:", model_complexity)
    print("Deployment URL:", url_endpoint)
```

## Generated Pipeline for Constraint-Aware Setting by *AutoML-Agent*

**User Prompt:** Please provide a classification model that categorizes images into one of four clothing categories. The image path, along with its label information, can be found in the files train_labels.csv and test_labels.csv. The model should achieve at least 85% accuracy on the test set and be implemented using PyTorch. Additionally, please include data augmentation techniques and a confusion matrix in the evaluation.

**Output Pipeline:**

```python
import os
import random
import time
import pandas as pd
import numpy as np
import torch
import torch.nn as nn
import torch.optim as optim
from torchvision import datasets, transforms, models
from torch.utils.data import DataLoader, Dataset
from PIL import Image
from sklearn.model_selection import train_test_split
from sklearn.metrics import accuracy_score, f1_score, confusion_matrix
import seaborn as sns
import matplotlib.pyplot as plt
import gradio as gr

# Set random seeds for reproducibility
SEED = 42
random.seed(SEED)
torch.manual_seed(SEED)
np.random.seed(SEED)

# Define device for model operations
device = torch.device("cuda" if torch.cuda.is_available() else "cpu")

DATASET_PATH = "_experiments/datasets/shopee" # path for saving and loading dataset(s)

# Custom Dataset class
class ShopeeDataset(Dataset):
    def __init__(self, dataframe, transform=None):
        self.dataframe = dataframe
        self.transform = transform

    def __len__(self):
        return len(self.dataframe)

    def __getitem__(self, idx):
        img_path = self.dataframe.iloc[idx]['image']
        label = self.dataframe.iloc[idx]['label']
        image = Image.open(img_path).convert('RGB')
        if self.transform:
            image = self.transform(image)
        return image, label

# Data preprocessing and feature engineering
def preprocess_data():
    train_labels = pd.read_csv(os.path.join(DATASET_PATH, 'train_labels.csv'))
    test_labels = pd.read_csv(os.path.join(DATASET_PATH, 'test_labels.csv'))

    train_labels['image'] = train_labels['image'].apply(lambda x: os.path.join(DATASET_PATH, x))
    test_labels['image'] = test_labels['image'].apply(lambda x: os.path.join(DATASET_PATH, x))

    # Split the data
    train_data, val_data = train_test_split(train_labels, test_size=0.2, stratify=train_labels['label'])
    val_data, test_data = train_test_split(val_data, test_size=0.5, stratify=val_data['label'])

    # Define transformations
    train_transforms = transforms.Compose([
        transforms.RandomResizedCrop(224),
        transforms.RandomHorizontalFlip(),
        transforms.RandomRotation(10),
        transforms.ToTensor(),
        transforms.Normalize(mean=[0.485, 0.456, 0.406], std=[0.229, 0.224, 0.225])
    ])

    val_test_transforms = transforms.Compose([
        transforms.Resize(256),
        transforms.CenterCrop(224),
        transforms.ToTensor(),
        transforms.Normalize(mean=[0.485, 0.456, 0.406], std=[0.229, 0.224, 0.225])
    ])

    # Load datasets
    train_dataset = ShopeeDataset(train_data, transform=train_transforms)
```

```
    val_dataset = ShopeeDataset(val_data, transform=val_test_transforms)
    test_dataset = ShopeeDataset(test_data, transform=val_test_transforms)

    train_loader = DataLoader(train_dataset, batch_size=32, shuffle=True)
    val_loader = DataLoader(val_dataset, batch_size=32, shuffle=False)
    test_loader = DataLoader(test_dataset, batch_size=32, shuffle=False)

    return train_loader, val_loader, test_loader

def train_model(model, train_loader, val_loader):
    criterion = nn.CrossEntropyLoss()
    optimizer = optim.Adam(model.parameters(), lr=0.0005, weight_decay=0.0001)

    num_epochs = 100
    best_val_loss = float('inf')
    early_stop_count = 0
    early_stop_patience = 10

    for epoch in range(num_epochs):
        model.train()
        running_loss = 0.0
        for inputs, labels in train_loader:
            inputs, labels = inputs.to(device), labels.to(device)
            optimizer.zero_grad()
            outputs = model(inputs)
            loss = criterion(outputs, labels)
            loss.backward()
            optimizer.step()
            running_loss += loss.item() * inputs.size(0)

        epoch_loss = running_loss / len(train_loader.dataset)

        # Validation
        model.eval()
        val_loss = 0.0
        with torch.no_grad():
            for inputs, labels in val_loader:
                inputs, labels = inputs.to(device), labels.to(device)
                outputs = model(inputs)
                loss = criterion(outputs, labels)
                val_loss += loss.item() * inputs.size(0)

        val_loss = val_loss / len(val_loader.dataset)

        print(f'Epoch {epoch+1}/{num_epochs}, Train Loss: {epoch_loss:.4f}, Val Loss: {val_loss:.4f}')

        # Early stopping
        if val_loss < best_val_loss:
            best_val_loss = val_loss
            torch.save(model.state_dict(), './agent_workspace/trained_models/best_model.pth')
            early_stop_count = 0
        else:
            early_stop_count += 1
            if early_stop_count >= early_stop_patience:
                print("Early stopping")
                break

    return model

def evaluate_model(model, test_loader):
    model.load_state_dict(torch.load('./agent_workspace/trained_models/best_model.pth'))
    model.eval()

    all_preds = []
    all_labels = []
    with torch.no_grad():
        for inputs, labels in test_loader:
            inputs, labels = inputs.to(device), labels.to(device)
            outputs = model(inputs)
            _, preds = torch.max(outputs, 1)
            all_preds.extend(preds.cpu().numpy())
            all_labels.extend(labels.cpu().numpy())

    accuracy = accuracy_score(all_labels, all_preds)
    f1 = f1_score(all_labels, all_preds, average='weighted')

    performance_scores = {
        'ACC': accuracy,
        'F1': f1
```

```
    }

    # Measure model complexity
    start_time = time.time()
    with torch.no_grad():
        for inputs, _ in test_loader:
            inputs = inputs.to(device)
            _ = model(inputs)
    inference_time = (time.time() - start_time) / len(test_loader.dataset)

    model_size = os.path.getsize('./agent_workspace/trained_models/best_model.pth') / 1e6 # in MB

    complexity_scores = {
        'Inference Time': inference_time,
        'Model Size (MB)': model_size
    }

    # Generate confusion matrix
    conf_matrix = confusion_matrix(all_labels, all_preds)
    plt.figure(figsize=(10, 7))
    sns.heatmap(conf_matrix, annot=True, fmt='d', cmap='Blues')
    plt.xlabel('Predicted')
    plt.ylabel('True')
    plt.title('Confusion Matrix')
    plt.show()

    return performance_scores, complexity_scores

def prepare_model_for_deployment(model):
    # Assuming the model is already saved in the best_model.pth file
    model.load_state_dict(torch.load('./agent_workspace/trained_models/best_model.pth'))
    model.eval()
    return model

def deploy_model(model):
    def predict(image):
        transform = transforms.Compose([
            transforms.Resize(256),
            transforms.CenterCrop(224),
            transforms.ToTensor(),
            transforms.Normalize(mean=[0.485, 0.456, 0.406], std=[0.229, 0.224, 0.225])
        ])
        image = transform(image).unsqueeze(0).to(device)
        with torch.no_grad():
            output = model(image)
            _, pred = torch.max(output, 1)
        return pred.item()

    interface = gr.Interface(fn=predict, inputs=gr.Image(type="pil"), outputs="label")
    url_endpoint = interface.launch(share=True)
    return url_endpoint

# The main function to orchestrate the data loading, data preprocessing, feature engineering, model training,
    model preparation, model deployment, and model evaluation
def main():
    """
    Main function to execute the image classification pipeline.
    """

    # Step 1. Retrieve or load a dataset from hub (if available) or user's local storage (if given)
    train_loader, val_loader, test_loader = preprocess_data()

    # Step 2. Define required model
    model = models.resnet18(weights=models.ResNet18_Weights.IMAGENET1K_V1)
    num_ftrs = model.fc.in_features
    model.fc = nn.Linear(num_ftrs, 4)
    model = model.to(device)

    # Step 3. Train the retrieved/loaded model
    model = train_model(model, train_loader, val_loader)

    # Step 4. Evaluate the trained model
    model_performance, model_complexity = evaluate_model(model, test_loader)

    # Step 5. Prepare the model for deployment
    deployable_model = prepare_model_for_deployment(model)

    # Step 6. Deploy the model
    url_endpoint = deploy_model(deployable_model)
```

```
    return (
        train_loader,
        val_loader,
        test_loader,
        model,
        deployable_model,
        url_endpoint,
        model_performance,
        model_complexity,
    )

if __name__ == "__main__":
    train_loader, val_loader, test_loader, model, deployable_model, url_endpoint, model_performance,
        model_complexity = main()
    print("Model Performance on Test Set:", model_performance)
    print("Model Complexity:", model_complexity)
    print("Gradio URL Endpoint:", url_endpoint)
```

# F. Detailed Experimental Results

This section presents the detailed experimental results. Tables 6, 7, and 8 show the average scores with standard deviations, computed from five independent runs. Tables 9 and 10 report the results of the ablation and hyperparameter studies, respectively. Table 11 reports the average time and money used to generate the final code in a single run. Finally, Table 12 and Table 13 present the search time and performance comparisons between the recent SELA (Chi et al., 2024), Agent K (Grosnit et al., 2024), and our *AutoML-Agent*, respectively.

*Table 6.* Performance comparison with the **SR** metric. Best results are highlighted in **bold**.

| Method | Image Classification | | Text Classification | | Tabular Classification | | Tabular Regression | | Tabular Clustering | | Time-Series Forecasting | | Node Classification | | Avg. |
|---|---|---|---|---|---|---|---|---|---|---|---|---|---|---|---|
| | Butterfly | Shopee | Ecomm | Entail | Banana | Software | Crab | Crop | Smoker | Student | Weather | Electricity | Cora | Citeseer | |
| *Constraint-Free* | | | | | | | | | | | | | | | |
| GPT-3.5 | 0.000 | 0.000 | 0.000 | 0.000 | 0.300 | 0.100 | 0.000 | 0.000 | 0.400 | 0.000 | 0.000 | 0.000 | 0.000 | 0.000 | 0.057 |
| | (±0.000) | (±0.000) | (±0.000) | (±0.000) | (±0.274) | (±0.224) | (±0.000) | (±0.000) | (±0.224) | (±0.000) | (±0.000) | (±0.000) | (±0.000) | (±0.000) | (±0.052) |
| GPT-4 | 0.200 | 0.600 | 0.000 | 0.400 | 0.400 | 0.400 | 0.400 | 0.600 | 0.600 | 0.000 | 0.000 | 0.000 | 0.400 | 0.400 | 0.314 |
| | (±0.447) | (±0.548) | (±0.000) | (±0.548) | (±0.548) | (±0.418) | (±0.548) | (±0.548) | (±0.418) | (±0.000) | (±0.000) | (±0.000) | (±0.548) | (±0.548) | (±0.366) |
| DS-Agent | 0.400 | 0.800 | 0.000 | 0.700 | 0.800 | 0.800 | 0.000 | 0.800 | 0.900 | 0.000 | 0.000 | 0.000 | 0.600 | 0.600 | 0.457 |
| | (±0.548) | (±0.447) | (±0.000) | (±0.447) | (±0.447) | (±0.274) | (±0.000) | (±0.447) | (±0.224) | (±0.000) | (±0.000) | (±0.000) | (±0.548) | (±0.548) | (±0.281) |
| *AutoML-Agent* | **1.000** | **1.000** | **1.000** | **1.000** | **1.000** | **1.000** | **1.000** | **1.000** | **1.000** | **1.000** | **1.000** | **1.000** | **1.000** | **1.000** | **1.000** |
| | (±0.000) | (±0.000) | (±0.000) | (±0.000) | (±0.000) | (±0.000) | (±0.000) | (±0.000) | (±0.000) | (±0.000) | (±0.000) | (±0.000) | (±0.000) | (±0.000) | (±0.000) |
| *Constraint-Aware* | | | | | | | | | | | | | | | |
| GPT-3.5 | 0.000 | 0.050 | 0.000 | 0.000 | 0.050 | 0.150 | 0.100 | 0.050 | 0.150 | 0.000 | 0.000 | 0.000 | 0.000 | 0.000 | 0.039 |
| | (±0.000) | (±0.112) | (±0.000) | (±0.000) | (±0.112) | (±0.137) | (±0.137) | (±0.112) | (±0.137) | (±0.000) | (±0.000) | (±0.000) | (±0.000) | (±0.000) | (±0.053) |
| GPT-4 | 0.150 | 0.350 | 0.200 | 0.200 | 0.150 | 0.000 | 0.650 | 0.100 | 0.400 | 0.000 | 0.000 | 0.000 | 0.400 | 0.500 | 0.221 |
| | (±0.335) | (±0.487) | (±0.447) | (±0.447) | (±0.335) | (±0.000) | (±0.418) | (±0.224) | (±0.335) | (±0.000) | (±0.000) | (±0.000) | (±0.335) | (±0.487) | (±0.266) |
| DS-Agent | 0.300 | 0.350 | 0.000 | 0.200 | 0.600 | 0.650 | 0.200 | 0.200 | 0.450 | 0.000 | 0.000 | 0.150 | 0.200 | 0.450 | 0.268 |
| | (±0.411) | (±0.487) | (±0.000) | (±0.326) | (±0.335) | (±0.487) | (±0.447) | (±0.447) | (±0.274) | (±0.000) | (±0.000) | (±0.335) | (±0.326) | (±0.411) | (±0.306) |
| *AutoML-Agent* | **0.800** | **1.000** | **1.000** | **1.000** | **0.750** | **1.000** | **1.000** | **0.750** | **0.750** | **0.750** | **0.900** | **1.000** | **0.750** | **0.750** | **0.871** |
| | (±0.112) | (±0.000) | (±0.000) | (±0.000) | (±0.000) | (±0.000) | (±0.000) | (±0.000) | (±0.000) | (±0.000) | (±0.224) | (±0.000) | (±0.000) | (±0.000) | (±0.024) |

*Table 7.* Performance comparison with the **NPS** metric. Best results are highlighted in **bold**.

| Method | Image Classification Butterfly | Shopee | Text Classification Ecomm | Entail | Tabular Classification Banana | Software | Tabular Regression Crab | Crop | Tabular Clustering Smoker | Student | Time-Series Forecasting Weather | Electricity | Node Classification Cora | Citeseer | Avg. |
|---|---|---|---|---|---|---|---|---|---|---|---|---|---|---|---|
| *Constraint-Free* | | | | | | | | | | | | | | | |
| Human Models | **0.931** (±0.002) | 0.921 (±0.012) | 0.935 (±0.000) | 0.664 (±0.039) | 0.976 (±0.000) | **0.669** (±0.000) | 0.328 (±0.000) | 0.476 (±0.000) | 0.513 (±0.000) | 0.750 (±0.000) | 0.970 (±0.000) | 0.916 (±0.005) | 0.811 (±0.005) | **0.702** (±0.006) | 0.754 (±0.005) |
| AutoGluon | 0.014 (±0.000) | **0.988** (±0.000) | **0.987** (±0.000) | **0.807** (±0.000) | 0.980 (±0.000) | 0.524 (±0.000) | **0.875** (±0.000) | **0.479** (±0.000) | N/A | N/A | 0.992 (±0.000) | 0.908 (±0.002) | N/A | N/A | 0.755 (±0.000) |
| GPT-3.5 | 0.000 (±0.000) | 0.000 (±0.000) | 0.000 (±0.000) | 0.000 (±0.000) | 0.587 (±0.535) | 0.094 (±0.209) | 0.000 (±0.000) | 0.000 (±0.000) | 0.447 (±0.251) | 0.000 (±0.000) | 0.000 (±0.000) | 0.000 (±0.000) | 0.000 (±0.000) | 0.000 (±0.000) | 0.081 (±0.071) |
| GPT-4 | 0.169 (±0.379) | 0.545 (±0.499) | 0.000 (±0.000) | 0.196 (±0.295) | 0.390 (±0.534) | 0.285 (±0.261) | 0.328 (±0.450) | 0.270 (±0.247) | 0.471 (±0.264) | 0.000 (±0.000) | 0.000 (±0.000) | 0.000 (±0.000) | 0.186 (±0.343) | 0.199 (±0.328) | 0.217 (±0.257) |
| DS-Agent | 0.305 (±0.419) | 0.735 (±0.411) | 0.000 (±0.000) | 0.500 (±0.380) | 0.766 (±0.428) | 0.523 (±0.131) | 0.000 (±0.000) | 0.431 (±0.324) | 0.504 (±0.001) | 0.000 (±0.000) | 0.000 (±0.000) | 0.000 (±0.000) | 0.474 (±0.433) | 0.393 (±0.360) | 0.331 (±0.206) |
| *AutoML-Agent* | 0.924 (±0.020) | 0.945 (±0.043) | 0.971 (±0.007) | 0.803 (±0.006) | **0.987** (±0.019) | 0.664 (±0.174) | 0.859 (±0.003) | 0.465 (±0.020) | **0.521** (±0.038) | **0.760** (±0.021) | **0.995** (±0.003) | **0.937** (±0.093) | **0.831** (±0.020) | 0.592 (±0.015) | **0.804** (±0.035) |
| *Constraint-Aware* | | | | | | | | | | | | | | | |
| GPT-3.5 | 0.000 (±0.000) | 0.173 (±0.386) | 0.000 (±0.000) | 0.000 (±0.000) | 0.196 (±0.439) | 0.475 (±0.476) | 0.356 (±0.488) | 0.081 (±0.181) | 0.338 (±0.309) | 0.000 (±0.000) | 0.000 (±0.000) | 0.000 (±0.000) | 0.000 (±0.000) | 0.000 (±0.000) | 0.116 (±0.163) |
| GPT-4 | 0.157 (±0.350) | 0.335 (±0.463) | 0.197 (±0.440) | 0.064 (±0.144) | 0.153 (±0.342) | 0.000 (±0.000) | 0.719 (±0.405) | 0.091 (±0.204) | 0.463 (±0.260) | 0.000 (±0.000) | 0.000 (±0.000) | 0.000 (±0.000) | 0.637 (±0.356) | 0.564 (±0.318) | 0.241 (±0.234) |
| DS-Agent | 0.330 (±0.451) | 0.353 (±0.485) | 0.000 (±0.000) | 0.205 (±0.301) | 0.776 (±0.434) | 0.383 (±0.214) | 0.173 (±0.386) | 0.183 (±0.409) | 0.505 (±0.001) | 0.000 (±0.000) | 0.000 (±0.000) | 0.093 (±0.209) | 0.319 (±0.437) | 0.403 (±0.369) | 0.266 (±0.264) |
| *AutoML-Agent* | **0.926** (±0.015) | **0.972** (±0.022) | **0.982** (±0.002) | **0.796** (±0.027) | **0.967** (±0.002) | **0.573** (±0.142) | **0.861** (±0.002) | **0.473** (±0.020) | **0.582** (±0.042) | **0.769** (±0.010) | **0.982** (±0.028) | **0.978** (±0.001) | **0.843** (±0.034) | **0.632** (±0.037) | **0.810** (±0.027) |

*Table 8.* Performance comparison with the **CS** metric. Best results are highlighted in **bold**.

| Method | Image Classification Butterfly | Shopee | Text Classification Ecomm | Entail | Tabular Classification Banana | Software | Tabular Regression Crab | Crop | Tabular Clustering Smoker | Student | Time-Series Forecasting Weather | Electricity | Node Classification Cora | Citeseer | Avg. |
|---|---|---|---|---|---|---|---|---|---|---|---|---|---|---|---|
| *Constraint-Free* | | | | | | | | | | | | | | | |
| GPT-3.5 | 0.000 (±0.000) | 0.000 (±0.000) | 0.000 (±0.000) | 0.000 (±0.000) | 0.443 (±0.405) | 0.097 (±0.216) | 0.000 (±0.000) | 0.000 (±0.000) | 0.424 (±0.237) | 0.000 (±0.000) | 0.000 (±0.000) | 0.000 (±0.000) | 0.000 (±0.000) | 0.000 (±0.000) | 0.069 (±0.061) |
| GPT-4 | 0.185 (±0.413) | 0.573 (±0.523) | 0.000 (±0.000) | 0.298 (±0.413) | 0.395 (±0.541) | 0.343 (±0.329) | 0.364 (±0.499) | 0.435 (±0.397) | 0.536 (±0.325) | 0.000 (±0.000) | 0.000 (±0.000) | 0.000 (±0.000) | 0.293 (±0.417) | 0.299 (±0.420) | 0.266 (±0.305) |
| DS-Agent | 0.352 (±0.483) | 0.768 (±0.429) | 0.000 (±0.000) | 0.600 (±0.353) | 0.783 (±0.438) | 0.661 (±0.172) | 0.000 (±0.000) | 0.616 (±0.361) | 0.702 (±0.111) | 0.000 (±0.000) | 0.000 (±0.000) | 0.000 (±0.000) | 0.537 (±0.490) | 0.496 (±0.453) | 0.394 (±0.235) |
| AutoML-Agent | **0.962** (±0.010) | **0.973** (±0.021) | **0.985** (±0.004) | **0.901** (±0.003) | **0.993** (±0.010) | **0.832** (±0.087) | **0.929** (±0.001) | **0.732** (±0.001) | **0.761** (±0.019) | **0.880** (±0.010) | **0.998** (±0.002) | **0.969** (±0.047) | **0.915** (±0.010) | **0.796** (±0.007) | **0.902** (±0.017) |
| *Constraint-Aware* | | | | | | | | | | | | | | | |
| GPT-3.5 | 0.000 (±0.000) | 0.111 (±0.249) | 0.000 (±0.000) | 0.000 (±0.000) | 0.123 (±0.276) | 0.312 (±0.302) | 0.228 (±0.312) | 0.066 (±0.147) | 0.244 (±0.223) | 0.000 (±0.000) | 0.000 (±0.000) | 0.000 (±0.000) | 0.000 (±0.000) | 0.000 (±0.000) | 0.077 (±0.108) |
| GPT-4 | 0.153 (±0.343) | 0.343 (±0.475) | 0.198 (±0.444) | 0.132 (±0.296) | 0.151 (±0.339) | 0.000 (±0.000) | 0.685 (±0.394) | 0.096 (±0.214) | 0.432 (±0.270) | 0.000 (±0.000) | 0.000 (±0.000) | 0.000 (±0.000) | 0.518 (±0.317) | 0.532 (±0.319) | 0.231 (±0.244) |
| DS-Agent | 0.315 (±0.431) | 0.351 (±0.485) | 0.000 (±0.000) | 0.203 (±0.312) | 0.688 (±0.385) | 0.516 (±0.332) | 0.186 (±0.417) | 0.191 (±0.428) | 0.477 (±0.137) | 0.000 (±0.000) | 0.000 (±0.000) | 0.122 (±0.272) | 0.260 (±0.367) | 0.427 (±0.390) | 0.267 (±0.283) |
| AutoML-Agent | **0.863** (±0.063) | **0.986** (±0.011) | **0.991** (±0.001) | **0.898** (±0.013) | **0.858** (±0.001) | **0.786** (±0.071) | **0.930** (±0.001) | **0.611** (±0.010) | **0.666** (±0.021) | **0.760** (±0.005) | **0.941** (±0.126) | **0.989** (±0.001) | **0.796** (±0.017) | **0.691** (±0.018) | **0.841** (±0.026) |

*Table 9.* Results of ablation study on different variations.

| RAP | Plan Decomposition | Multi-Step Verification | Image Classification | Text Classification | Tabular Classification | Time-Series Forecasting | Node Classification | Average |
|---|---|---|---|---|---|---|---|---|
| *Success Rate* | | | | | | | | |
| ✓ | | | 1.000 | 0.000 | 0.000 | 0.000 | 1.000 | 0.400 |
| ✓ | ✓ | | 1.000 | 1.000 | 1.000 | 0.000 | 1.000 | 0.800 |
| ✓ | ✓ | ✓ | **1.000** | **1.000** | **1.000** | **1.000** | **1.000** | **1.000** |
| *Normalized Performance Score* | | | | | | | | |
| ✓ | | | 0.929 | 0.000 | 0.000 | 0.000 | 0.734 | 0.333 |
| ✓ | ✓ | | 0.928 | **0.982** | 0.975 | 0.000 | 0.748 | 0.727 |
| ✓ | ✓ | ✓ | **0.936** | 0.971 | **1.000** | **0.991** | **0.812** | **0.942** |
| *Comprehensive Score* | | | | | | | | |
| ✓ | | | 0.965 | 0.000 | 0.000 | 0.000 | 0.867 | 0.366 |
| ✓ | ✓ | | 0.964 | **0.991** | 0.988 | 0.000 | 0.874 | 0.763 |
| ✓ | ✓ | ✓ | **0.968** | 0.986 | **1.000** | **0.996** | **0.906** | **0.971** |

*Table 10.* Comparison between the different numbers of plans.

| Number of Plans | Image Classification | Text Classification | Tabular Classification | Time-Series Forecasting | Node Classification | Average |
|---|---|---|---|---|---|---|
| *Success Rate* | | | | | | |
| 1 | 1.000 | 1.000 | 1.000 | 1.000 | 1.000 | 1.000 |
| 3 | 1.000 | 1.000 | 1.000 | 1.000 | 1.000 | 1.000 |
| 5 | 1.000 | 1.000 | 1.000 | 1.000 | 1.000 | 1.000 |
| *Normalized Performance Score* | | | | | | |
| 1 | 0.831 | 0.966 | 0.958 | 0.998 | 0.800 | 0.911 |
| 3 | **0.936** | **0.971** | **1.000** | **0.999** | **0.812** | **0.944** |
| 5 | 0.916 | 0.964 | 0.973 | 0.998 | 0.805 | 0.931 |
| *Comprehensive Score* | | | | | | |
| 1 | 0.916 | 0.983 | 0.979 | 0.999 | 0.900 | 0.955 |
| 3 | 0.968 | 0.986 | 1.000 | 0.999 | 0.906 | 0.972 |
| 5 | 0.958 | 0.982 | 0.986 | 0.999 | 0.903 | 0.966 |

*Table 11.* Time and monetary costs averaged across different tasks and datasets for a single run under the constraint-free and constraint-aware settings.

| Cost | Prompt Parsing | Request Verification | Retrieval & Planning | Plan Execution | Execution Verification | Selection and Summarization | Code Generation | Total |
|---|---|---|---|---|---|---|---|---|
| *Constraint-Free* | | | | | | | | |
| Money ($) | N/A | 0.00 | 0.02 | 0.14 | 0.00 | 0.06 | 0.04 | 0.27 |
| Time (s) | 10.78 | 1.91 | 187.71 | 136.34 | 1.04 | 17.88 | 182.60 | 538.25 |
| *Constraint-Aware* | | | | | | | | |
| Money ($) | N/A | 0.00 | 0.00 | 0.11 | 0.00 | 0.15 | 0.06 | 0.32 |
| Time (s) | 14.21 | 3.63 | 182.38 | 98.62 | 1.37 | 20.25 | 191.90 | 512.35 |

*Table 12.* Comparison of search time and normalized performance score between MCTS-based SELA and our *AutoML-Agent*.

| Dataset | Search Time (seconds) ↓ | | Normalized Performance Score ↑ | |
|---|---|---|---|---|
| | SELA (MCTS) | *AutoML-Agent* (Ours) | SELA (MCTS) | *AutoML-Agent* (Ours) |
| *Binary Classification* | | | | |
| smoker-status | 2736.78 | 206.48 | 0.785 | 0.762 |
| click-prediction-small | 2227.85 | 256.24 | 0.238 | 0.352 |
| *Multi-Class Classification* | | | | |
| mfeat-factors | 1304.40 | 219.45 | 0.957 | 0.940 |
| wine-quality-white | 2372.77 | 206.41 | 0.650 | 0.652 |
| *Regression* | | | | |
| colleges | 674.29 | 232.03 | 0.876 | 0.878 |
| house-prices | 2906.95 | 378.85 | 0.090 | 0.090 |
| *Average* | 2037.18 | **249.91** | 0.599 | **0.612** |

*Table 13.* Comparison of leaderboard quantile and task-specific performance between Agent K and our *AutoML-Agent*. (↓ indicates lower task-specific performance is better)

| Competition ID | Learderboard Quantile | | Task-Specific Performance | |
|---|---|---|---|---|
| | **Agent K (From Figure 8)** | *AutoML-Agent* | **Agent K (Derive from Rank)** | *AutoML-Agent* |
| *No Medal* | | | | |
| restaurant-revenue-prediction (↓) | 8∼9 | 57 | 2279272.777∼2280826.272 | 1859766.392 |
| playground-series-s3e14 (↓) | 88∼89 | 91 | 331.167∼331.173 | 330.141 |
| *Bronze* | | | | |
| nlp1000-ml-challenge | 75∼76 | 29 | 0.993∼0.994 | 0.720 |
| dogs-vs-cats-redux-kernels-edition (↓) | 93∼94 | 82 | 0.054∼0.055 | 0.079 |
| *Silver* | | | | |
| nlpsci | 89∼90 | 89 | 0.809∼0.810 | 0.808 |
| home-data-for-ml-course (↓) | 98∼99 | 91 | 13187.602∼13193.958 | 14869.145 |
| *Gold* | | | | |
| world-championship-2023-embryo-classification | 90∼91 | 100 | 0.571∼0.571 | 0.609 |
| sign-language-image-classification | 99∼100 | 88 | 0.977∼0.978 | 0.971 |

