# OpenReview forum: "AutoML-Agent: A Multi-Agent LLM Framework for Full-Pipeline AutoML"
_ICML.cc/2025/Conference — ICML 2025 poster_

### Official Review · Reviewer_L7kx · 2025-03-14

**Overall Recommendation:** 5

**Summary:**

&nbsp;

The authors introduce a multi-agent LLM framework for full pipeline AutoML. The authors perform an extensive empirical study against relevant baselines, demonstrating that their AutoML-Agent outperforms both frontier models and partial pipeline language agents. The authors' experiments are informative, showcasing that for simpler, tabular tasks, partial pipeline agents remain competitive. The paper is well-written and the authors release an extensive codebase for their agent. As such I recommend acceptance with the potential to increase my score if the points below are addressed.

&nbsp;

## **Post-Rebuttal Update**

&nbsp;

I have upgraded my score to 5 post-rebuttal follow the authors' extensive additional experiments and clarifications during the author-reviewer discussion phase. I am strongly in favor of accepting the paper and wish to express to the AC my interest in championing the paper for consideration as a spotlight/oral.

&nbsp;

**Claims And Evidence:**

&nbsp;

The claims made by the authors are empirical in nature. In terms of reproducibility, the authors provide an anonymous GitHub link to their codebase which is well documented.

&nbsp;

**Essential References Not Discussed:**

&nbsp;

Below I list the references I have cited elsewhere in the review.

&nbsp;

**__REFERENCES__**

&nbsp;

[1] Narayanan, S., Braza, J.D., Griffiths, R.R., Ponnapati, M., Bou, A., Laurent, J., Kabeli, O., Wellawatte, G., Cox, S., Rodriques, S.G. and White, A.D., 2024. [Aviary: training language agents on challenging scientific tasks.](https://arxiv.org/abs/2412.21154) arXiv preprint arXiv:2412.21154.

[2] Grosnit, A., Maraval, A., Doran, J., Paolo, G., Thomas, A., Beevi, R.S.H.N., Gonzalez, J., Khandelwal, K., Iacobacci, I., Benechehab, A. and Cherkaoui, H., 2024. [Large language models orchestrating structured reasoning achieve Kaggle grandmaster level.](https://arxiv.org/abs/2411.03562) arXiv preprint arXiv:2411.03562.

[3] Tang, X., Liu, Y., Cai, Z., Shao, Y., Lu, J., Zhang, Y., Deng, Z., Hu, H., An, K., Huang, R. and Si, S., 2023. [ML-Bench: Evaluating Large Language Models and Agents for Machine Learning Tasks on Repository-Level Code.](https://arxiv.org/abs/2311.09835) arXiv preprint arXiv:2311.09835.

[4] Liu, T., Astorga, N., Seedat, N. and van der Schaar, M., [Large Language Models to Enhance Bayesian Optimization.](https://openreview.net/forum?id=OOxotBmGol) In The Twelfth International Conference on Learning Representations 2024.

[5] Tang, J., Fan, T. and Huang, C., 2025. [AutoAgent: A Fully-Automated and Zero-Code Framework for LLM Agents.](https://arxiv.org/abs/2502.05957) arXiv e-prints, pp.arXiv-2502.

[6] Kon, P.T.J., Liu, J., Ding, Q., Qiu, Y., Yang, Z., Huang, Y., Srinivasa, J., Lee, M., Chowdhury, M. and Chen, A., 2025. [Curie: Toward Rigorous and Automated Scientific Experimentation with AI Agents.](https://arxiv.org/abs/2502.16069) arXiv preprint arXiv:2502.16069.

[7] Fourney, A., Bansal, G., Mozannar, H., Tan, C., Salinas, E., Niedtner, F., Proebsting, G., Bassman, G., Gerrits, J., Alber, J. and Chang, P., 2024. [Magentic-One: A generalist multi-agent system for solving complex tasks.](https://arxiv.org/abs/2411.04468) arXiv preprint arXiv:2411.04468.

[8] Hu, X., Zhao, Z., Wei, S., Chai, Z., Ma, Q., Wang, G., Wang, X., Su, J., Xu, J., Zhu, M. and Cheng, Y., 2024, July. [InfiAgent-DABench: Evaluating Agents on Data Analysis Tasks.](https://proceedings.mlr.press/v235/hu24s.html) In International Conference on Machine Learning (pp. 19544-19572). PMLR.

[9] Lála, J., O'Donoghue, O., Shtedritski, A., Cox, S., Rodriques, S.G. and White, A.D., 2023. [PaperQA: Retrieval-augmented generative agent for scientific research.](https://arxiv.org/abs/2312.07559) arXiv preprint arXiv:2312.07559.

&nbsp;

**Ethical Review Concerns:**

&nbsp;

Not applicable.

&nbsp;

**Experimental Designs Or Analyses:**

&nbsp;

1. What is the sensitivity of AutoML-Agent's performance to the user’s prompt with task description, requirements, and/or constraints? It may be worth designing a sensitivity analysis by having e.g. 3 levels of clarity in the user prompt ranging from vague to precise instructions assuming they pass the request verification stage.

2. What is the sensitivity to the system prompts used by the various agents?

&nbsp;

**Methods And Evaluation Criteria:**

&nbsp;

1. It would be worth discussing the relationship between the current approach and the Orchestrator-Workers paradigm of [7]. It seems as though the Agent Manager functions as an orchestrator.

2. How does AutoML-Agent compare against Agent K [2] on Kaggle-style problems? This seems like it would be a relevant comparison given that the bespoke AutoML baselines in the current paper are chosen based on their performance in Kaggle notebooks.

&nbsp;

**Other Comments Or Suggestions:**

&nbsp;

1. There are some missing capitalizations in the references e.g. "GPT-4", "LLM", "AutoML", "AI".

2. In the introduction, no parentheses around "e.g. natural language and computer vision)

3. The link to the codebase was quite difficult to find (I believe it is only given in Section C of the appendix?). It may be worth moving this to the abstract or introduction.

4. It would be worth running GPT or Claude over the codebase code to write documentation.

5. It would be worth spelling out the acronym RMSLE as root mean square log error when it is first introduced in the main paper.

6. Line 183, typo, there is a space that should be inserted. Same on line 370.

&nbsp;

**Other Strengths And Weaknesses:**

&nbsp;

Overall, I think the paper could be strengthened by performing a sensitivity analysis on the various agent prompts as well as comparing against Agent K [2] on external Kaggle datasets. Otherwise I believe the contribution is still solid.

&nbsp;

**Questions For Authors:**

&nbsp;

1. The prompt agent requires instruction fine-tuning. How is this instruction fine-tuning expected to generalize to new problem classes not considered in the paper? Will the prompt agent LLM need to be instruction-tuned from scratch? Is EvoInstruct expected to be effective in generating instruction, response pairs for all AutoML problems?

2. What is the reason for the gulf in the level of encouragement for the various agents in the system prompt e.g. prompt agent receives the underwhelming, "You are an assistant project manager in the AutoML development team." Whereas data agent receives the hyperbolic, "You are the world’s best data scientist of an automated machine learning project".

3. For the arXiv paper search component in Appendix D, what do the authors think of the potential for a tool such as PaperQA [9] for improving performance in this step?

4. The prompt for Retrieval-Augmented Planning contains in-context examples. How were these generated? Are they contained in the GitHub repo? If so, it is difficult to find them.

5. In Section 3.5, the authors state, "if no dataset is found, we rely on the inherent knowledge of the LLM". Presumably this does not result in a successful outcome if there is no data present?

&nbsp;

**Relation To Broader Scientific Literature:**

&nbsp;

It may be worth referencing [4] in relation to the use of LLMs for HPO. It may also be worth discussing the relationship. Between AutoML-Agent and the works of [1,3,5,6,7,8]. Of particular note is [2] which tackles Kaggle competitions and by its nature will need to perform aspects of AutoML.

&nbsp;

**Theoretical Claims:**

&nbsp;

Not applicable.

&nbsp;

---

> ### Author Rebuttal · Authors · 2025-04-01
>
> We thank the reviewer for the detailed and constructive review and appreciate your recognition of our paper’s strengths. Below, we address your specific comments.
>
> > Relationships to Magentic-One [7] and Agent K [2]
>
> **R1** Thank you for pointing out these valuable concurrent papers that we had previously missed. Although our Agent Manager functions similarly to [7], it performs constraint-aware plan selection and multi-stage verification, making it more task-specific and grounded.
>
> For Agent K [2], the current version is not open-sourced, which limits direct comparison. Nevertheless, we provide a qualitative analysis. Although Agent K also leverages LLMs to orchestrate ML pipelines, it is tailored for Kaggle competition settings and relies on a training-based approach with high search overhead. In contrast, AutoML-Agent introduces a platform-agnostic, RAP and multi-stage verification framework designed for broader AutoML applications beyond Kaggle with a *training-free* search method.
>
> We have clarified these connections and distinctions in Related Work, including those in [1–8].
>
> > Sensitivity to prompts and rationale for system prompts
>
> **R2** While agent-specific prompt design is not the primary focus of this paper, for **user prompts**, our results suggest that AutoML-Agent is minimally sensitive to prompt specificity. In both *constraint-free (**somewhat vague**)* and *constraint-aware (**precise**)* settings, the system achieves high success rates with comparable average NPS (0.804 vs. 0.810), indicating robustness to prompt variation. Clearer prompts generally lead to better outcomes, but the system is designed to handle vague instructions gracefully: the combination of the verification mechanism and RAP allows the system to tolerate a range of prompt qualities. Even when the Prompt Agent interprets vague inputs broadly, execution verification ensures that only valid and high-performing solutions are accepted.
>
> For **system prompts**, please refer to `Reviewer mnKA-R4`.
>
> > Restrictive skeleton Python scripts
>
> **R3** The provided skeleton serves as a default scaffold for typical ML tasks, offering a high-level structure without enforcing task- or data-specific code. It enables agents to complete the template using *theoretically any* specialized modules by following the TODO list. We believe this approach is significantly more flexible than the data-specific templates used in the DS-Agent framework. Typically, the only component that requires adjustment for new tasks is the evaluation metric.
>
> > Instruction fine-tuning in Prompt Agent
>
> **R4** For entirely new problem classes, we could extend the existing model through additional fine-tuning or by providing a few in-context examples. EvoInstruct can be used to generate instruction–response pairs in the new domain to support this adaptation.
>
> To provide some context, instruction fine-tuning was initially necessary to ensure precise parsing of user inputs for interoperability between agents, as this work began before OpenAI supported accurate structured outputs. With recent updates to GPT models, we can now directly use a JSON schema, eliminating the need for fine-tuning in many cases. The Prompt Agent can accurately parse user queries by referencing the schema explicitly. The schema is available at `/prompt_agent/schema.json`.
>
> > Potential use of PaperQA [9]
>
> **R5** We agree that a tool like [9] could improve the literature retrieval component. Incorporating PaperQA-style scientific querying could enhance citation relevance and support more context-aware prompt generation, ultimately leading to better planning decisions. We will acknowledge this direction in the paper and consider it for future iterations.
>
> > In-context examples for Retrieval-Augmented Planning
>
> **R6** The in-context examples (`plan_knowledge`) are **automatically generated on the fly**. They are *extracted*, *summarized*, and *organized* from raw retrieved documents and texts (e.g., search results, arXiv papers, Kaggle notebooks) before being passed to the Agent Manager for planning. We have updated the GitHub repository to include examples at `/example_plans/plan_knowledge.md`.
>
> > "if no dataset is found...".
>
>  **R7** When we mention relying on the LLM’s inherent knowledge, we are referring to a fallback mechanism intended to produce the most plausible output given limited context. However, in the absence of actual data, the pipeline will ultimately fail during final implementation verification due to runtime errors. Our approach assumes that a dataset is either provided (e.g., image classification) or retrievable via search (e.g., node classification). If no dataset is found, the agent will prompt the user to supply one when in interactive mode. We will revise the text to clarify this assumption.
>
> > Writing suggestions
>
> **R8** We have addressed all the mentioned issues, including a table with standardized details (see anonymous GitHub). This will be included in the final version.

---

> > ### Comment · Reviewer_L7kx · 2025-04-05
> >
> > &nbsp;
> >
> > Many thanks to the authors for their rebuttal. I consider the majority of points addressed.
> >
> > &nbsp;
> >
> > 1. **Comparison to Agent K**: Many thanks to the authors for pointing out that the source code for Agent K is not yet open-sourced. As such, the authors cannot be expected to run Agent K. The Kaggle competitions that Agent K is evaluated on however, are present in Figure 8 of the paper. Given that the authors state that AutoML agent is capable of tackling a broader range of AutoML problems relative to Agent K, is there any reason why the authors could not evaluate AutoML agent on the same Kaggle competitions to enable a direct comparison?
> >
> > &nbsp;
> >
> > 2. **Prompt Sensitivity Analysis**: Many thanks for the authors for clarifying that there is an implicit prompt sensitivity analysis present when considering the constraint-free vs. the constraint-aware settings.
> >
> > &nbsp;
> >
> > I remain in favor of accepting the paper. If a direct comparison against Agent K on a subset of the same Kaggle competitions used in the paper (or a convincing justification for why this is not possible could be provided) I will increase my score.
> >
> > &nbsp;

---

> > > ### Author Response · Authors · 2025-04-08
> > >
> > > Thank you very much for acknowledging our responses and giving a chance to further clarifications. We greatly appreciate your support for acceptance.
> > >
> > >
> > > > Comparison to Agent K
> > >
> > > **R1** Thank you for the clarification and for pointing us to Figure 8. During the initial rebuttal phase, our primary focus was on locating the source code to enable a direct and fair comparison using our experimental setup and the NPS metric.
> > >
> > > We would like to clarify that while we can indeed evaluate AutoML-Agent on the same Kaggle competition datasets, a fully direct, apples-to-apples comparison remains infeasible due to the absence of Agent K’s numerical results. However, as suggested by the reviewer, we estimated the quantile and task-specific performance of Agent K from Figure 8 using the Kaggle API.  We report results on *eight* competitions—selected from both the lowest and highest performance quantiles in each category—in the table below, given time constraints.
> > >
> > > | **Competition ID**                                      | **Learderboard Quantile** |              | **Task-Specific Performance** |              |
> > > | ------------------------------------------------------- | ------------------------- | ------------ | ----------------------------- | ------------ |
> > > | (↓ indicates lower task-specific performance is better) | **Agent K** (From Figure 8)   | **AutoML-Agent** | **Agent K**  (Derive from Rank)   | **AutoML-Agent** |
> > > | _No Medal_                                              |                           |              |                               |              |
> > > | restaurant-revenue-prediction (↓)                       | 8~9                       | 57           | 2279272.777~2280826.272       | 1859766.392  |
> > > | playground-series-s3e14 (↓)                             | 88~89                     | 91           | 331.167~331.173               | 330.141      |
> > > | _Bronze_                                                |                           |              |                               |              |
> > > | nlp1000-ml-challenge                                    | 75~76                     | 29           | 0.993~0.994                   | 0.720        |
> > > | dogs-vs-cats-redux-kernels-edition (↓)                  | 93~94                     | 82           | 0.054~0.055                   | 0.079        |
> > > | _Silver_                                                |                           |              |                               |              |
> > > | nlpsci                                                  | 89~90                     | 89           | 0.809~0.810                   | 0.808        |
> > > | home-data-for-ml-course (↓)                             | 98~99                     | 91           | 13187.602~13193.958           | 14869.145    |
> > > | _Gold_                                                  |                           |              |                               |              |
> > > | world-championship-2023-embryo-classification           | 90~91                     | 100          | 0.571~0.571                   | 0.609        |
> > > | sign-language-image-classification                      | 99~100                    | 88           | 0.977~0.978                   | 0.971        |
> > >
> > >
> > > In terms of task-specific downstream performance, **except for the `nlp1000-ml-challenge` and `home-data-for-ml-course` datasets, AutoML-Agent performs comparably or better than Agent K**. It is worth noting that, unlike Agent K, our AutoML-Agent does not leverage iterative feedback directly from leaderboard scores or expert-built, task-specific tools during the search process. We believe that incorporating these kinds of tools could further enhance AutoML-Agent's performance.
> > >
> > > We hope these experimental results, though not a perfect direct comparison, sufficiently address your concern.
> > >
> > > > Prompt Sensitivity Analysis
> > >
> > > **R2** Thank you for acknowledging our response. In addition to the analysis of user prompts, the reviewer may also refer to https://anonymous.4open.science/r/AutoML-Agent/example_plans/prompt_sensitivity.md, which presents results related to the "system prompt" sensitivity.

---

### Official Review · Reviewer_EaoR · 2025-03-14

**Overall Recommendation:** 5

**Summary:**

The paper introduces AutoML-Agent, a novel multi-agent framework leveraging large language models (LLMs) to automate the entire AutoML pipeline, including data retrieval, preprocessing, model selection, hyperparameter optimization, and deployment. The proposed framework employs a retrieval-augmented planning strategy, specialized agents for decomposing and executing tasks, and multi-stage verification processes to enhance reliability and efficiency. Extensive experimental evaluation demonstrates that AutoML-Agent consistently outperforms various baselines across multiple datasets and tasks, achieving higher success rates, superior downstream task performance, and better computational efficiency.

**Claims And Evidence:**

The authors' claims regarding the superiority of AutoML-Agent in handling the full AutoML pipeline and achieving higher efficiency and performance are convincingly supported by robust empirical evidence. Results from experiments on seven diverse tasks and fourteen datasets show clear performance improvements over existing state-of-the-art AutoML frameworks and LLM-based agents, with success rates and performance metrics (accuracy, RMSLE, F1-score) explicitly presented. The ablation studies effectively support the claims regarding the importance of the retrieval-augmented planning and multi-stage verification mechanisms.

**Essential References Not Discussed:**

n/a

**Experimental Designs Or Analyses:**

The experimental designs and analyses are sound and rigorous. The comparison against robust baselines (e.g., AutoGluon, DS-Agent, SELA, GPT-3.5, GPT-4) under both constraint-free and constraint-aware scenarios effectively validates the proposed framework. The detailed ablation studies further substantiate the significance of each component, such as retrieval-augmented planning and multi-stage verification.

**Methods And Evaluation Criteria:**

The proposed methods and evaluation criteria make sense and align with current practices in the field of AutoML. The comprehensive experimental setup, including diverse data modalities (image, text, tabular, graph, and time series data), and clearly defined metrics (e.g., accuracy, RMSLE, F1-score, Rand index) offer a fair and thorough assessment of the framework’s capabilities. The multi-agent architecture, retrieval-augmented planning, and multi-stage verification approaches are logically well-structured and suitable for tackling complex, full-pipeline AutoML challenges.

**Other Comments Or Suggestions:**

- Clarifying the computational and monetary cost implications more explicitly in practical deployments could strengthen the paper’s discussion.

- A more explicit discussion of limitations and potential failure modes beyond provided tasks could further improve the presentation.

**Other Strengths And Weaknesses:**

Potential high computational cost associated with iterative planning and multi-agent verification processes.

Dependence on external APIs and models (GPT-4), which could impact cost.

**Questions For Authors:**

Have you evaluated the robustness of your system when external knowledge sources or APIs provide noisy or outdated information?

Can you elaborate on your framework’s adaptability to ML tasks significantly differing from those tested (e.g., reinforcement learning or recommendation systems)? How might AutoML-Agent be extended to accommodate these areas?

**Relation To Broader Scientific Literature:**

The paper appropriately situates its contributions within the broader literature, effectively referencing previous AutoML frameworks (AutoGluon, AutoML-GPT, SELA), LLM-driven frameworks (DS-Agent, HuggingGPT), and planning methodologies (retrieval-augmented planning, multi-stage verification). Its innovative integration of retrieval-augmented strategies and structured task decomposition clearly addresses existing limitations in the efficiency and generalizability of prior approaches.

**Theoretical Claims:**

The paper does not make specific theoretical claims requiring formal proofs.

---

> ### Author Rebuttal · Authors · 2025-04-01
>
> We thank the reviewer for the insightful and encouraging review. We greatly appreciate your positive assessment of our paper’s novelty, empirical rigor, and technical soundness. Below, we address your thoughtful concerns.
>
> > High computational cost
>
> **R1** We would like to clarify that, apart from the final implementation verification step, AutoML-Agent relies solely on inference throughout the entire search process, including planning. Unlike many prior works, this design ensures efficiency, as runtime does not scale with dataset size or model complexity. Our modular design also enables parallel execution of sub-tasks, further reducing wall-clock time. That said, while computational demands remain low (as discussed in `Reviewer Son5-R1`), using high-performing LLMs for each agent may still incur monetary costs (as noted in `Reviewer mnKA-R3`).
>
>
> > Dependence on external APIs and models
>
> **R2** We agree that relying on a proprietary model like GPT-4o entails cost and reliability implications. However, our framework is model-agnostic—agents can operate with any sufficiently capable LLM. We used GPT-4o to demonstrate state-of-the-art performance, but open-source or local models can be substituted to eliminate external API dependencies, albeit with some performance trade-offs. Given the rapid advancements in LLMs, we expect costs to decrease and become increasingly justified by the resulting performance gains.
>
> > Computational and monetary cost implications in practical deployments
>
> **R3** In practical deployments, the computational and monetary costs of running AutoML-Agent primarily depend on the number of planning iterations (typically 3–5), the complexity of the downstream task (e.g., data modality, task type), and the size and access method of the LLM (e.g., API vs. self-hosted). After pipeline generation, the cost shifts to standard training and inference of the downstream models, which depends on the specific components selected.
>
>
> > Explicit discussion of limitations and potential failure modes
>
> **R4** We have revised the limitations section to include a more explicit discussion of potential failure modes beyond the evaluated tasks. For specific failure cases, please also refer to our response to `Reviewer mnKA-R3`.
>
>
> > Robustness to noisy or outdated information
>
> **R5** Our design does incorporate measures to be robust to bad information. During RAP, the agent cross-verifies information by retrieving from multiple sources against user's requirements before deriving insight knowledge, and the multi-stage verification will catch issues if a retrieved piece of instruction leads to an error (since the code won’t execute or will produce a poor result, prompting a fix). As the inclusion of outdated information is highly unlikely due to retrieval constraints, we focused our evaluation on robustness to noisy information. We simulated two scenarios:
> - Pre-Summary Injection: Injecting unrelated or low-quality examples *before* insight extraction and planning.
> - Post-Summary Injection: Injecting noisy examples *after* insight extraction but before planning, i.e., mixing noisy inputs with the useful insights.
>
> Given the user requirements, these noises are generated by an extra agent (i.e., adversarial agent) prompted to create 'unhelpful' and 'fake' insights that hinder the planning process.
> | **Dataset**            | **AutoML-Agent** | **w/ Pre-Injection** | **w/ Post-Injection** |
> | ---------------------- | ---------------- | -------------------- | --------------------- |
> | smoker-status          | 0.762            | 0.768                | 0.853                 |
> | click-prediction-small | 0.352            | 0.347                | 0.133                 |
> | mfeat-factors          | 0.940            | 0.930                | 0.915                 |
> | wine-quality-white     | 0.652            | 0.615                | 0.670                 |
> | colleges               | 0.878            | 0.658                | 0.896                 |
> | house-prices           | 0.090            | 0.089                | 0.087                 |
> | **_Average_**          | 0.612            | 0.568                | 0.593                 |
>
> Our results show that AutoML-Agent is robust to such noise. Its built-in error correction and multi-stage verification mechanisms significantly mitigate the impact of noisy inputs, ensuring that the final model performance **remains largely unaffected**. Thanks to this suggestion, we observed that noise injection can even lead to better performance in particular cases. We conjecture that this may be because the Agent Manager is implicitly forced to further distinguish between useful and non-useful information.
>
> Due to the space limit, the generated plans can be found at `/example_plans/plan_with_noises.md` in the anonymous GitHub.
>
> > Adaptability to other ML tasks (e.g., RL or RecSys)
>
> **R6** Please kindly refer to `Reviewer Son5-R5`.

---

### Official Review · Reviewer_Son5 · 2025-03-14

**Overall Recommendation:** 4

**Summary:**

The paper introduces AutoML-Agent, a multi-agent LLM framework that automates the full AutoML pipeline from data retrieval to model deployment. Unlike prior approaches that focus on specific pipeline components (e.g., hyperparameter optimization or feature engineering), AutoML-Agent leverages retrieval-augmented planning (RAP) and multi-stage verification to ensure correctness, efficiency, and adaptability.
The paper presents a modular, multi-agent architecture where specialized agents handle data processing,model selection, hyperparameter tuning, and deployment. The retrieval agumented planning strategy enhances exploration while multi stage verification ensures the correctness of the generated code. Experiments across seven ML tasks and fourteen datasets show that AutoML-Agent outperforms AutoGluon, DS-Agent, GPT-3.5, GPT-4 and human baselines in terms of success rate, normaluzed performance score, and comprehensive score. Additionally, AutoML-Agent is eight times faster than search-based methods like SELA while maintaining compettive performance.

## update after rebuttal
I thank the authors for the additional clarifications and stay with my evaluation that I consider this paper to be an accept for the conference.

**Claims And Evidence:**

AutoML-Agent automates the entire ML pipeline frm data retrieval to deployment: The paper provides an end-to-end framework design with clear descriptions of agents handline each stage (data, model, deployment). Experiemnts show successful deployment-ready models.

The retrievel-augmented planning strategy imprvoes search efficiency: The ablation study shows that RAP + plan decomposition outperforms naive planning, confirming its effectiveness.

The multi-stage verification improves code correctness: Experimental results indicate that verification prevents failueres, reducing deployment errors.

AutoML-Agent is more efficient than searhc-based AutoML methods like SELA: AutoML-Agent achievs similar performance in the experiments while being eight times faster.

**Essential References Not Discussed:**

Hollmann, Noah, et al. "TabPFN: A Transformer That Solves Small Tabular Classification Problems in a Second." The Eleventh International Conference on Learning Representations.

Hollmann, Noah, et al. "Accurate predictions on small data with a tabular foundation model." Nature 637.8045 (2025): 319-326.

**Experimental Designs Or Analyses:**

Experiments span seven different ML tasks, making the results generalizable in general.

The ablation study properly isolates effects of RAP and multi-stage verification.

The hyperparaameter study analyzes how varying the number of plans affects performance.

However, scalability on larlge datasets is not evaluated and the study lacks a statistical significance test in the comparisons with the baselines.

**Methods And Evaluation Criteria:**

The proposed evaluation metrics make sense, as they measure both pipelines correctness and downstream model performance. The benchmark datasets are diverse, ensuring sufficient generalizability. However, scalability testing on large datasets is missing, which could impact real-world applicability.

**Other Comments Or Suggestions:**

The authors should test AutoML-Agent on large-scale datasets and compare to TabPFN on the small scale datasets of tabular data.

**Other Strengths And Weaknesses:**

# Strengths
- The multi-agent design is a novel approach to AutoML
- The full-pipeline AutoML capability is impactful for democratizing ML
- The paper is well-structured and well-written.

# Weaknesses
- Lack of real-world deployment testing.

**Questions For Authors:**

- How does AutoML-Agent scale on large datasets? Would it remain efficient?
- Can AutoML-Agent be extended to reinforcement learning tasks?
- What are the main failure cases of AutoML-Agent?
- How does the framework handle edge cases in code generation? Do hallucinations still occur?

**Relation To Broader Scientific Literature:**

The paper covers traditional AutoML techniques such as AutoGluon, AutoSklearn, TPOT. It also discssues LLM-based AutoML methods and it highlights how AutoML-Agent improves upon past work by incorporating multi-agent collaboration and RAP. However, TabPFN as another very recent AutoML tool is missing in the comparisons.

**Theoretical Claims:**

N/A.

---

> ### Author Rebuttal · Authors · 2025-04-01
>
> Thank you for recognizing our contributions and for your thoughtful, detailed feedback. Below, we respond to your specific concerns.
>
> > Scalability concerns
>
> **R1** This is an important point. A key motivation behind AutoML-Agent is to reduce the computational overhead of the search process. Our framework achieves this by relying solely on LLM inferences for agent communication and leveraging retrieval-augmented knowledge for planning, avoiding expensive training-feedback loops.
>
> `Figure 4c` and `Table 11` demonstrate the scalability of our approach. AutoML-Agent has a time usage standard deviation of only **1 minute**, compared to SELA’s **14 minutes**, across datasets ranging **from ~1K to 143K** instances. As a training-free method, AutoML-Agent maintains stable search time, whereas training-based methods like SELA exhibit fluctuations depending on dataset and model size. This suggests our framework scales efficiently without performance degradation.
>
> More precisely, our computational complexity is approximately $O(1 + m)$, compared to $O(s + m)$ for training-based methods, where $s$ denotes the search-time training feedback and $m$ the model training time before deployment.
>
>
> > Lack of a statistical significance test
>
> **R2** Currently, `Tables 5`, `6`, and `7` report the results with standard deviations over *five* independent runs. While we did not include formal statistical tests, reporting standard deviations is a common practice to convey the reliability of performance differences. Although we believe this is sufficient, please let us know if there is a specific test you would like to see.
>
> > Comparisons with TabPFN
>
> **R3** TabPFN is indeed a highly relevant and efficient method for tabular classification. We would like to clarify that TabPFN is *already included* in our baselines under the **Human Models** category ($\S$C.3). For your convenience, we summarize the results below.
> | **Models**                    | **Banana** | **Software** |
> | ----------------------------- | ---------- | ------------ |
> | Human Models (**TabPFN**)     | 0.976      | 0.669        |
> | AutoGluon                     | 0.980      | 0.524        |
> | GPT-3.5                       | 0.587      | 0.094        |
> | GPT-4                         | 0.390      | 0.285        |
> | DS-Agent                      | 0.766      | 0.523        |
> | ***AutoML-Agent***            | 0.987      | 0.664        |
>
>
> > Lack of real-world deployment testing
>
> **R4** We acknowledge the importance of real-world deployment testing. To this end, we designed our framework with a deployment stage via a Gradio API, showing a proof-of-concept using benchmark datasets. While full deployment in production environments involves additional considerations (e.g., infrastructure, latency, data privacy, and so on), we view this as a promising direction for future work.
>
> > Extension to RL tasks
>
> **R5** AutoML-Agent’s modularity allows for extension to RL tasks. While our current implementation focuses on (un-)supervised pipelines, the framework is not fundamentally limited to these settings. Extending AutoML-Agent to RL would require incorporating domain-specific modules—for example, agents for environment interaction, action space design, and reward handling (see https://www.automl.org/autorl-survey). As noted in the paper, tasks such as RL or RecSys would benefit from additional agents to manage actor-environment interactions and reward modeling.
>
> Thanks to its modular design, one could integrate an “RL agent” that interfaces with an environment (e.g., OpenAI Gym), while the planner generates RL-specific steps such as policy training and evaluation. Although this extension is certainly feasible, it could be substantial enough that an AutoML-Agent for RL—perhaps termed AutoRL-Agent—would merit contributions worthy of a separate paper.
>
> > What are the main failure cases of AutoML-Agent?
>
> **R6** Please kindly refer to `Reviewer mnKA-R3`.
>
> > Edge cases in code generation
>
> **R7** We appreciate the reviewer’s concern regarding LLM hallucinations in code generation. Our framework is specifically designed to mitigate this issue through two core mechanisms: RAP and multi-stage verification.
> - RAP grounds the planning agent’s decisions in real-world information by retrieving external resources, rather than relying solely on the LLM’s internal memory, which can produce hallucinated content. This ensures that generated plan steps are more likely to be valid and based on established approaches.
> - Our multi-stage verification process acts as a safety net. If the LLM produces incorrect or nonsensical code, the implementation stage will flag it, triggering a corrective attempt.
>
> This dual mechanism—prevention through retrieval and correction through verification—has proven effective (please refer to `Reviewer EaoR-R5` for robustness experiments on noisy information). To further reduce risk, we provide agents with structured pipeline skeletons, which help constrain generation.

---

### Official Review · Reviewer_mnKA · 2025-03-16

**Overall Recommendation:** 4

**Summary:**

In this paper, the authors propose a novel multi-agent framework tailored for full-pipeline AutoML, including initialization, planning, and execution, incorporating  5 types of agents, such as Agent Manager, Prompt Agent, Data Agent, Model Agent, and Operation Agent. Results on 14 tasks and 5 baselines demonstrate the stronger performance of the proposed models.

**Claims And Evidence:**

The introduction of the paper is well-written and the challenges of this topic are well-motivated. The authors aim to propose an Auto-ML multiagent system that can take care of both data and model aspects and try to solve the challenges of planning and accurate implementation.

**Essential References Not Discussed:**

Generally sufficient.

**Experimental Designs Or Analyses:**

The experiments are sound because of sufficient datasets, baselines, and experiments. The biggest concern is that the ablation study is not sufficient enough. For instance, is each agent important? How do these workers contribute to the final score? What are the errors or detailed analyses of these workers? Also, how did you design the prompt for each agent? Why not use others? Since the system is rather complex, a very careful analysis is needed to make it clear how it works and how it makes errors.

**Methods And Evaluation Criteria:**

Although the method is complex and not easy to understand the full details. The author provides clear figures and an appendix that helps the read er to understand it.

**Other Comments Or Suggestions:**

See above

**Other Strengths And Weaknesses:**

The paper is well-written and well-motivated.

**Questions For Authors:**

See above

**Relation To Broader Scientific Literature:**

Auto-ML domain. The full-stack Auto-ML using LLM is the main related work of this paper.

**Theoretical Claims:**

N/A

---

> ### Author Rebuttal · Authors · 2025-04-01
>
> We thank the reviewer for the positive comments on the novelty, motivation, and empirical rigor of our work. We are glad to address your concerns below.
>
> > Method complexity
>
> **R1** Thank you for pointing this out. In the final version, which permits an extra page, we will **move key details currently located in the appendix, such as pseudocode summaries, into the main text**. We hope these changes will make the framework easier to understand for a broader audience.
>
> > Is each agent important? How do these workers contribute to the final score?
>
> **R2** Yes, each agent is essential. Without even one of them, the entire framework would not function properly, considering the current implementation. This is due to a tight coupling between each agent, integrated tool use, and its corresponding step in the pipeline at the code level for stable interoperability. As a result, removing any single agent would cause the system to fail at runtime.
>
> Specifically, the *Prompt Agent* ensures that user instructions are correctly interpreted and converted into structured input, enabling reliable downstream planning and execution. The *Data Agent* performs crucial data-related tasks that inform model search with domain-aware characteristics, directly enhancing model quality. The *Model Agent* drives performance by executing training-free model search, hyperparameter tuning, and profiling to select optimal candidates, thereby raising the normalized performance score. The *Operation Agent* ensures that high-performing models are correctly implemented and executable, completing the pipeline with valid, deployable code—vital for success rate metrics. Finally, the *Agent Manager* orchestrates the entire process, ensuring that all steps execute in the correct sequence and that each agent’s output is validated before proceeding, thereby enforcing the correctness of the entire pipeline.
>
>
> > What are the errors or detailed analyses of these workers?
>
> **R3** In this submission, we provide intermediate outputs at each stage ($\S$E) to illustrate what each agent produces—for example, the parsed JSON from the Prompt Agent and outputs from the Data and Model Agents.
>
> Building on this, we will expand our analysis to include typical failure modes. For instance, the Data Agent may mishandle uncommon data formats (e.g., failing to impute missing values or referencing incorrect column names), while the Model Agent may select suboptimal models under constrained search spaces. **Our multi-stage verification process detects and corrects most of these issues.** As shown in $\S4.3$, certain variants fail to produce runnable code without specific modules, whereas the full system identifies and fixes such errors via feedback. The Agent Manager flags plans that violate user constraints (e.g., low accuracy), while implementation verification catches runtime errors (e.g., incomplete code or incorrect references) from the Operation Agent and requests fixes.
>
> Notably, we also encountered consistent challenges with smaller or less capable models. As discussed in $\S$B.4, models like LLaMA-2-7B, Qwen1.5-14B, and even GPT3.5 often failed at complex planning or executable code generation. Common issues included incomplete code, altered comments without proper continuation, or unchanged code outputs. These patterns align with prior findings in DS-Agent and Data Interpreter, suggesting such limitations are systemic to smaller models rather than specific to our framework. We aim to reassure the reviewer that we have carefully studied “how it makes errors” and that the final results are reliable.
>
>
> > How did you design the prompt for each agent?
>
> **R4** We deliberately design prompts to optimize agent behavior with specific instructional steps based on role:
> - Creative or interpretive agents (Agent Manager, Prompt Agent) benefit from low-pressure, neutral framing to avoid overconfidence. The Prompt Agent adopts a neutral “assistant project manager” persona to support coordination and interpretation.
> - Execution-oriented agents (Data, Model, Operation Agents) respond better to strong, authoritative personas that promote precision and task adherence. The Data Agent is framed as “the world’s best data scientist” to encourage confident, detailed analysis and emphasize responsibilities.
>
> These specialized prompts guide the LLM toward more accurate outputs compared to using a single, general-purpose prompt. Persona-driven system prompts reliably shape agent behavior. As suggested by `Reviewer L7kx`, we tested *five* prompt variations for each agent, differing in tone and task specificity. Due to space limitations, the experimental results with example outputs can be found at `/example_plans/prompt_sensitivity.md` in our anonymous GitHub repository. Overall, agents are not highly sensitive to exact phrasing **as long as their roles were clearly defined**, which is also reinforced through the user prompts—making the framework robust to variations in system prompts.

---

### Decision · Program_Chairs · 2025-05-01

**Decision:**

Accept (poster)

**Comment:**

In this submission, the authors present their multi-agent system tailored for AutoML, AutoML-Agent, which can deliver deployment-ready models according to user's task descriptions. Extensive experiments on several downstream tasks also confirm its benefits and great potentials.

During the discussion, all the reviewers appreciate this work. And please incorporate the new experiments and analysis (such as the comparison against other approaches and the analysis of prompt sensitivity) into the final version.

Hope the authors find the discussions with reviewers useful and make this submission a better one.